# Rex: A Family of Reversible Exponential (Stochastic) Runge-Kutta Solvers

**Zander W. Blasingame** [1 2]   **Chen Liu** [2]

## Abstract

Deep generative models based on neural differential equations have become state-of-the-art for many generation tasks. These models rely on ODE/SDE solvers that integrate from a prior distribution to the data distribution; in many applications it is also highly desirable to integrate in the inverse direction. Standard solvers, however, accumulate discretization errors that prohibit *exact inversion*, an inaccuracy that is unacceptable in precision-critical applications. Existing inversion methods suffer from poor stability and low order of convergence, and are strictly limited to the ODE setting. In this work, we propose *Rex*, a family of reversible exponential (stochastic) Runge-Kutta solvers obtained by applying Lawson methods to convert any explicit (stochastic) Runge-Kutta scheme into an algebraically reversible one for both diffusion ODEs *and* SDEs. Beyond a rigorous theoretical analysis—establishing arbitrary-order convergence and a non-zero region of linear stability—we empirically demonstrate that *Rex* achieves near-machine-precision reconstruction and improves Boltzmann sampling with flow models as well as image generation and editing with diffusion models. Our code is available at: https://github.com/zblasingame/Rex-solver

## 1. Introduction

Deep generative models based on *neural differential equations* (Kidger, 2022) have quickly become the state-of-the-art in generation tasks across many varied modalities from image generation (Rombach et al., 2022), protein generation (Skreta et al., 2025b), Boltzmann sampling (Rehman et al., 2026a), and biometrics (Blasingame & Liu, 2024d). These models use the language of a neural Itô *stochastic differen-*

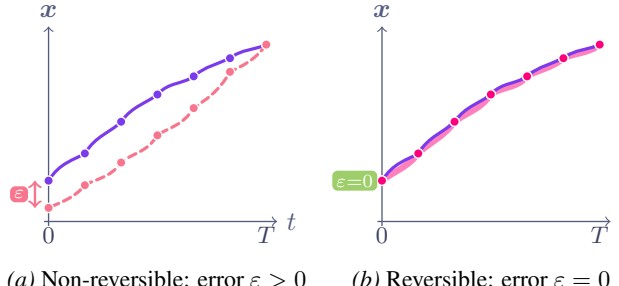

*(a) Non-reversible: error $\varepsilon > 0$*   *(b) Reversible: error $\varepsilon = 0$*

*Figure 1.* Conventional solvers (left) accumulate discretization error $\varepsilon > 0$ when integrated forward (solid) and then backward (dashed), so the reconstructed trajectory drifts from the original. *Rex* (right) is *algebraically reversible*: the backward step (shaded) lands exactly on the forward iterates, yielding $\varepsilon = 0$ regardless of step size or precision.

*tial equation* (SDE) (Kidger et al., 2021) or neural *ordinary differential equation* (ODE) (Chen et al., 2018)—sometimes referred to as a *continuous normalizing flow* (CNF)—to describe the (stochastic) mapping from a prior source distribution to the target data distribution. This can be achieved through a variety of numerical schemes (Lu et al., 2022b; Zhang et al., 2023; Zhang & Chen, 2023; Gonzalez et al., 2024) which integrate from the source to the target distribution. The *exact* inversion of this numerical method, *i.e.*, going from the target distribution back to the source distribution, is invaluable in several key applications where precision is concerned. *E.g.*, gradient descent through these models (Ben-Hamu et al., 2024; Blasingame & Liu, 2024a; McCallum & Foster, 2024) for training, fine-tuning, and differentiable rewards; image editing (Wallace et al., 2023; Wang et al., 2024); and calculating accurate likelihoods of the generative models, enabling sampling from Boltzmann distributions (Rehman et al., 2026a).

Whilst useful, designing such inversion methods is very tricky, as such solvers are plagued by issues of low order of convergence, lack of stability, amongst other undesirable properties; moreover, it is *even* more difficult to construct such schemes for SDEs. Recent work has developed exact inversion methods specifically for diffusion models (Wallace et al., 2023; Zhang et al., 2024; Wang et al., 2024). Unfortunately, these schemes suffer from poor numerical stability which can hamper their real-world utility, particularly in contexts like editing of real samples. Moreover, the

---

[1]AITHYRA [2]Clarkson University. Correspondence to: ZB <zblasingame@aithyra.at>.

*Proceedings of the 43$^{rd}$ International Conference on Machine Learning*, Seoul, South Korea. PMLR 306, 2026. Copyright 2026 by the author(s).

previous approaches do not support the often useful SDE formulation of diffusion models. Prior work (Nie et al., 2024; Wu & la Torre, 2023) has attempted to tackle SDEs, but these methods resort to storing the entire process in memory, which is only a trivial notion of reversibility.

To address these challenges we propose *Rex*, a family of **r**eversible **ex**ponential (stochastic) Runge-Kutta solvers for diffusion models. Our contributions are:

- We construct *Rex*, an *algebraically reversible* family of solvers for both diffusion ODEs and SDEs. The *Rex* ODE solver inherits arbitrary order of convergence and a non-zero region of linear stability from the McCallum-Foster method, and *Rex* supports adaptive step sizes.

- We show that *Rex* is the reversible version of many popular solvers for diffusion models, including DDIM, DPM-Solver, and SEEDS-1.

- We empirically show that *Rex* achieves near-machine-precision reconstruction under exact inversion, while remaining competitive with prior reversible methods on unconditional generation, text-conditioned generation, and image editing.

- We demonstrate that *Rex* enables accurate likelihood-based Boltzmann sampling on tri-alanine.

## 2. Preliminaries

### 2.1. Reversible Solvers

Recently, researchers studying *neural differential equations* have begun to propose several *algebraically reversible solvers*. Consider some prototypical neural ODE of the form $\dot{\boldsymbol{x}}_t = \boldsymbol{u}_\theta(t, \boldsymbol{x}_t)$ with vector field $\boldsymbol{u}_\theta \in \mathcal{C}^r(\mathbb{R} \times \mathbb{R}^d; \mathbb{R}^d)$ which satisfies the usual regularity conditions. Then consider a single-step numerical scheme of the form $\boldsymbol{x}_{n+1} = \boldsymbol{x}_n + \boldsymbol{\Phi}_h(t_n, \boldsymbol{x}_n, \boldsymbol{u}_\theta)$. Every numerical scheme $\boldsymbol{\Phi}$ is reversible in the sense that we can rewrite the forward step as an implicit scheme of the form $\boldsymbol{x}_n = \boldsymbol{x}_{n+1} - \boldsymbol{\Phi}_h(t_n, \boldsymbol{x}_n, \boldsymbol{u}_\theta)$; however, this requires fixed point iteration[1] and is both *approximate* and computationally *expensive*. This type of reversibility is known as *analytic reversibility* within the neural differential equations community (Kidger, 2022, Section 5.3.2.1). What we would prefer, however, is a form of reversibility that can be expressed in *closed-form*.

There are only a few such *algebraically reversible* solvers which have been proposed within the last few years (Zhuang et al., 2021; Kidger et al., 2021; McCallum & Foster, 2024) and only one of these works for SDEs, namely, the work of Kidger et al. (2021). Whilst only for ODEs the recent work of McCallum & Foster (2024) is highly interesting as it is the *only* algebraically reversible scheme with a non-

---

[1]If the step size $h$ is small enough.

zero region of stability. We refer to the method proposed in McCallum & Foster (2024) as the *McCallum-Foster* method and summarize it below in Definition 2.1.

**Definition 2.1.** Initialize $\hat{\boldsymbol{x}}_0 = \boldsymbol{x}_0$ and let $\zeta \in (0, 1]$. Consider a step size of $h$, then a forward step of the McCallum-Foster method is defined as

$$\boldsymbol{x}_{n+1} = \zeta \boldsymbol{x}_n + (1 - \zeta)\hat{\boldsymbol{x}}_n + \boldsymbol{\Phi}_h(t_n, \hat{\boldsymbol{x}}_n), \quad (1a)$$

$$\hat{\boldsymbol{x}}_{n+1} = \hat{\boldsymbol{x}}_n - \boldsymbol{\Phi}_{-h}(t_{n+1}, \boldsymbol{x}_{n+1}), \quad (1b)$$

and the backward step is given as

$$\hat{\boldsymbol{x}}_n = \hat{\boldsymbol{x}}_{n+1} + \boldsymbol{\Phi}_{-h}(t_{n+1}, \boldsymbol{x}_{n+1}), \quad (2a)$$

$$\boldsymbol{x}_n = \zeta^{-1}\boldsymbol{x}_{n+1} + (1 - \zeta^{-1})\hat{\boldsymbol{x}}_n - \zeta^{-1}\boldsymbol{\Phi}_h(t_n, \hat{\boldsymbol{x}}_n). \quad (2b)$$

### 2.2. Diffusion Models

Diffusion models (Sohl-Dickstein et al., 2015; Ho et al., 2020; Song et al., 2021a;b) have become one of the most popular paradigms for constructing generative models. Consider the following Itô *stochastic differential equation* (SDE) defined on time interval $[0, T]$:

$$\mathrm{d}\boldsymbol{X}_t = f(t)\boldsymbol{X}_t \, \mathrm{d}t + g(t) \, \mathrm{d}\boldsymbol{W}_t, \quad (3)$$

where $f, g \in \mathcal{C}^\infty([0, T])$[2] form the drift and diffusion coefficients of the SDE and where $\{\boldsymbol{W}_t\}_{t \in [0,T]}$ is the standard Brownian motion on the time interval. The coefficients $f, g$ are chosen such that the SDE maps clean samples from the data distribution $\boldsymbol{X}_0 \sim q(\boldsymbol{X})$ at time 0 to an isotropic Gaussian at time $T$. More specifically, for a *noise schedule* $\alpha_t, \sigma_t \in \mathcal{C}^\infty([0, T]; \mathbb{R}_{\geq 0})$ consisting of a strictly monotonically decreasing function $\alpha_t$ and strictly monotonically increasing function $\sigma_t$, the drift and diffusion coefficients are given by

$$f(t) = \frac{\dot{\alpha}_t}{\alpha_t}, \qquad g^2(t) = \dot{\sigma}_t^2 - 2\frac{\dot{\alpha}_t}{\alpha_t}\sigma_t^2, \quad (4)$$

where, with abuse of notation, $\dot{\sigma}_t^2$ denotes the time derivative of the function $\sigma_t^2$ (Lu et al., 2022b; Kingma et al., 2021)—this ensures that $\boldsymbol{X}_t \sim \mathcal{N}(\alpha_t \boldsymbol{X}_0, \sigma_t^2 \boldsymbol{I})$. However, we wish to map from *noise* back to *data*; as such, we employ the result of Anderson (1982) to construct the *reverse-time* diffusion SDE of Equation (3), which is given by

$$\mathrm{d}\boldsymbol{X}_t = [f(t)\boldsymbol{X}_t - g^2(t)\nabla_{\boldsymbol{x}} \log p_t(\boldsymbol{X}_t)] \, \mathrm{d}t + g(t) \, \mathrm{d}\overline{\boldsymbol{W}}_t, \quad (5)$$

where $\mathrm{d}t$ is a *negative* timestep, $\{\overline{\boldsymbol{W}}_t\}_{t \in [0,T]}$ is the standard Brownian motion in reverse-time, and $p_t(\boldsymbol{x}) := p(t, \boldsymbol{x})$ is the marginal density function. Then, if we can learn the *score function* $(t, \boldsymbol{x}) \mapsto \nabla_{\boldsymbol{x}} \log p_t(\boldsymbol{x})$ (Song et al., 2021b)— or some other *equivalent* reparameterization, *e.g.*, noise

---

[2]We let $\mathcal{C}^r(X; Y)$ denote the class of $r$-th differentiable functions from $X$ to $Y$. If $Y$ is omitted then $Y = \mathbb{R}$.

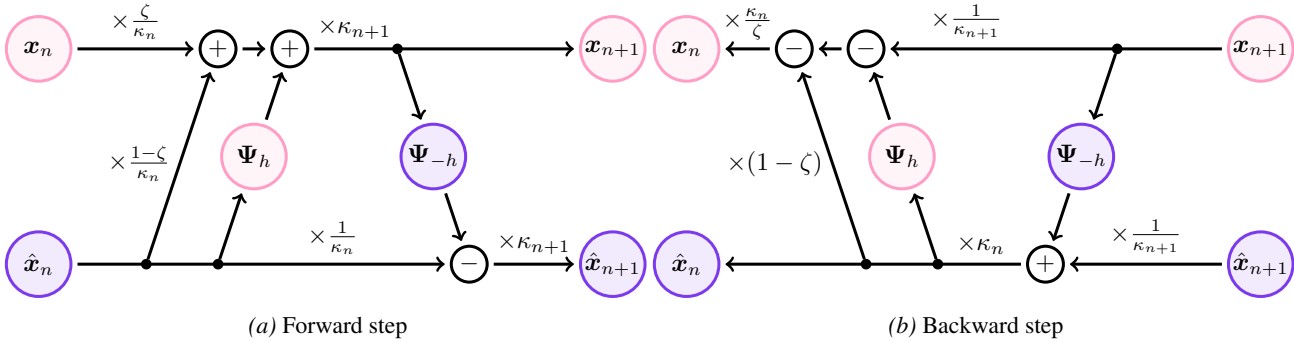

*(a)* Forward step                                              *(b)* Backward step

*Figure 2.* An overview of the *Rex* solver. Here $\boldsymbol{\Psi}_h$ denotes the *Princeps* scheme (see Section 3.1), $\zeta \in (0, 1)$ is a coupling parameter, and $\{\kappa_n\}_{n=1}^{N}$ denotes the set of weighting variables derived from the exponential schemes. The particular values of $\kappa_n$ are discussed in Proposition 3.2. The visualization of the computation graph is inspired by McCallum & Foster (2024, Figure 2).

prediction (Song et al., 2021a; Ho et al., 2020) or data prediction (Kingma et al., 2021)—we can then draw samples from our data distribution $q(\boldsymbol{X})$ by first sampling some $\boldsymbol{X}_T \sim p(\boldsymbol{X})$ from the Gaussian prior and then employing a numerical SDE solver, *e.g.*, Euler-Maruyama, to solve Equation (5) in reverse-time. Notably, through careful massaging of the Fokker-Planck-Kolmogorov equation for the marginal density, one can construct an ODE which is equivalent in *distribution* to Equation (5) (Song et al., 2021b; Maoutsa et al., 2020), yielding the popular *probability flow ODE*

$$\frac{\mathrm{d}\boldsymbol{x}_t}{\mathrm{d}t} = f(t)\boldsymbol{x}_t - \frac{g^2(t)}{2}\nabla_{\boldsymbol{x}} \log p_t(\boldsymbol{x}_t). \quad (6)$$

## 3. Rex: A Family of Reversible Exponential (Stochastic) Runge-Kutta Solvers

In this section we introduce the *Rex* family of reversible exponential Runge-Kutta solvers. This family of *bespoke* numerical schemes is specifically tailored to exploit the semi-linear structure of the diffusion ODE/SDE. The elegance of *Rex* is that it can be built rather generally from many popular pre-existing ODE/SDE solvers. Let $\boldsymbol{\Phi}$ denote our explicit (S)RK scheme of choice, *e.g.*, Euler or the Dormand-Prince method. Then we massage the probability flow ODE and reverse-time SDE into a sufficiently *nice* form and apply $\boldsymbol{\Phi}$—we refer to this construction as *Princeps*, $\boldsymbol{\Psi}$. Lastly, we construct a reversible scheme from this nice form of $\boldsymbol{\Psi}$. We summarize this three-step recipe below.

> **Rex recipe.**
>
> 1. $\boldsymbol{\Phi}$: select an explicit (S)RK scheme.
>
> 2. $\boldsymbol{\Psi}$: build *Princeps* from the RK scheme.
>
> 3. $\boldsymbol{\Upsilon}$: build *Rex* out of *Princeps*.

In Figure 2 we present an overview of the *Rex* computational graph. The graph is identical for both the ODE and SDE formulations; only the weighting terms $\{\kappa_n\}$ and the underlying numerical scheme $\boldsymbol{\Psi}_h$ differ. The rest of this section is organized as follows: first we construct the *Princeps* scheme $\boldsymbol{\Psi}$ from some explicit (S)RK scheme $\boldsymbol{\Phi}$ in Section 3.1, then we construct *Rex*, $\boldsymbol{\Upsilon}$, from *Princeps* in Section 3.2, and lastly in Section 3.3 we discuss the theoretical properties of *Rex*.

### 3.1. Princeps

In this section we discuss how to build the underlying scheme, $\boldsymbol{\Psi}$, from which we construct the reversible *Rex* scheme. As this scheme is *one that is first* we hereafter refer to it as the *Princeps* scheme. For simplicity of presentation we derive *Princeps* from the reverse-time diffusion SDE in Equation (5); however, this framework generalizes the *probability flow* ODE formulation, see Equation (6), with the explicit derivations detailed in Appendix C.1.

As the data and noise prediction formulations differ we will derive *Princeps* from a rather general view of the problem and elide the particular details, reserving them for Appendix C.2. Recall that the Itô SDE in Equation (5) has a semi-linear drift term and additive noise, this nice structure allows us to greatly simplify our discussion. We will rewrite Equation (5) as

$$\mathrm{d}\boldsymbol{X}_t = [a(t)\boldsymbol{X}_t + b(t)\boldsymbol{f}_\theta(t, \boldsymbol{X}_t)]\,\mathrm{d}t + g(t)\,\mathrm{d}\overline{\boldsymbol{W}}_t, \quad (7)$$

where $a(t), b(t)$ are the appropriate generalizations for the data and noise parameterizations and $\boldsymbol{f}_\theta$ denotes either the noise or data prediction model (see Appendix C; for background see Section 2.2). This form can then be simplified via exponential integrators, *i.e.*, $\exp -\int_0^t a(\tau)\,\mathrm{d}\tau$, and then a suitable change of variables, the result of which we show in Proposition 3.1 with the proof in Appendix C.2.1.

$$\mathrm{d}\boldsymbol{X}_t = [a(t)\boldsymbol{X}_t + b(t)\boldsymbol{f}_\theta(t, \boldsymbol{X}_t)]\ \mathrm{d}t + g(t)\ \mathrm{d}\boldsymbol{W}_t \xrightarrow[\text{change-of-variables}]{\text{Exponential integrators \&}} \mathrm{d}\boldsymbol{Y}_\varsigma = \boldsymbol{f}_\theta(\varsigma, \Xi(\varsigma)\boldsymbol{Y}_\varsigma) + \mathrm{d}\boldsymbol{W}_\varsigma$$

$$\boldsymbol{X}_{n+1} = \frac{\Xi(t_n)}{\Xi(t_{n+1})}\boldsymbol{X}_n + \boldsymbol{\Psi}_h(t_n, \boldsymbol{X}_n, \boldsymbol{W}_n(\omega)) \xleftarrow{\text{Lawson method}} \boldsymbol{Y}_{n+1} = \boldsymbol{Y}_n + \boldsymbol{\Phi}_h(\varsigma_n, \boldsymbol{Y}_n, \boldsymbol{W}_n(\omega))$$

*Figure 3.* Construction of the *Princeps* $\boldsymbol{\Psi}$ for the diffusion SDE in Equation (7) from an underlying explicit stochastic Runge-Kutta scheme $\boldsymbol{\Phi}$; the probability flow ODE case follows *mutatis mutandis*.

**Proposition 3.1.** *Assume that* $g(t) = \sqrt{a(t)b(t)}$, $b(t)/a(t) \to \infty$ *as* $t \to \infty$, *and* $b(t) > 0$. *Let* $\Xi(t) \coloneqq \exp\left(\int_0^t a(\tau)\ \mathrm{d}\tau\right)$ *be the reciprocal of the integrating factor of Equation* (7). *Then the SDE in Equation* (7) *can be rewritten as*

$$\mathrm{d}\boldsymbol{Y}_\varsigma = \boldsymbol{f}_\theta(\varsigma, \Xi(\varsigma)\boldsymbol{Y}_\varsigma)\ \mathrm{d}\varsigma + \mathrm{d}\boldsymbol{W}_\varsigma, \qquad (8)$$

*where* $\boldsymbol{Y}_t = \Xi^{-1}(t)\boldsymbol{X}_t$ *and* $\varsigma_t = \int \Xi^{-1}(t)b(t)\ \mathrm{d}t$.

**Remark 3.1.** In Appendices C.2.2 and C.2.3 we provide the particular realizations of the time-changed SDE for the data and noise parametrizations.

Proposition 3.1 highlights the first half of constructing *Princeps* and is the upper pathway in Figure 3. The rest of the section is devoted to constructing the lower pathway, *i.e.*, how we create the exponentially weighted Stochastic Runge-Kutta scheme. The next question is which stochastic Runge-Kutta formulation to use, as unlike in the ODE case there are many possible different formulations to choose from.

**Stochastic Runge-Kutta.** Constructing a numerical scheme for SDEs is greatly more complicated than ODEs due to the complexities of stochastic processes and in particular stochastic integrals. Unlike numerical schemes for ODEs which are usually built upon truncated Taylor expansions, SDEs require constructing truncated Itô or Stratonovich-Taylor expansions (Kloeden & Platen, 1991) which results in numerous iterated stochastic integrals. Approximating these iterated integrals, or equivalently Lévy areas, of Brownian motion is quite difficult (Clark & Cameron, 2005; Mrongowius & Rößler, 2022); however, SDEs with certain constraints on the diffusion term may use specialized solvers to further achieve a strong order of convergence with simple approximations of these iterated stochastic integrals. As such there are several ways to express SRK methods depending on the choice of approximating these iterated integrals. We choose to follow the work of Foster et al. (2024) which makes usage of the *space-time Lévy area* in constructing such methods. The space-time Lévy area (see Foster et al., 2020, Definition 3.5; *cf.* Rößler, 2010) is

defined below in Definition 3.2.

**Definition 3.2** (Space-time Lévy area). The rescaled space-time Lévy area of a Brownian motion $\{W_t\}$ on the interval $[s, t]$ corresponds to the signed area of the associated bridge process

$$H_{s,t} \coloneqq \frac{1}{h}\int_s^t \left(W_{s,u} - \frac{u-s}{h}W_{s,t}\right)\ \mathrm{d}u, \qquad (9)$$

where $h \coloneqq t - s$ and $W_{s,u} = W_u - W_s$ for $u \in [s, t]$.

In particular, for additive-noise SDEs which our SDE in Equation (8) is, the Itô and Stratonovich integrals coincide and the numerical scheme is significantly simpler, for more details we refer to Appendix B.

For the Itô SDE in Equation (8) we follow the conventions of Foster et al. (2024) and write an $s$-stage SRK as below; this form is a direct generalization of Foster et al. (2024, Equation (6.1)) to an arbitrary extended Butcher tableau (see the `diffrax` documentation).

$$\boldsymbol{f}_\theta^i = \boldsymbol{f}_\theta(\varsigma_n + c_i h, \Xi(\varsigma_n + c_i h)\boldsymbol{Z}_i), \qquad (10\text{a})$$

$$\boldsymbol{Z}_i = \boldsymbol{Y}_n + h\left(\sum_{j=1}^{i-1} a_{ij}\boldsymbol{f}_\theta^j\right) + a_i^W \boldsymbol{W}_n + a_i^H \boldsymbol{H}_n, \qquad (10\text{b})$$

$$\boldsymbol{Y}_{n+1} = \boldsymbol{Y}_n + h\left(\sum_{i=1}^{s} b_i \boldsymbol{f}_\theta^i\right) + b^W \boldsymbol{W}_n + b^H \boldsymbol{H}_n. \qquad (10\text{c})$$

where $h = \varsigma_{n+1} - \varsigma_n$ is the step size and $\boldsymbol{W}_n \coloneqq \boldsymbol{W}_{t_n, t_{n+1}}$ and $\boldsymbol{H}_n \coloneqq \boldsymbol{H}_{t_n, t_{n+1}}$ are the Brownian and Lévy increments respectively; and where $a_{ij}, a_i^W, a_i^H \in \mathbb{R}^{s \times s}$, $b_i, b^W, b^H \in \mathbb{R}^s$, and $c_i \in \mathbb{R}^s$ are the coefficients of an *extended* Butcher tableau (*cf.* Rößler, 2025; Foster et al., 2024); *cf.* the deterministic ODE setting in Stewart (2022, Section 6.1.4). Then by straightforward substitution for the identity of $\boldsymbol{Y}$ we arrive at the *Princeps* scheme denoted $\boldsymbol{\Psi}$ in Equation (11) below (see Appendices C.2.2 and C.2.3 for

the explicit data and noise prediction realizations).

$$\boldsymbol{f}_\theta^i = \boldsymbol{f}_\theta(\varsigma_n + c_i h, \Xi(\varsigma_n + c_i h)\boldsymbol{Z}_i), \tag{11a}$$

$$\boldsymbol{Z}_i = \Xi^{-1}(\varsigma_n)\boldsymbol{X}_n + h\left(\sum_{j=1}^{i-1} a_{ij}\boldsymbol{f}_\theta^j\right) \tag{11b}$$
$$+ a_i^W \boldsymbol{W}_n + a_i^H \boldsymbol{H}_n,$$

$$\boldsymbol{X}_{n+1} = \frac{\Xi(\varsigma_{n+1})}{\Xi(\varsigma_n)}\boldsymbol{X}_n$$
$$+ \Xi(\varsigma_{n+1})\underbrace{\left[h\left(\sum_{i=1}^{s} b_i \boldsymbol{f}_\theta^i\right) + b^W \boldsymbol{W}_n + b^H \boldsymbol{H}_n\right]}_{=:\boldsymbol{\Psi}}, \tag{11c}$$

from which we will build the *Rex* scheme.

## 3.2. Rex

Equipped with Equation (11) we are now ready to construct *Rex*. The key idea is to construct a reversible scheme from an explicit (S)RK scheme (we provide more detail in Appendix B) for the reparameterized differential equation using the McCallum-Foster method and then apply Lawson methods to bring the scheme back to the original state variable; see Figure 3. We provide a brief summary below; the full derivation is in Appendix C.3.

> **Proposition 3.2** (*Rex*). *Without loss of generality let $\boldsymbol{\Phi}$ denote an explicit SRK scheme for the SDE in Equation* (8) *with extended Butcher tableau $a_{ij}, b_i, c_i, a_i^W, a_i^H, b^W, b^H$. Fix an $\omega \in \Omega$ and let $\boldsymbol{W}$ be the Brownian motion over time variable $\varsigma$. Then the reversible solver constructed from $\boldsymbol{\Phi}$ in terms of the underlying state variable $\boldsymbol{X}_t$ is given by the forward step*
>
> $$\boldsymbol{X}_{n+1} = \frac{\kappa_{n+1}}{\kappa_n}\left(\zeta\boldsymbol{X}_n + (1-\zeta)\hat{\boldsymbol{X}}_n\right)$$
> $$+ \kappa_{n+1}\boldsymbol{\Psi}_h(\varsigma_n, \hat{\boldsymbol{X}}_n, \boldsymbol{W}_n(\omega)),$$
> $$\hat{\boldsymbol{X}}_{n+1} = \frac{\kappa_{n+1}}{\kappa_n}\hat{\boldsymbol{X}}_n - \kappa_{n+1}\boldsymbol{\Psi}_{-h}(\varsigma_{n+1}, \boldsymbol{X}_{n+1}, \boldsymbol{W}_n(\omega)), \tag{12}$$
>
> *and backward step*
>
> $$\hat{\boldsymbol{X}}_n = \frac{\kappa_n}{\kappa_{n+1}}\hat{\boldsymbol{X}}_{n+1} + \kappa_n\boldsymbol{\Psi}_{-h}(\varsigma_{n+1}, \boldsymbol{X}_{n+1}, \boldsymbol{W}_n(\omega)),$$
> $$\boldsymbol{X}_n = \frac{\kappa_n}{\kappa_{n+1}}\zeta^{-1}\boldsymbol{X}_{n+1} + (1-\zeta^{-1})\hat{\boldsymbol{X}}_n$$
> $$- \kappa_n\zeta^{-1}\boldsymbol{\Psi}_h(\varsigma_n, \hat{\boldsymbol{X}}_n, \boldsymbol{W}_n(\omega)), \tag{13}$$
>
> *with step size $h := \varsigma_{n+1} - \varsigma_n$ and where $\boldsymbol{\Psi}$ follows Equation (11) with $\kappa_n = \Xi(\varsigma_n)$.*

**Remark 3.3.** We can recover all four cases with the appropriate choice of weighting coefficient and time variable. For data prediction SDEs we have, $\kappa_n = \frac{\sigma_n}{\gamma_n}$ and $\varsigma_t = \frac{\alpha_n^2}{\sigma_n^2}$; the ODE case is recovered for an explicit RK scheme with $\kappa_n = \sigma_n$ and $\varsigma_t = \frac{\alpha_n}{\sigma_n}$. For noise prediction models we have $\boldsymbol{f}_\theta$ denoting the noise prediction model with $\kappa_n = \alpha_n$ and $\varsigma_t = \frac{\sigma_n}{\alpha_n}$. The Brownian motion has a slightly different time-change in the noise prediction formulation; we defer the nuances to Appendix C.2.3.

We now justify the design choices in Proposition 3.2, in particular our handling of stochasticity. The key idea is to use the *same* realization of the Brownian motion in both the forward pass and backward pass. This has been explored in prior works studying the continuous adjoint equations for neural SDEs (Li et al., 2020; Kidger et al., 2021) and essentially amounts to fixing the realization of the Brownian motion along with clever strategies for reconstructing the same realization of the Brownian motion.

**Numerical Simulation of the Brownian Motion.** The naïve approach to fixing the realization is to cache the entire trajectory, which is both expensive and prohibits adaptive step-size solvers. Instead, recent work by Li et al. (2020); Kidger et al. (2021); Jelinčič et al. (2024) enables one to recalculate *any* realization of the Brownian motion from a single seed given access to a splittable *pseudo-random number generator* (PRNG) (Salmon et al., 2011); we adopt this approach and discuss the technical details in Appendix F.3.

## 3.3. Properties of Rex

**Convergence Order.** A nice property of the McCallum-Foster method is that the convergence order of the underlying explicit RK scheme $\boldsymbol{\Phi}$ is inherited by the resulting reversible scheme (McCallum & Foster, 2024, Theorem 2.1). However, does this property hold true for *Rex*? We show that *Rex* can achieve an arbitrarily high order of convergence in Theorem 3.3 with the proof provided in Appendix D.2.

> **Theorem 3.3** (*Rex* is a $k$-th order solver). *Let $\boldsymbol{\Phi}$ be a $k$-th order explicit Runge-Kutta scheme for the reparameterized probability flow ODE in Equation* (87) *with variance preserving noise schedule $(\alpha_t, \sigma_t)$. Then Rex constructed from $\boldsymbol{\Phi}$ is a $k$-th order solver, i.e., given the reversible solution $\{\boldsymbol{x}_n, \hat{\boldsymbol{x}}_n\}_{n=1}^N$ and true solution $\boldsymbol{x}_{t_n}$ we have*
>
> $$\|\boldsymbol{x}_n - \boldsymbol{x}_{t_n}\| \leq Ch^k, \tag{14}$$
>
> *for constants $C, h_{max} > 0$ and for step sizes $h \in [0, h_{max}]$.*

This result is for the ODE case; we refer to the generalized formulation of the probability flow ODE in Equation (87). Therefore, *Rex* inherits the convergence order of the base

scheme, $\Phi$, used to construct *Princeps*. As a clear corollary we have

**Corollary 3.3.1.** *Let $\Phi$ be a $k$-th order explicit Runge-Kutta scheme for the reparameterized probability flow ODE in Equation (87) with variance preserving noise schedule $(\alpha_t, \sigma_t)$. Then Princeps constructed from $\Phi$ is a $k$-th order solver.*

An analogous result can be shown for the *Princeps* scheme for diffusion SDEs, *i.e.*, that it inherits the strong order of convergence from the underlying SRK scheme $\Phi$. We show this below with the full proof provided in Appendix D.3.

> **Theorem 3.4** (Convergence order for *Princeps*)**.** *Let $\Phi$ be a SRK scheme with strong order of convergence $\xi > 0$ for the reparameterized reverse-time diffusion SDE in Equation (8) with variance preserving noise schedule $(\alpha_t, \sigma_t)$ and $\alpha_T > 0$. Then $\Psi$ constructed from $\Phi$ has strong order of convergence $\xi$.*

**Relation to Existing Solvers.** Next we show that several variants of *Rex* are actually the *reversible versions* of several well-known solvers in the literature for diffusion models, *e.g.*, the DPM-Solvers (Lu et al., 2022b). We first begin by showing that *Princeps* subsumes many popular previous solvers developed for diffusion models in Theorem 3.5 with the full derivations in Appendix E.

> **Theorem 3.5** (*Princeps* subsumes previous solvers)**.** *Princeps subsumes the following solvers for diffusion models*
> 1. *DDIM (Song et al., 2021a),*
> 2. *DPM-Solver-1, DPM-Solver-2, DPM-Solver-12 (Lu et al., 2022b),*
> 3. *DPM-Solver++1, DPM-Solver++(2S), SDE-DPM-Solver-1, SDE-DPM-Solver++1 (Lu et al., 2022a),*
> 4. *SEEDS-1 (Gonzalez et al., 2024), and*
> 5. *gDDIM (Zhang et al., 2023).*

As a natural corollary we have that *Rex* is then the reversible version of these highly popular solvers for diffusion models, *i.e.*, we have that *Rex* (Euler) is *Reversible DDIM &c.*

**Corollary 3.5.1.** *Rex is the reversible version of the well-known solvers for diffusion models in Theorem 3.5.*

**Stability.** One drawback of reversible solvers is their rather unimpressive stability, in fact until the work of Mc-Callum & Foster (2024) there were no reversible methods which had a non-zero region of stability. We discuss this in more detail in Appendix A.2 along with illustrating the poor stability characteristics of BDIA and O-BELM (see Corollaries A.4.1 and A.3.2). However, since *Rex* is built upon the McCallum-Foster method the ODE solver has a non-zero

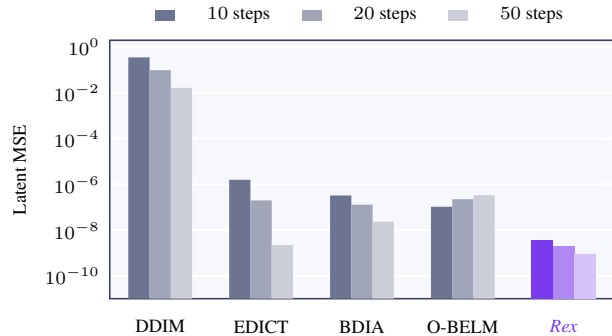

*Figure 4.* Latent-space reconstruction MSE at fp32 precision (log scale; lower is better) for Stable Diffusion v1.5 ($512 \times 512$, CFG 1.0) over 100 real images. *Rex* (Euler) is orders of magnitude below all baselines. Full numbers in Table 4.

region of stability.[3] This property will prove valuable later in our empirical studies.

## 4. Empirical Results

In this section we conduct a number of empirical studies to illustrate the utility of *Rex* in a wide variety of contexts. We primarily explore this in two contexts: 1) image generation and editing with exact inversion, and 2) ensuring invertibility for accurate likelihood calculations for Boltzmann sampling. Unless stated otherwise $\zeta = 0.999$ for all experiments (see Appendix A.2 for the rationale behind this choice).

### 4.1. Reconstruction Error Under Finite Precision

While *Rex* is algebraically reversible, finite-precision arithmetic on GPUs can in principle break strict reversibility. We measure the round-trip reconstruction error (forward solve followed by reverse solve) over 100 real images from the `pix2pix` dataset encoded with Stable Diffusion v1.5 at $512 \times 512$, reporting latent-space MSE to exclude VAE reconstruction error from the measurement. Figure 4 summarizes the fp32 results across 10, 20, and 50 steps with CFG scale 1.0; *Rex* (Euler) is the most accurate solver at every step count, by one to several orders of magnitude over most baselines. Notably, O-BELM's error *grows* with steps—consistent with its lack of a non-trivial linear stability region (see Appendix A.2). Full results (fp16 and pixel-space MSE) and additional discussion are deferred to Appendix H.1.

### 4.2. Image Generation

#### 4.2.1. UNCONDITIONAL IMAGE GENERATION

Following prior works (Wang et al., 2024; Wallace et al., 2023), we begin by evaluating *Rex* as a standard sampling solver for diffusion models. To evaluate this we drew $10^4$

---

[3]*I.e.*, in the sense of the linear test equation, see Appendix A.2 for more details.

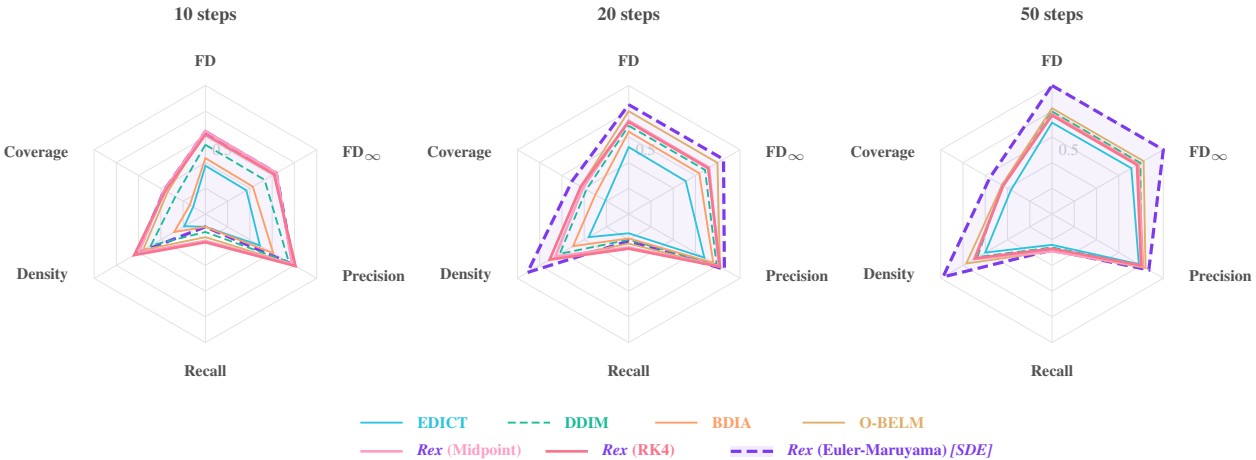

*Figure 5.* Radar charts comparing reversible solvers for unconditional image generation on CelebA-HQ ($256 \times 256$) with a pre-trained DDPM at 10, 20, 50 steps. Six metrics (FD, $\text{FD}_\infty$, Precision, Recall, Density, Coverage); *Rex* (Euler-Maruyama, mauve) attains the largest polygon at 20 and 50 steps. Raw values in Table 5.

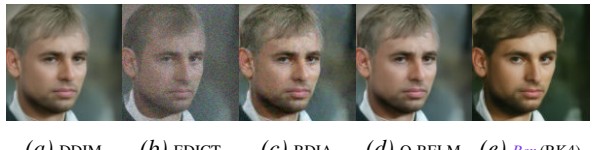

*(a)* DDIM    *(b)* EDICT    *(c)* BDIA    *(d)* O-BELM    *(e)* *Rex* (RK4)

*Figure 6.* Qualitative comparison of unconditional sampling on CelebA-HQ ($256 \times 256$) with a pre-trained DDPM at 10 discretization steps; non-reversible DDIM included as a baseline.

samples using a DDPM model (Ho et al., 2020) pre-trained on the CelebA-HQ (Karras et al., 2018) dataset with the various solvers each using the same fixed seed. Following Stein et al. (2023), we report the *Fréchet distance* (FD) using a DINOv2 (Oquab et al., 2023) feature extractor along with $\text{FD}_\infty$ (Chong & Forsyth, 2020). We also report precision and recall (Kynkäänniemi et al., 2019) along with density and coverage (Naeem et al., 2020), which together proxy fidelity and sample diversity.

In Table 5 we compare pre-existing methods for exact inversion with diffusion models against *Rex*, and include the non-reversible DDIM (Song et al., 2021a) solver as a baseline. The *Rex* family performs strongly, outperforming EDICT (Wallace et al., 2023) and BDIA (Zhang et al., 2024) by a wide margin and frequently outperforming O-BELM (Wang et al., 2024); *Rex* even surpasses the non-reversible DDIM baseline. Our reversible SDE scheme performs well outside of the small step-size regime, a well-known limitation of SDE schemes. Unlike the other reversible solvers, *Rex*'s hyperparameters were not tuned for this benchmark. In Figure 6 we present a visual qualitative comparison of the different solvers using the same initial noise. We provide additional experimental details in Appendix G.1.

For a discussion of why higher-order *Rex* (RK4) underper-

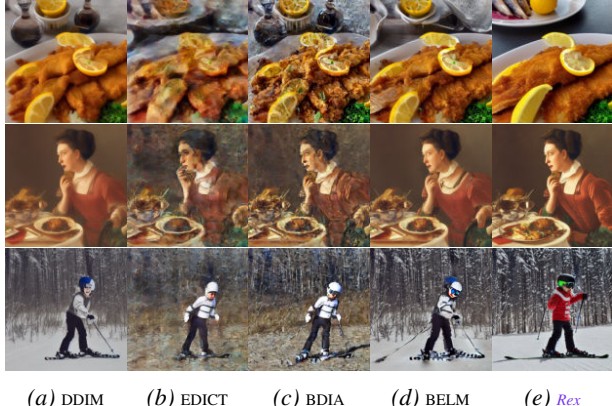

*(a)* DDIM    *(b)* EDICT    *(c)* BDIA    *(d)* BELM    *(e)* *Rex*

*Figure 7.* Qualitative comparison of text-to-image sampling with Stable Diffusion v1.5 ($512 \times 512$) at 10 discretization steps. Prompts (top to bottom): "White plate with fried fish and lemons sitting on top of it.", "A lady enjoying a meal of some sort.", "A young boy riding skis with ski poles.".

forms lower-order *Rex* (Euler) and the SDE variant on this benchmark, see Appendix H.2.

### 4.2.2. CONDITIONAL IMAGE GENERATION

To further evaluate *Rex* we drew text-conditioned samples using Stable Diffusion v1.5 (Rombach et al., 2022) from 1000 randomly selected COCO captions (Lin et al., 2014), with the various solvers each using the same fixed seed. We report performance in terms of the CLIP Score (Hessel et al., 2021), the state-of-the-art text-to-image scoring function PickScore (Kirstain et al., 2023), and the Image Reward metric (Xu et al., 2023), which assigns a score reflecting human preferences—namely, aesthetic quality and prompt adherence—and has recently become a popular evaluation metric for diffusion models (Skreta et al., 2025a). In Table 6

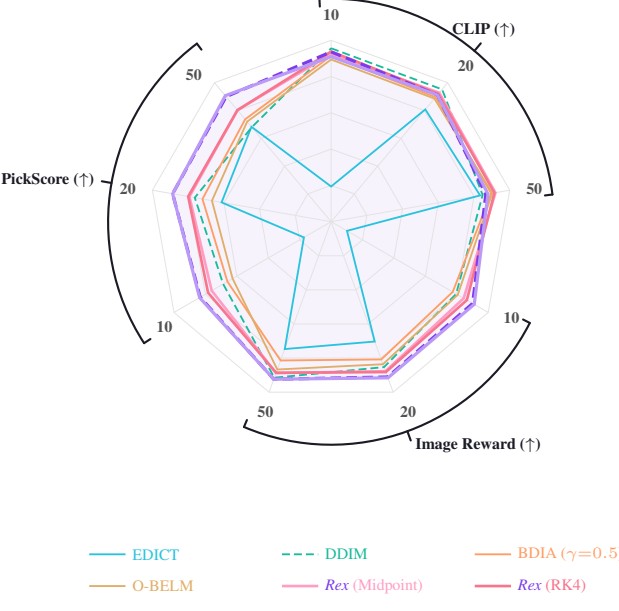

*Figure 8.* Radar chart comparing reversible solvers for text-to-image generation with Stable Diffusion v1.5 ($512 \times 512$) on 1000 COCO captions. Three metrics (CLIP score, Image Reward, PickScore), each split across 10/20/50 sampling steps; axes normalised to CLIP $\in [27, 32]$, Image Reward $\in [-1.4, 0.4]$, PickScore $\in [19, 22]$. Raw values in Table 6.

we compare pre-existing methods for exact inversion with diffusion models against *Rex*, and include the non-reversible DDIM solver as a baseline. We observe that every *Rex* variant outperforms the other reversible solvers across all three metrics; the stochastic variants (Euler-Maruyama and ShARK) lead on Image Reward and PickScore. In Figure 7 we present a visual qualitative comparison of the different solvers using the same initial noise. We provide additional experimental details in Appendix G.2.

### 4.3. Image Editing

A central application of exact-inversion solvers is *round-trip image editing*: given a real image $x_0$ and a source caption $c_{\text{src}}$, the solver inverts $x_0$ to a latent $x_T$, after which the diffusion model is re-sampled from $x_T$ under a new edit caption $c_{\text{edit}}$ to produce an edited image $x_0'$. The quality of the edit depends critically on the fidelity of the inversion: any reconstruction error propagates into spurious changes to regions that the prompt does not target. We follow Brooks et al. (2023) and use the `pix2pix` dataset, in which each example pairs an image with both a source description (*e.g.*, "a man riding a horse") and an editing instruction (*e.g.*, "have him ride a dragon"). For each pair we invert $x_0$ to time $t = 0.6$ using $c_{\text{src}}$ and then re-sample back to $t = 0$ using $c_{\text{edit}}$. In addition to the CLIP Score, Image Reward, and PickScore metrics used previously, we evaluate LPIPS

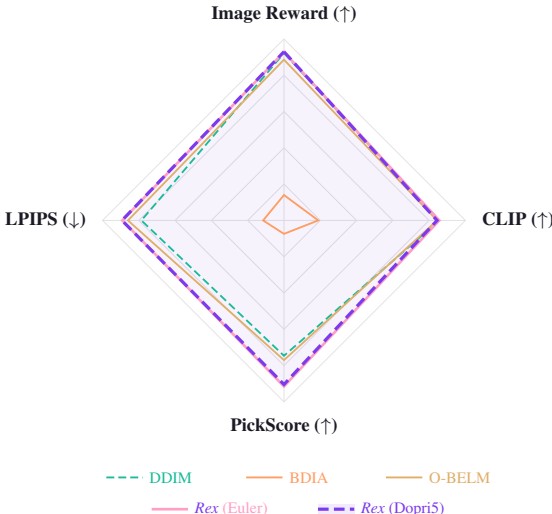

*Figure 9.* Radar chart comparing reversible solvers on round-trip image editing with Stable Diffusion v1.5 on the `pix2pix` dataset (50 inversion + 50 generation steps); LPIPS is inverted to $1 -$ LPIPS. Raw values in Table 7.

(Zhang et al., 2018) between $x_0$ and $x_0'$ to measure the perceptual preservation of non-edited content; further setup details are in Appendix G.3.

Figure 9 summarises the results, with raw numbers in Table 7. Both the fixed-step *Rex* (Euler) and adaptive-step *Rex* (Dopri5) match or exceed the non-reversible DDIM baseline on every metric. LPIPS, the most direct measure of edit faithfulness, drops from 0.214 (DDIM) and 0.140 (O-BELM) to 0.107 for *Rex* (Dopri5)—roughly a $2\times$ improvement over the strongest reversible baseline—while Image Reward and PickScore likewise top the table. BDIA fails catastrophically (LPIPS 0.885, Image Reward $-2.21$), consistent with its lack of a non-zero region of stability (see Appendix A.2 and Corollary A.3.2). We omit EDICT from Figure 9 entirely because it collapsed to the (approximate) identity map on this benchmark, producing edited images visually indistinguishable from the source. We highlight that *Rex* (Dopri5) is, to our knowledge, the first *adaptive step-size* reversible solver applied to diffusion editing.

### 4.4. Boltzmann Sampling

We evaluate the usefulness of *Rex* on equilibrium conformation sampling of tri-alanine. In particular, we are interested in drawing samples from a target Boltzmann distribution $p_{\text{target}}$ defined on $\mathbb{R}^d$ as

$$p_{\text{target}}(x) \propto \exp\left(-\mathcal{E}(x)\right), \tag{15}$$

where $\mathcal{E} : \mathbb{R}^d \to \mathbb{R}$ is the energy of the system which can be efficiently computed for any $x$. The Boltzmann distribution is notoriously difficult to sample with classical simulation-based techniques such as molecular dynamics;

instead, recent work has turned to deep generative models equipped with exact likelihoods, trained on a small biased dataset. This biased model can then be corrected using self-normalized importance sampling (Liu & Liu, 2001). Chen et al. (2018) showed that the exact likelihood of a neural ODE with learnt vector field $\boldsymbol{u}_t^\theta$ can be found as the solution to the following augmented ODE

$$\begin{bmatrix} \boldsymbol{x}_t \\ \log p_t^\theta(\boldsymbol{x}_t) \end{bmatrix} = \begin{bmatrix} \boldsymbol{x}_0 \\ \log p_0^\theta(\boldsymbol{x}_0) \end{bmatrix} + \int_0^t \begin{bmatrix} \boldsymbol{u}_s^\theta(\boldsymbol{x}_s) \\ -\left\langle \nabla_{\boldsymbol{x}}, \boldsymbol{u}_s^\theta(\boldsymbol{x}_s) \right\rangle \end{bmatrix} \mathrm{d}s. \tag{16}$$

In practice, however, one has to use a discretized numerical scheme $\boldsymbol{\Phi}$, yet its inverse $\boldsymbol{\Phi}^{-1}$ may not exist. This means the change-of-variables used to compute probabilities may no longer be valid, introducing errors; as discussed in Rehman et al. (2026a), this can pose a significant problem if not properly addressed. *Rex*, denoted $\boldsymbol{\Upsilon}$, is reversible by construction: $\boldsymbol{\Upsilon}^{-1}$ exists, so the change-of-variables yields proper probabilities up to discretization error.

**Baselines.** We compare against a broad variety of standard baselines which includes equivariant CNFs (Klein et al., 2023). In particular, we compare against an improved version, dubbed ECNF++, proposed by Tan et al. (2025a), which introduces a modified flow matching loss, a deeper architecture, an improved optimizer and learning rate schedule, and exponential moving average. We additionally compare to discrete normalizing flows in RegFlow (Rehman et al., 2026b) and the state-of-the-art *Sequential Boltzmann Generator* (SBG) (Tan et al., 2025a). Following Rehman et al. (2026a), we train our own *diffusion transformer* (DiT) (Peebles & Xie, 2023) on tri-alanine, which represents our final baseline. All the CNFs used the Dormand-Prince method (Dormand & Prince, 1980), a 5th-order Runge-Kutta scheme with an embedded 4th-order method for adaptive step sizing; the Butcher tableau is from Shampine (1986), and we use $\mathrm{atol} = \mathrm{rtol} = 10^{-5}$.

*Table 1.* Quantitative results on tri-alanine over $10^4$ samples. Best in **bold**, second best underlined.

| Model | Numerical scheme | ESS ($\uparrow$) | $\mathcal{E}$-$\mathcal{W}_2(\downarrow)$ | $\mathbb{T}$-$\mathcal{W}_2(\downarrow)$ |
|---|---|---|---|---|
| RegFlow | - | 0.029 | 1.051 | 1.612 |
| SBG (IS) | - | 0.052 | 0.758 | 0.502 |
| SBG (SMC) | - | - | 0.598 | 0.503 |
| ECNF++ | Dopri5 | 0.003 | 2.206 | 0.962 |
| DiT | Dopri5 | **0.140** | 0.737 | **0.468** |
| DiT | *Rex* (Dopri5) | 0.104 | **0.495** | 0.497 |

**Results.** In Table 1 we report the results of the sampling from the Boltzmann distribution in terms of the *effective sample size* (ESS) and the 2-Wasserstein distance between both the energy distributions ($\mathcal{E}$-$\mathcal{W}_2$) and dihedral angles ($\mathbb{T}$-$\mathcal{W}_2$)—further info on these metrics is provided in Appendix G.5.3. We see that applying the reversible *Rex* (Dopri5) to the DiT improves the $\mathcal{E}$-$\mathcal{W}_2$ metric to the best in the

table, with a modest drop in ESS and a small increase in $\mathbb{T}$-$\mathcal{W}_2$; all three remain competitive with the state-of-the-art. We note that for this experiment we choose $\zeta = 0.001$: we do not need to invert the solver exactly, only to guarantee that the scheme is invertible, leaving us free to optimize for stability (see Appendix A.2). A qualitative comparison of the resulting energy distributions is provided in Figure 11.

## 5. Related Work

As mentioned earlier in the preliminaries (see Section 2.1) there have been several works which have explored algebraically reversible schemes for (neural) differential equations, namely, the *asynchronous leapfrog method* (Mutze, 2013; Zhuang et al., 2021), *reversible Heun method* (Kidger et al., 2021), and *McCallum-Foster method* (McCallum & Foster, 2024). We discuss these methods in more detail in Appendix A.1. Contemporary work by Shmelev & Salvi (2025) explores *approximately* invertible stochastic Runge-Kutta schemes.

Within the literature on diffusion models the work on solvers for these models (see Song et al., 2021a; Lu et al., 2022b;a; Zhang et al., 2023) is relevant to our discussion on *Princeps* (see Theorem 3.5), in particular, the work by Gonzalez et al. (2024) which we discuss in more detail in Appendix A.4. Additionally, a few works have explored the exact inversion of diffusion models which we discuss in great detail in Appendix A.3.

## 6. Conclusion

We propose *Rex*, a family of reversible exponential (stochastic) Runge-Kutta solvers for diffusion models which can obtain an arbitrarily high order of convergence (for the ODE case). The construction also naturally admits reversible adaptive step-size solvers, enabling key AI4Science applications. Moreover, to the best of our knowledge, we propose the first method for exact inversion of diffusion SDEs without storing the entire trajectory of the Brownian motion. We also showed that *Princeps* subsumes several previous popular solvers, recovering reversible versions of these schemes. Our empirical studies show that *Rex* is both theoretically well-motivated and a capable, robust numerical scheme across a range of diffusion-model tasks. While we have presented *Rex* primarily in the context of diffusion models, the construction is substantially more general: it applies to any additive-noise SDE that can be written in the semi-linear form. This covers a broad class of generative models, including all the standard affine probability path flow matching formulations. *Rex* can be incorporated into existing applications in which preserving the bijectivity of flow maps is important.

## Acknowledgements

ZB thanks Sam McCallum for his feedback and insight on material related to the McCallum-Foster method, ShARK, and space-time Lévy area. ZB also wishes to acknowledge Alexander Tong and Danyal Rehman for their feedback on the Boltzmann generator experiments; and Danyal Rehman for generously providing a pre-trained DiT checkpoint for our Boltzmann sampling experiments.

## Impact Statement

We recognize that improving generative AI efficiency carries broader societal risks, including: (1) misuse for synthetic media generation, where more efficient inversion lowers the barrier to creating deepfakes or deceptive content; (2) amplification of training-data biases in higher-fidelity outputs; and (3) increased accessibility of generative tools, which may facilitate disinformation at scale.

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

# Appendices

## A. Detailed Discussion on Related Work

In this section we provide a detailed comparison with relevant related works. We begin by discussing algebraically reversible solvers in Appendix A.1. Then in Appendix A.2 we introduce the stability of an ODE solver, a helpful tool in comparing reversible solvers. Using this tool along with examining the convergence order we compare a variety of reversible solvers for diffusion models in Appendix A.3. Lastly, in Appendix A.4 we explore related work on constructing SDE solvers for diffusion models.

### A.1. Reversible Solvers

The earliest work on reversible solvers can be traced back to the pioneering work on symplectic integrators by Vogelaere (1956); Ruth (1983); Feng (1984). Due to symplectic integrators being developed for solving Hamiltonian systems they are intrinsically reversible by construction (Greydanus et al., 2019). More recently, Matsubara et al. (2021) explored the use of symplectic solvers for solving the continuous adjoint equations. Likewise, work by Pan et al. (2023) extended this idea, making use of symplectic solvers for solving the continous adjoint equations for diffusion models. However, in this section we will focus on non-symplectic reversible solvers.

Throughout this section we consider solving the following $d$-dimensional IVP:

$$\boldsymbol{x}(0) = \boldsymbol{x}_0, \qquad \frac{\mathrm{d}\boldsymbol{x}}{\mathrm{d}t}(t) = \boldsymbol{f}(t, \boldsymbol{x}(t)), \tag{17}$$

over the time interval $[0, T]$ with numerical solution $\{\boldsymbol{x}_n\}_{n=0}^{N}$.

### A.1.1. ASYNCHRONOUS LEAPFROG METHOD

To the best of our knowledge the *asynchronous leapfrog definition* was the first algebraically reversible non-symplectic solver, initially proposed by Mutze (2013) and popularized in a modern deep learning context by Zhuang et al. (2021). The asynchronous leapfrog method is a modification of the leapfrog method which converts it from a multi-step to single-step method. The method keeps track of a second state, $\{v_n\}$ which is supposed to be *sufficiently close* to the value of the vector field. We define the method below in Definition A.1.

**Definition A.1** (Asynchronous leapfrog method). Initialize $v_0 = f(0, x_0)$. Consider a step size of $h$ and let $\hat{t}_n = t_n + h/2$, then a forward step of the asynchronous leapfrog method is defined as

$$
\begin{aligned}
\hat{x}_n &= x_n + \frac{1}{2} v_n h, \\
v_{n+1} &= 2 f(\hat{t}_n, \hat{x}_n) - v_n, \\
x_{n+1} &= x_n + f(\hat{t}_n, \hat{x}_n) h,
\end{aligned}
\tag{18}
$$

and the backward step is given as

$$
\begin{aligned}
\hat{x}_n &= x_{n+1} - \frac{1}{2} v_{n+1} h, \\
x_n &= x_{n+1} - f(\hat{t}_n, \hat{x}_n) h, \\
v_n &= 2 f(\hat{t}_n, \hat{x}_n) - v_{n+1}.
\end{aligned}
\tag{19}
$$

**Remark A.2.** The method is a second-order solver (Zhuang et al., 2021, Theorem 3.1).

### A.1.2. REVERSIBLE HEUN METHOD

Later work by Kidger et al. (2021) proposed the *reversible Heun method*, a general purpose reversible solver which is symmetric and is an algebraically reversible SDE solver in addition to being a reversible ODE solver. This solver keeps track of an auxiliary state variable $\hat{x}_n$ and an extra copy of previous evaluations of the drift and diffusion coefficients. We present this method below in Definition A.3.

**Definition A.3** (Reversible Heun method for ODEs). Initialize $\hat{x}_0 = x_0$. Consider a step size of $h$, then a forward step of the reversible Heun method is defined as

$$
\begin{aligned}
\hat{x}_{n+1} &= 2 x_n - \hat{x}_n + f(t_n, \hat{x}_n) h, \\
x_{n+1} &= x_n + \frac{1}{2} \left( f(t_{n+1}, \hat{x}_{n+1}) + f(t_n, \hat{x}_n) \right) h.
\end{aligned}
\tag{20}
$$

and the backward step is given as

$$
\begin{aligned}
\hat{x}_n &= 2 x_{n+1} - \hat{x}_{n+1} - f(t_{n+1}, \hat{x}_{n+1}) h, \\
x_n &= x_{n+1} - \frac{1}{2} \left( f(t_{n+1}, \hat{x}_{n+1}) + f(t_n, \hat{x}_n) \right) h.
\end{aligned}
\tag{21}
$$

**Remark A.4.** This method is a second-order solver (Kidger, 2022, Theorem 5.18).

Recall that simulating SDEs in reverse-time is much trickier than simulating ODEs in reverse-time. This observation is even more true of algebraically reversible methods for SDEs. To the best of our knowledge, the only general reversible solver for SDEs is the reversible Heun method. The main idea of the SDE formulation of the reversible Heun method is to extend the Euler-Heun method[4] like how Heun's method was extended to the reversible Heun solver for ODEs. We define the method in Kidger et al. (2021, Algorithm 1) below in Definition A.5.

**Definition A.5** (Reversible Heun method for SDEs). Initialize $\hat{x}_0 = x_0$. Consider a step size of $h$ and let $W_h := W_{t_{n+1}} - W_{t_n}$, then a forward step of the reversible Heun method is defined as

$$
\begin{aligned}
\hat{x}_{n+1} &= 2 x_n - \hat{x}_n + \mu(t_n, \hat{x}_n) h + \sigma(t_n, \hat{x}_n) W_h, \\
x_{n+1} &= x_n + \frac{1}{2} \left( \mu(t_{n+1}, \hat{x}_{n+1}) + \mu(t_n, \hat{x}_n) \right) h \\
&\quad + \frac{1}{2} \left( \sigma(t_{n+1}, \hat{x}_{n+1}) + \sigma(t_n, \hat{x}_n) \right) W_h.
\end{aligned}
\tag{22}
$$

---

[4]This converges with strong order $\frac{1}{2}$ in the Stratonovich sense (Rüemelin, 1982).

and the backward step is given as

$$\hat{\boldsymbol{x}}_n = 2\boldsymbol{x}_{n+1} - \hat{\boldsymbol{x}}_{n+1} - \boldsymbol{\mu}(t_{n+1}, \hat{\boldsymbol{x}}_{n+1})h - \boldsymbol{\sigma}(t_n, \hat{\boldsymbol{x}}_n)\boldsymbol{W}_h,$$

$$\boldsymbol{x}_n = \boldsymbol{x}_{n+1} - \frac{1}{2}\left(\boldsymbol{\mu}(t_{n+1}, \hat{\boldsymbol{x}}_{n+1}) + \boldsymbol{\mu}(t_n, \hat{\boldsymbol{x}}_n)\right)h \tag{23}$$

$$- \frac{1}{2}\left(\boldsymbol{\sigma}(t_{n+1}, \hat{\boldsymbol{x}}_{n+1}) + \boldsymbol{\sigma}(t_n, \hat{\boldsymbol{x}}_n)\right)\boldsymbol{W}_h.$$

**Remark A.6.** This method requires some tractable solution for recalculating the Brownian motion from a splittable PRNG.

### A.1.3. MCCALLUM-FOSTER METHOD

Recent work by McCallum & Foster (2024) created a general method for constructing $n$-th order solvers from preexisting explicit single-step solvers while also addressing the stability issues that earlier methods suffered from. As McCallum & Foster (2024) simply refer to their method as *reversible X* where *X* is the underlying single-step solver we opt to refer to their method as the *McCallum-Foster method*. We restate the definition below.

**Definition A.7** (McCallum-Foster method). Initialize $\hat{\boldsymbol{x}}_0 = \boldsymbol{x}_0$ and let $\zeta \in (0, 1]$. Consider a step size of $h$, then a forward step of the McCallum-Foster method is defined as

$$\boldsymbol{x}_{n+1} = \zeta\boldsymbol{x}_n + (1 - \zeta)\hat{\boldsymbol{x}}_n + \boldsymbol{\Phi}_h(t_n, \hat{\boldsymbol{x}}_n), \tag{24a}$$

$$\hat{\boldsymbol{x}}_{n+1} = \hat{\boldsymbol{x}}_n - \boldsymbol{\Phi}_{-h}(t_{n+1}, \boldsymbol{x}_{n+1}), \tag{24b}$$

and the backward step is given as

$$\hat{\boldsymbol{x}}_n = \hat{\boldsymbol{x}}_{n+1} + \boldsymbol{\Phi}_{-h}(t_{n+1}, \boldsymbol{x}_{n+1}), \tag{25a}$$

$$\boldsymbol{x}_n = \zeta^{-1}\boldsymbol{x}_{n+1} + (1 - \zeta^{-1})\hat{\boldsymbol{x}}_n - \zeta^{-1}\boldsymbol{\Phi}_h(t_n, \hat{\boldsymbol{x}}_n). \tag{25b}$$

**Remark A.8.** *N.B.*, the $\zeta$ and $\zeta^{-1}$ terms in the forward and backward steps determine the stability of the system.

Interestingly, McCallum & Foster (2024, Theorem 2.1) showed that this reversible method inherits the convergence order of single-step solver $\boldsymbol{\Phi}_h$ enabling the construction of an arbitrarily high-order reversible solver. We restate this result below in Theorem A.1.

---

**Theorem A.1** (Convergence order of the McCallum-Foster method). *Consider the ODE in Equation (17) over $[0, T]$ with fixed time horizon $T > 0$. Let $T = Nh$ where $N > 0$ is the number of discretization steps and $h > 0$ is the step size. Let $\boldsymbol{\Phi}$ be a $k$-th order ODE solver such that it satisfies the Lipschitz condition*

$$\|\boldsymbol{\Phi}_\eta(\cdot, \boldsymbol{a}) - \boldsymbol{\Phi}_\eta(\cdot, \boldsymbol{b})\| \leq L|\eta|\|\boldsymbol{a} - \boldsymbol{b}\|, \tag{26}$$

*for all $\boldsymbol{a}, \boldsymbol{b} \in \mathbb{R}^d$ and $\eta \in [-h_{max}, h_{max}]$ for some $h_{max} > 0$. Consider the reversible solution $\{\boldsymbol{x}_n, \hat{\boldsymbol{x}}_n\}_{n in \mathbb{N}}$ admitted by Equation (24). Then there exists constants $h_{max} > 0$, $C > 0$, such that, for $h \in (0, h_{max}]$,*

$$\|\boldsymbol{x}_n - \boldsymbol{x}(t_n)\| \leq Ch^k. \tag{27}$$

---

### A.1.4. EXPLICIT AND EFFECTIVELY SYMMETRIC SCHEMES

Contemporary work by Shmelev & Salvi (2025) explores an *explicit and effectively symmetric* (EES) Runge-Kutta schemes (Shmelev et al., 2025) for neural SDEs. The *key* difference is that these schemes are only *approximately* invertible rather than *exactly* invertible. The other large difference is they construct the scheme for *rough differential equations* (RDEs) driven by a $\alpha$-Hölder branched rough path, $\alpha \in (0, 1]$. More concretely, for some driving signal $\boldsymbol{X} : [0, T] \to \mathbb{R}^d$, *e.g.*, a semi-martingale, lifted to a *rough path* $\mathcal{X}$ (see Appendix B.4), we consider the rough differential equation

$$\mathrm{d}\boldsymbol{Y}_t = \boldsymbol{f}(\boldsymbol{Y}_t)\,\mathrm{d}\mathcal{X}_t, \tag{28}$$

where $\boldsymbol{f}$ is sufficiently smooth and bounded with bounded derivatives. Following Redmann & Riedel (2020) assume that there exists smooth paths $\{\boldsymbol{X}^h\}_{h>0}$ for step-sizes $h > 0$ whose natural lifts to branch rough paths $\{\mathcal{X}^h\}_{h>0}$ converge almost surely to $\mathcal{X}$ under the metric of $\alpha$-Hölder rough paths as $h \to 0$. Then we have the solution with drive $\mathcal{X}^h$ given by

$$\mathrm{d}\boldsymbol{Y}_t^h = \boldsymbol{f}(\boldsymbol{Y}_t^h)\,\mathrm{d}\mathcal{X}_t^h. \tag{29}$$

Shmelev & Salvi (2025) then uses the following Runge-Kutta scheme for the RDE given by

$$\boldsymbol{y}_{n+1}^h = \boldsymbol{y}_n^h + \sum_{m=1}^{d} \sum_{i=1}^{s} b_i \boldsymbol{f}_m(\boldsymbol{k}_i) \boldsymbol{X}_{t_n, t_{n+1}}^{(m)}, \tag{30a}$$

$$\boldsymbol{k}_i = \boldsymbol{y}_n^h + \sum_{m=1}^{d} \sum_{i=1}^{s} a_{ij} \boldsymbol{f}_m(\boldsymbol{k}_i) \boldsymbol{X}_{t_n, t_{n+1}}^{(m)}, \tag{30b}$$

where $\boldsymbol{X}_{t_n, t_{n+1}}$ denotes the increment of $\boldsymbol{X}^h$ over $[t_n, t_{n+1}]$. With an appropriate choice of coefficients $a_{ij}$ and $b_i$ the Runge-Kutta scheme for the RDE is *approximately* reversible (Shmelev & Salvi, 2025). Cleary, this scheme is quite different from the SRK schemes we study and construct *exactly* reversible schemes from (see Appendix B).

### A.2. A Note on Stability

Historically, the stability properties of reversible solvers has been one of their weakest attributes (Kidger, 2022), limiting their use in practical applications. We formally introduce the notation of stability following Kidger (2022, Definition C.39), which we rewrite below in Definition A.9.

**Definition A.9** (Region of stability). Fix some numerical differential equation solver and let $\{\boldsymbol{x}_n^{\lambda,h}\}_{n\in\mathbb{N}}$ be the solution admitted by the numerical scheme solving the linear (or Dahlquist) test equation

$$\boldsymbol{x}(0) = \boldsymbol{x}_0, \qquad \frac{\mathrm{d}\boldsymbol{x}}{\mathrm{d}t} = \lambda \boldsymbol{x}(t), \tag{31}$$

where $\lambda \in \mathbb{C}$, $h > 0$ is the step size, and $\boldsymbol{x}_0 \in \mathbb{R}^d$ is a non-zero initial condition. The region of stability is defined as

$$\{h\lambda \in \mathbb{C} : \{\boldsymbol{x}_n^{\lambda,h}\}_{n\in\mathbb{N}} \text{ is uniformly bounded over } t_n\}. \tag{32}$$

*I.e.*, there exists a constant $C$ depending on $\lambda$ and $h$ but independent of $t_n$ such that $\|\boldsymbol{x}_n^{\lambda,h}\| < C$.

With the linear test equation Equation (31) the ODE converges asymptotically when $\Re(\lambda) \leq 0$,[5] and thus we are interested in numerical schemes which are bounded when the underlying analytical solution converges. Ideally, a numerical scheme would converge for all $h\lambda$ with $\Re(\lambda) < 0$.[6] Thus, the larger the region of stability the larger the step size we can take, wherein the numerical scheme still converges.

**Remark A.10.** Regrettably, the reversible Heun, leapfrog, and asynchronous leapfrog methods have poor stability properties. Specifically, the region of stability for all the methods is the complex interval $[-i, i]$, see Kidger (2022, Theorem 5.20) for reversible Heun, Shampine (2009, Section 2) for leapfrog, and Zhuang et al. (2021, Appendix A.4) for asynchronous leapfrog.

In other words, all previous reversible solvers are nowhere linearly stable for any step size $h$.[7] The instability in both asynchronous leapfrog and reversible Heun can be attributed to a step of general form $2A - B$, *i.e.*, we can write the source of instability as

$$2\boldsymbol{f}(\hat{t}_n, \hat{\boldsymbol{x}}_n) - \boldsymbol{v}_n, \qquad \text{(asynchronous leapfrog)}$$
$$2\boldsymbol{x}_{n+1} - \hat{\boldsymbol{x}}_{n+1}. \qquad \text{(reversible Heun)}$$

Thus the instability in these reversible schemes is caused by a decoupling between $\boldsymbol{v}_n$ and $\boldsymbol{f}(t_n, \boldsymbol{x}_n)$ (asynchronous leapfrog); and $\boldsymbol{x}_n$ and $\hat{\boldsymbol{x}}_n$ (reversible Heun). The strategy of McCallum & Foster (2024) is to couple $\boldsymbol{x}_n$ and $\hat{\boldsymbol{x}}_n$ together with the coupling parameter $\zeta$. Using this strategy, they showed that it was possible to construct a reversible solver with a non-trivial region of convergence. Let $\boldsymbol{\Phi}_h(t_n, \boldsymbol{x}_n) = R(h\lambda)\boldsymbol{x}_n$ and let $R(h\lambda)$ denote the *transfer function* used in analysis of Runge-Kutta methods with step size $h$ (see Stewart, 2022). We restate McCallum & Foster (2024, Theorem 2.3) below.

---

[5]The ODE converges to 0 when $\Re(\lambda) < 0$.

[6]A region of stability which satisfies is known as a region of absolute stability.

[7]Linearly stability refers to stability for linear test equations with $\Re(\lambda) < 0$.

**Theorem A.2** (Region of stability for the McCallum-Foster method)**.** *Let $\Phi$ be given by an explicit Runge-Kutta solver. Then the reversible numerical solution $\{\boldsymbol{x}_n, \hat{\boldsymbol{x}}_n\}_{n\in\mathbb{N}}$ given by Equation (24) is linearly stable iff*

$$|\Gamma| < 1 + \zeta, \tag{33}$$

*where*

$$\Gamma = 1 + \zeta - (1 - \zeta)R(-h\lambda) - R(-h\lambda)R(h\lambda). \tag{34}$$

**Remark A.11.** The McCallum-Foster method when constructed from explicit Runge-Kutta methods have a *non-trivial* region of stability. Note, however, that this region of stability is smaller than the original region of stability from the original Runga-Kutta method.

## A.3. Exact Inversion of Diffusion Models

Independent of the work on reversible solvers for neural ODEs several researchers have developed reversible methods for solving the probability flow ODE—often in the literature on diffusion models this is called the *exact inversion* of diffusion models. The discussion here covers prior *reversible* solvers; the (non-reversible) explicit (S)RK solvers that *Princeps* generalizes—DDIM, the DPM-Solver family, SEEDS-1, and gDDIM—are catalogued separately in Appendix E.

### A.3.1. EDICT SAMPLER

The first work to explore this topic of exact inversion with diffusion models was that of Wallace et al. (2023), who inspired by coupling layers in normalizing flows (Dinh et al., 2015) proposed a reversible solver which they refer to as *exact diffusion inversion via coupled transformations* (EDICT). Like all reversible solvers this method keeps track of an extra state, denoted by $\{\boldsymbol{y}_n\}_{n\in\mathbb{N}}$, with $\boldsymbol{y}_0 = \boldsymbol{x}_0$. Letting $a_n = \frac{\alpha_{n+1}}{\alpha_n}$ and $b_n = \sigma_{n+1} - \frac{\alpha_{n+1}}{\alpha_n}\sigma_n$, this numerical scheme can be described as

$$
\begin{aligned}
\boldsymbol{x}_n^{\text{inter}} &= a_n\boldsymbol{x}_n + b_n\boldsymbol{x}_{T|t_n}^{\theta}(\boldsymbol{y}_n), \\
\boldsymbol{y}_n^{\text{inter}} &= a_n\boldsymbol{y}_n + b_n\boldsymbol{x}_{T|t_n}^{\theta}(\boldsymbol{x}_n^{\text{inter}}), \\
\boldsymbol{x}_{n+1} &= \xi\boldsymbol{x}_n^{\text{inter}} + (1-\xi)\boldsymbol{y}_n^{\text{inter}} \\
\boldsymbol{y}_{n+1} &= \xi\boldsymbol{x}_n^{\text{inter}} + (1-\xi)\boldsymbol{x}_{n+1},
\end{aligned}
\tag{35}
$$

where $\xi \in (0,1)$ is a mixing parameter.[8] This method can be inverted to obtain a closed form expression for backward step:

$$
\begin{aligned}
\boldsymbol{y}_n^{\text{inter}} &= \frac{\boldsymbol{y}_{n+1} - (1-\xi)\boldsymbol{x}_{n+1}}{\xi}, \\
\boldsymbol{x}_n^{\text{inter}} &= \frac{\boldsymbol{y}_{n+1} - (1-\xi)\boldsymbol{y}_n^{\text{inter}}}{\xi}, \\
\boldsymbol{y}_n &= \frac{\boldsymbol{y}_n^{\text{inter}} - b_n\boldsymbol{x}_{T|t_n}^{\theta}(\boldsymbol{x}_n^{\text{inter}})}{a_n}, \\
\boldsymbol{x}_n &= \frac{\boldsymbol{x}_n^{\text{inter}} - b_n\boldsymbol{x}_{T|t_n}^{\theta}(\boldsymbol{y}_n)}{a_n}.
\end{aligned}
\tag{36}
$$

Notably, the EDICT solver was developed in the context of discrete-time diffusion models and the connection to reversible solvers for ODEs was not considered in the original work. *N.B.*, to the best of our knowledge our work is the first to draw the connection between the work on reversible ODE solvers and exact inversion with diffusion models. Unfortunately, this method suffers from poor convergence issues (see Remark A.12) and generally has poor performance when used to perform sampling with diffusion models, thereby limiting its utility in practice (Zhang et al., 2024; Wang et al., 2024).

**Remark A.12.** Later work by Wang et al. (2024, Proposition 6) showed that EDICT is actually a zero-order method, *i.e.*, the local truncation error is $\mathcal{O}(h)$, making it generally unsuitable in practice.

---

[8]In practice, when used for image editing the authors found that the parameter $\xi$ controlled how closely the EDICT sampler aligned with the original sample, with lower values corresponding to higher agreement with the original sample.

### A.3.2. BDIA SAMPLER

Later work by Zhang et al. (2024) proposed a reversible solver for the probability flow ODE which they call *bidirectional integration approximation* (BDIA). The core idea is to use both single-step methods $\mathbf{\Phi}_{t_n, t_{n-1}}$ and $\mathbf{\Phi}_{t_n, t_{n+1}}$ to induce reversibility.[9] Then using these two approximations—both of which are computed from a discretization centered around $\boldsymbol{x}_n$—the process is update via a multistep process with a forward step of[10]

$$\boldsymbol{x}_{n+1} = \boldsymbol{x}_{n-1} - \mathbf{\Phi}_{t_n, t_{n-1}}(\boldsymbol{x}_n) + \mathbf{\Phi}_{t_n, t_{n+1}}(\boldsymbol{x}_n). \tag{37}$$

The backwards step can easily be expressed as

$$\boldsymbol{x}_{n-1} = \boldsymbol{x}_{n+1} + \mathbf{\Phi}_{t_n, t_{n-1}}(\boldsymbol{x}_n) + \mathbf{\Phi}_{t_n, t_{n+1}}(\boldsymbol{x}_n). \tag{38}$$

In practice, BDIA uses the DDIM solver (*i.e.*, Euler) for $\mathbf{\Phi}$, but in theory one could use a higher-order method—this was not explored in Zhang et al. (2024).

> **Proposition A.3** (BDIA is the leapfrog/midpoint method). *The BDIA method described in Equation* (37) *is the leapfrog/midpoint method when* $\mathbf{\Phi}_h(t, \boldsymbol{x}) = h\boldsymbol{u}_t^\theta(\boldsymbol{x})$, i.e., *the Euler step.*

*Proof.* This can be shown rather straightforwardly by substitution, *i.e.*,

$$\boldsymbol{x}_{n+1} = \boldsymbol{x}_{n-1} + 2h\boldsymbol{u}_{t_n}^\theta(\boldsymbol{x}_n). \tag{39}$$

$\square$

**Corollary A.3.1** (BDIA is a first-order method). *BDIA is first-order method,* i.e., *the local truncation error is* $\mathcal{O}(h^2)$.

**Remark A.13.** This result was also observed in Wang et al. (2024, Proposition 6).

**Corollary A.3.2** (BDIA is nowhere linearly stable). *BDIA is nowhere linearly stable,* i.e., *the region of stability is the complex interval* $[-i, i]$.

*Proof.* This follows straightforwardly from Proposition A.3 and Shampine (2009, Section 2). $\square$

Zhang et al. (2024) introduce a hyperparameter $\gamma \in [0, 1]$ which is used below

$$\hat{\mathbf{\Phi}}_{t_n, t_{n-1}}(\boldsymbol{x}_n) = (1 - \gamma)(\boldsymbol{x}_{n-1} - \boldsymbol{x}_n) + \gamma\mathbf{\Phi}_{t_n, t_{n-1}}(\boldsymbol{x}_n), \tag{40}$$

to modify the BDIA update rule in Equation (37). Thus, $\gamma$ can be viewed as a parameter which interpolates between the midpoint and Euler schemes. For image editing applications the authors found this parameter to control how closely the BDIA sampler aligned with the original image, with lower values corresponding to higher agreement with the original image (making it similar to the $\xi$ parameter from BDIA).

### A.3.3. BELM SAMPLER

Recently, Wang et al. (2024) proposed a linear multi-step reversible solver for the probability flow ODE called the *bidirectional explicit linear multi-step* (BELM) sampler. First, they reparameterize the probability flow ODE as

$$\mathrm{d}\overline{\boldsymbol{x}}(t) = \overline{\boldsymbol{x}}_{T|\overline{\sigma}_t}^\theta(\overline{\boldsymbol{x}}(t)) \, \mathrm{d}\overline{\sigma}_t, \tag{41}$$

where $\overline{\boldsymbol{x}}(t) := \boldsymbol{x}(t)/\alpha_t$, $\overline{\sigma}(t) := \sigma_t/\alpha_t$, and $\overline{\boldsymbol{x}}_{T|\overline{\sigma}_t}^\theta(\overline{\boldsymbol{x}}(t)) = \boldsymbol{x}_{T|t}^\theta(\boldsymbol{x}(t))$.[11] The BELM sampler makes use of the variable-stepsize-variable-formula (VSVF) linear multi-step methods (Crouzeix & Lisbona, 1984) to construct the numerical solver.

---

[9] *N.B.*, in the original paper, Zhang et al. (2024) use quite different notation for explaining their idea; however, we find our presentation to be simpler for the reader as it more easily enables comparison to other methods.

[10] In some sense, this is reminiscent of the idea from the more general McCallum-Foster method; however, this approach results in a multi-step method unlike the single-step method of McCallum & Foster (2024).

[11] *N.B.*, this is a popular parameterization of diffusion models and affine conditional flows. This can be done *mutatis mutandis* for target prediction models retrieving (Blasingame & Liu, 2025, Proposition D.2).

The $k$-step VSVF linear multi-step method for solving the reparameterized probability flow ODE in Equation (41) is given by

$$\overline{\boldsymbol{x}}_{n+1} = \sum_{m=1}^{k} a_{n,m}\overline{\boldsymbol{x}}_{n+1-m} \tag{42}$$

$$+ \sum_{m=1}^{k-1} b_{n,m}h_{n+1-m}\overline{\boldsymbol{x}}_{T|\overline{\sigma}_{n+1-m}}^{\theta}(\overline{\boldsymbol{x}}_{n+1-m}). \tag{43}$$

where $a_{n,m} \neq 0$,[12] and $b_{n,m}$ are coefficients chosen using dynamic multi-step formulæ to find the coefficients (Crouzeix & Lisbona, 1984); and $h_n$ are step sizes chosen beforehand. This scheme can be reversed via the backward step

$$\overline{\boldsymbol{x}}_{n+1-k} = \frac{1}{a_{n,k}}\overline{\boldsymbol{x}}_{n+1} - \sum_{m=1}^{k-1}\frac{a_{n,m}}{a_{n,k}}\overline{\boldsymbol{x}}_{n+1-m} \tag{44}$$

$$- \sum_{m=1}^{k-1}\frac{b_{n,m}}{a_{n,k}}h_{n+1-m}\overline{\boldsymbol{x}}_{T|\overline{\sigma}_{n+1-m}}^{\theta}(\overline{\boldsymbol{x}}_{n+1-m}). \tag{45}$$

**Remark A.14.** The BELM samplers require $k-1$ extra to be stored in memory in order to be reversible. In contrast, McCallum & Foster (2024) only requires storing one extra states, irregardless of the desired convergence order. Additionally, poor stability is a concern with such linear multi-step methods (see Kidger, 2022, Remark 5.24).

**Remark A.15.** Interestingly, the earlier EDICT and BDIA methods can be viewed as instances of the BELM method (Wang et al., 2024, Appendicies A.7 and A.8).

By solving the multi-step formulæ to minimize the local truncation error Wang et al. (2024) propose an instance of the BELM solver which they refer to as *O-BELM* defined as[13]

$$\overline{\boldsymbol{x}}_{n+1} = \frac{h_n^2}{h_{n-1}^2}\overline{\boldsymbol{x}}_{n-1} + \frac{h_{n-1}^2 + h_n^2}{h_{n-1}^2}\overline{\boldsymbol{x}}_n - \frac{h_n(h_n + h_{n+1})}{h_{n+1}}\overline{\boldsymbol{x}}_{0|\overline{\sigma}_n}(\overline{\boldsymbol{x}}_n). \tag{46}$$

Notably, the O-BELM sampler can also be viewed as instance of the leapfrog/midpoint method.

> **Theorem A.4** (O-BELM is the leapfrog/midpoint method). *Fix a step size $h_n = h$ for all $n$, then O-BELM is the leapfrog/midpoint method.*

*Proof.* This follows from substitution of $h_n = h$. $\qquad\square$

**Corollary A.4.1** (O-BELM is nowhere linearly stable). *Fix a step size $h_n = h$, then O-BELM is nowhere linearly stable, i.e., the region of stability is the complex interval $[-i, i]$.*

### A.3.4. CYCLEDIFFUSION

To our knowledge, the *only* other work to propose exact inversion with the SDE formulation of the diffusion models is the work of Wu & la Torre (2023). However, there a *several* noticeable distinctions, the largest being that they store the entire solution trajectory in memory. Given a particular realization of the Wiener process that admits $\boldsymbol{x}_t \sim \mathcal{N}(\alpha_t\boldsymbol{x}_0 \mid \sigma_t^2\mathbf{I})$, then given $\boldsymbol{x}_s$ and noise $\boldsymbol{\epsilon}_s \sim \mathcal{N}(\mathbf{0}, \mathbf{I})$ we can calculate

$$\boldsymbol{x}_t = \frac{\alpha_t}{\alpha_s}\boldsymbol{x}_s + 2\sigma_t(e^h - 1)\hat{\boldsymbol{x}}_{T|s}(\boldsymbol{x}_s) + \sigma_t\sqrt{e^{2h} - 1}\boldsymbol{\epsilon}_s. \tag{47}$$

Wu & la Torre (2023) propose to invert this by first calculating, for two samples $\boldsymbol{x}_t$ and $\boldsymbol{x}_s$, the noise $\boldsymbol{\epsilon}_s$. This can be calculated by rearranging the previous equation to find

$$\boldsymbol{\epsilon}_s = \frac{\boldsymbol{x}_t - \frac{\alpha_t}{\alpha_s}\boldsymbol{x}_s + 2\sigma_t(e^h - 1)\boldsymbol{\epsilon}_\theta(\boldsymbol{x}_s, \boldsymbol{z}, s)}{\sigma_t\sqrt{e^{2h} - 1}} \tag{48}$$

---

[12]This is to ensure that the method is reversible.

[13]*N.B.*, the original equation in Wang et al. (2024, Equation (18)) had a sign difference for the coefficient of $b_{i,1}$; however, this is due to differences in convention in handling integration in reverse-time.

*Table 2.* Comparison of different (non-symplectic) reversible solvers. We note that some of the solvers were developed particularly for the probability flow ODE (an affine conditional flow) whilst others work for general ODEs/SDEs. In the first column we denote the number of extra states the numerical scheme needs to keep in memory to ensure algebraic reversibility. For BELM $k$ denotes the number of steps and for McCallum-Foster $k$ denotes the convergence order of the underlying single-step solver. For the column labelled *region of linear stability* we mean there exists some subset of $\mathbb{C}$ which is the region of stability and the set is not a null set. The proof of convergence for BELM is only provided for the special case (called *O-BELM* in Wang et al. (2024)) with $k = 2$.

| Solver | SDE | Exponential integrators | Number of extra states | Local truncation error | Region of linear stability | Proof of convergence |
|---|---|---|---|---|---|---|
| Asynchronous leapfrog | ✗ | ✗ | 1 | $\mathcal{O}(h^3)$ | ✗ | ✓ |
| Reversible Heun | ✓ | ✗ | 1 | $\mathcal{O}(h^3)$ | ✗ | ✓ |
| McCallum-Foster | ✗ | ✗ | 1 | $\mathcal{O}(h^{k+1})$ | ✓ | ✓ |
| EDICT | ✗ | ✗ | 1 | $\mathcal{O}(h)$ | ✗ | ✗ |
| BDIA | ✗ | ✗ | 1 | $\mathcal{O}(h^2)$ | ✗ | ✗ |
| BELM | ✗ | ✓ | $k-1$ | $\mathcal{O}(h^{k+1})$ | ✗ | ∼ |
| *Rex* | ✓ | ✓ | 1 | $\mathcal{O}(h^{k+1})$ | ✓ | ✓ |

With this the sequence $\{\boldsymbol{\epsilon}_{t_i}\}_{i=1}^N$ of added noises can be calculated which can be used to reconstruct the original input from the initial realization of the Wiener process. However, unlike our approach, this process requires storing the entire realization in memory.

### A.3.5. SUMMARY

We present a summary of related works on either *exact inversion* or *reversible solvers* below in Table 2. *N.B.*, we omit *CycleDiffusion* because it is more orthogonal to the general concept of a reversible solver and is only reversible in the trivial sense.

### A.4. SDE Solvers for Diffusion Models

Next we discuss related works on SDE solvers for the reverse-time diffusion SDE in Equation (5). Now there are numerous *stochastic Runge-Kutta* (SRK) methods in the literature all tailor to specific types of SDEs, which we can distinguish by the their strong order of convergence (see Definition D.1) and strong order conditions. For example the classic Euler-Maruyama scheme (Kloeden & Platen, 1992) has strong order of convergence of 0.5 and was straightforwardlly applied to the reverse-time diffusion SDE in Jolicoeur-Martineau et al. (2021) as a baseline. Song et al. (2021b) proposed an ancestral sampling scheme for a discretization of the forward-time diffusion SDE in Equation (3) with additional Langevin dynamics; likewise, the DDIM solver from Song et al. (2021a) can be viewed a sort of Euler-Maruyama scheme. Other classic SDE schemes like SRA1/SRA2/SRA3 schemes (Rößler, 2010) all have strong order of convergence 1.5 for additive noise SDEs and were tested for diffusion models in Jolicoeur-Martineau et al. (2021).

More recently, researchers have explored exponential solvers for SDEs, *e.g.*, the exponential Euler-Maruyama method (Komori et al., 2017) and the *stochastic Runge-Kutta Lawson* (SRKL) schemes (Debrabant et al., 2021). From an initial inspection the SRKL schemes of Debrabant et al. (2021, Algorithm 1) is somewhat similar to our method for constructing $\Psi$; however, upon closer inspection they are some key fundamental differences.[14] The largest of these is how the underlying SRK schemes are represented. In particular the SRKL schemes choose to follow the conventions of Burrage & Burrage (2000) (for Stratonovich SDEs) in constructing the underlying SRK schemes; whereas we follow the SRK schemes outlined by Foster et al. (2024) (see Appendix B). These differences stem from how one chooses to handle the the iterated stochastic integrals from the Stratonovich-Taylor (or Itô-Taylor) expansions.

### A.4.1. COMPARISON WITH SEEDS

Mostly directly relevant to our work on constructing a stochastic $\Psi$ is the SEEDS family of solvers proposed by Gonzalez et al. (2024). Similar to us, they also approach using exponential methods to simplify the expression of diffusion models Gonzalez et al. (2024, Appendix B.1). There are two *key* distinctions, namely, 1) that they use the *stochastic exponential*

---

[14]*N.B.*, in general Debrabant et al. (2021) consider full stochastic Lawson schemes where the integrating factor is a stochastic process given by the matrix exponential applied to linear terms in the drift and diffusion coefficients; conversely, the drift stochastic Lawson schemes are more similar to what we study.

*time differencing* (SETD) method (Adamu, 2011), whereas, we construct stochastic Lawson schemes;[15] and 2) that they use a different technique for modeling the iterated stochastic integrals for high-order solvers. In particular, SEEDS introduces a decomposition for the iterated stochastic integrals produced by the Itô-Taylor expansions of Equation (5) such that the decomposition preserves the Markov property, *i.e.*, the random variables used to construct model the Brownian increments from iterated integrals are independent on non-overlapping intervals and dependent on overlapping intervals (see Gonzalez et al., 2024, Proposition 4.3). By making use of the SRK schemes of Foster et al. (2024) developed from using the space-time Lévy area to construct high-order splitting methods we have an alternative method for ensuring this property. This results in our solver based on ShARK (see Appendix B.3 and Theorem 3.4) having a strong order of convergence of 1.5; whereas, SEEDS-3 only achieves a *weak* order of convergence of 1.

This brings us to another large difference, the SEEDS solvers focus on the *weak* approximation to Equation (5); whereas, as we are concerned with the *strong* approximation to Equation (5). The difference between these two is that the weak convergence is considered with the precisions of the *moments*; whereas, strong convergence is concerned with the precision of the *path*. Moreover, by definition a strong order of convergence implies a weak order of convergence, the converse is not true. In particular, for our application of developing *reversible* schemes this strong order of convergence is particularly important as we care about the path. Thus the technique SEEDS uses to replace iterated Itô integrals with other random variables with equivalent moment conditions is *wholly unsuitable* for our purposes as we desire a *strong* approximation.

## B. Stochastic Runge-Kutta Methods

Recall that the general Butcher tableau for a $s$-stage explicit RK scheme (Stewart, 2022, Section 6.1.4) for a generic ODE is written as

$$
\begin{array}{c|ccccc}
c_1 & & & & & \\
c_2 & a_{21} & & & & \\
c_3 & a_{31} & a_{32} & & & \\
\vdots & \vdots & \vdots & \ddots & & \\
c_s & a_{s1} & a_{s2} & \cdots & a_{s(s-1)} & \\
\hline
& b_1 & b_2 & \cdots & b_{s-1} & b_s
\end{array}
= \begin{array}{c|c} c & a \\ \hline & b \end{array}. \tag{49}
$$

*E.g.*, the famous 4-th order Runge-Kutta (RK4) method is given by

$$
\begin{array}{c|cccc}
0 & & & & \\
\frac{1}{2} & \frac{1}{2} & & & \\
\frac{1}{2} & 0 & \frac{1}{2} & & \\
1 & 0 & 0 & 1 & \\
\hline
& \frac{1}{6} & \frac{1}{3} & \frac{1}{3} & \frac{1}{6}
\end{array}. \tag{50}
$$

However, for SDEs this is much trickier due to the presense of iterated stochastic integrals in the Itô-Taylor or Stratonovich-Taylor expansions (Kloeden & Platen, 1992). Consider a $d$-dimensional *Stratonovich* SDE driven by $d_w$-dimensional Brownian motion $\{\boldsymbol{W}_t\}_{t \in [0,T]}$ defined as

$$
\mathrm{d}\boldsymbol{X}_t = \boldsymbol{\mu}_\theta(t, \boldsymbol{X}_t)\, \mathrm{d}t + \boldsymbol{\sigma}_\theta(t, \boldsymbol{X}_t) \circ \mathrm{d}\boldsymbol{W}_t, \tag{51}
$$

where $\boldsymbol{\mu}_\theta \in \mathcal{C}^2(\mathbb{R} \times \mathbb{R}^d; \mathbb{R}^d)$ and $\boldsymbol{\sigma}_\theta \in \mathcal{C}^3(\mathbb{R} \times \mathbb{R}^d; \mathbb{R}^{d \times d_w})$ satisfy the usual regularity conditions for Stratonovich SDEs (Øksendal, 2003, Theorem 5.2.1) and where $\circ \mathrm{d}\boldsymbol{W}_t$ denotes integration in the Stratonovich sense.

Rößler (2025) write one such class of an $s$-stage explicit SRK methods (*cf.* Burrage & Burrage, 2000; Rößler, 2010) for

---

[15]*N.B.*, for certain scenarios these two different viewpoints converge, particularly, in the deterministic case. See our discussion on the family of DPM-Solvers which also use (S)ETD in Appendix E.

Equation (51) as

$$
\boldsymbol{Z}_i^{(0)} = \boldsymbol{X}_n + h \sum_{j=1}^{i-1} a_{ij}^{(0)} \boldsymbol{\mu}_\theta(t_n + c_j^{(0)}, \boldsymbol{Z}_j^{(0)}),
$$

$$
\boldsymbol{Z}_i^{(k)} = \boldsymbol{X}_n + h \sum_{j=1}^{i-1} a_{ij}^{(1)} \boldsymbol{\mu}_\theta(t_n + c_j^{(0)}, \boldsymbol{Z}_j^{(0)}) + \sum_{j=1}^{i-1} \sum_{l=1}^{d_w} a_{ij}^{(2)} \boldsymbol{I}_{(l,k),n} \boldsymbol{\sigma}_\theta(t_n + c_j^{(1)}, \boldsymbol{Z}_i^{(l)}), \tag{52}
$$

$$
\boldsymbol{X}_{n+1} = \boldsymbol{X}_n + h \sum_{i=1}^{s} b_i^{(0)} \boldsymbol{\mu}_\theta(t_n + c_i^{(0)}, \boldsymbol{Z}_j^{(0)}) + \sum_{i=1}^{s} \sum_{k=1}^{d_w} \left( b_i^{(1)} \boldsymbol{I}_{(k),n} + b_i^{(2)} \right) \boldsymbol{\sigma}_\theta(t_n + c_j^{(1)}, \boldsymbol{Z}_i^{(k)}),
$$

for $k = 1, \ldots, d_w$ and where

$$
\boldsymbol{I}_{(k),n} = \int_{t_n}^{t_{n+1}} \circ \mathrm{d} \boldsymbol{W}_u^k = \boldsymbol{W}_{t_{n+1}}^k - \boldsymbol{W}_{t_n}^k, \tag{53}
$$

$$
\boldsymbol{I}_{(l,k),n} = \int_{t_n}^{t_{n+1}} \int_{t_n}^{u} \circ \mathrm{d} \boldsymbol{W}_v^l \circ \mathrm{d} \boldsymbol{W}_u^k, \tag{54}
$$

let $\hat{\boldsymbol{I}}$ denote the iterated integrals for the Itô case *mutatis mutandis*. This scheme is described the by the *extended* Butcher tableau (Rößler, 2025)

$$
\begin{array}{c|c|c|c}
c^{(0)} & a^{(0)} & & \\
\hline
c^{(1)} & a^{(1)} & a^{(2)} & \\
\hline
& b^{(0)} & b^{(1)} & b^{(2)}
\end{array}. \tag{55}
$$

These iterated integrals $\boldsymbol{I}_{(l,k),n}$ are very tricky to work with and can raise up many practical concerns. As alluded to earlier (see Appendix A.4.1) it is common to use a weak approximation of such integrals via a random variables with corresponding moments. This results in two drawbacks: 1) the resulting SDE scheme only converges in the *weak* sense and 2) the solution yielding by the scheme is not a Markov chain in general. SEEDS overcomes the second issue by using a special decomposition to preserve the Markov property, see the ablations in Gonzalez et al. (2024) for more details on this topic in practice.

## B.1. Foster-Reis-Strange SRK Scheme

Conversely, Foster et al. (2024) propose another SRK scheme based on higher-order splitting methods for Stratonovich SDEs. For the Stratonovich SDE in Equation (51) Foster et al. (2024) write an $s$-stage SRK as

$$
\boldsymbol{\mu}_\theta^i = \boldsymbol{\mu}_\theta(t_n + c_i h, \boldsymbol{Z}_i),
$$

$$
\boldsymbol{\sigma}_\theta^i = \boldsymbol{\sigma}_\theta(t_n + c_i h, \boldsymbol{Z}_i),
$$

$$
\boldsymbol{Z}_i = \boldsymbol{X}_n + h \left( \sum_{j=1}^{i-1} a_{ij} \boldsymbol{\mu}_\theta^j \right) + \boldsymbol{W}_n \left( \sum_{j=1}^{i-1} a_{ij}^W \boldsymbol{\sigma}_\theta^j \right) + \boldsymbol{H}_n \left( \sum_{j=1}^{i-1} a_{ij}^H \boldsymbol{\sigma}_\theta^j \right), \tag{56}
$$

$$
\boldsymbol{X}_{n+1} = \boldsymbol{X}_n + h \left( \sum_{i=1}^{s} b_i \boldsymbol{\mu}_\theta^i \right) + \boldsymbol{W}_n \left( \sum_{i=1}^{s} b_i^W \boldsymbol{\sigma}_\theta^i \right) + \boldsymbol{H}_n \left( \sum_{i=1}^{s} b_i^H \boldsymbol{\sigma}_\theta^i \right),
$$

where $h = t_{n+1} - t_n$ is the step size and $\boldsymbol{W}_n \coloneqq \boldsymbol{W}_{t_n, t_{n+1}}$ and $\boldsymbol{H}_n \coloneqq \boldsymbol{H}_{t_n, t_{n+1}}$ are the Brownian and space-time Lévy increments (see Definition 3.2) respectively; and where $a_{ij}, a_{ij}^W, a_{ij}^H \in \mathbb{R}^{s \times s}, b_i, b_i^W, b_i^H \in \mathbb{R}^s$, and $c_i \in \mathbb{R}^s$ for the coefficients for an *extended* Butcher tableau (Foster et al., 2024) which is given as

$$
\begin{array}{c|c|c|c}
c & a & a^W & a^H \\
\hline
& b & b^W & b^H
\end{array}. \tag{57}
$$

*E.g.*, we can write the famous Euler-Maruyama scheme as

$$
\begin{array}{c|c|c|c}
0 & 0 & 0 & 0 \\
\hline
& 1 & 1 & 0
\end{array}. \tag{58}
$$

## B.2. Independence of the Brownian and Lévy Increments

Remarkably, Foster et al. (2020, Theorem 2.2) present a polynomial Karhunen-Loève theorem for the Brownian bridge (see Definition F.1)—picture a stochastic analogue to the Fourier series of a function on a bounded interval—which leads to a most useful remark (Foster et al., 2020, Remark 3.6) which we restate below.

**Remark B.1.** We have that $H_{s,t} \sim \mathcal{N}(0, \frac{1}{12}h)$ is independent of $W_{s,t}$ when $d = 1$, likewise, since the coordinate processes of a Brownian motion are independent, one can write $\boldsymbol{W}_{s,t} \sim \mathcal{N}(\boldsymbol{0}, h\boldsymbol{I})$ and $\boldsymbol{H}_{s,t} \sim \mathcal{N}(\boldsymbol{0}, \frac{1}{12}h\boldsymbol{I})$ are independent.

Thus we have found another remedy to the problem of independent increments, whilst still being able to obtain a *strong* approximation of the SDE.

## B.3. ShARK

Recently, Foster et al. (2024) developed *shifted additive-noise Runge-Kutta* (ShARK) for additive noise SDEs which is based on Foster et al. (2024, Equation (6.1)). This scheme converges strongly with order 1.5 for additive-noise SDEs and makes two evaluations of the drift and diffusion per step.

ShARK is described via the following extended Butcher tableau

$$
\begin{array}{cc|cc|c|c}
0 & & & & 0 & 1 \\
\frac{5}{6} & \frac{5}{6} & & & \frac{5}{6} & 1 \\
\hline
& 0.4 & 0.6 & & 1 & 0 \\
& -0.6 & 0.6 & & &
\end{array}
\tag{59}
$$

The second row for the $b$ variable describes the coefficients used for adaptive-step size solvers to approximate the error at each step. The Butcher tableau for this scheme can be found here: `https://github.com/patrick-kidger/diffrax/blob/main/diffrax/_solver/shark.py`.

## B.4. A Brief Note on the Theory of Rough Paths

To perform reversibility it is useful to consider the pathwise interpretation of SDEs (Lyons, 1998), as such we introduce a few notations from rough path theory. Let $\{\boldsymbol{W}_t\}$ be a $d_w$-dimensional Brownian motion and let $\boldsymbol{W}$ be enhanced by

$$
\mathbb{W}_{s,t} = \int_s^t \boldsymbol{W}_{s,r} \otimes \circ \mathrm{d}\boldsymbol{W}_r,
\tag{60}
$$

where $\otimes$ is the tensor product. Then, the pair $\mathcal{W} \coloneqq (\boldsymbol{W}, \mathbb{W})$ is the *Stratonovich enhanced Brownian rough path*.[16] Thus consider the $d_x$-dimensional *rough differential equation* RDE of the form:

$$
\mathrm{d}\boldsymbol{X}_t = \boldsymbol{\mu}(t, \boldsymbol{X}_t)\,\mathrm{d}t + \boldsymbol{\sigma}(t, \boldsymbol{X}_t)\,\mathrm{d}\mathcal{W}_t, \qquad \boldsymbol{X}_0 = \boldsymbol{x}_0.
\tag{61}
$$

where $\boldsymbol{\mu} : [0, T] \times \mathbb{R}^{d_x} \to \mathbb{R}^{d_x}$ is Lipschitz continuous in its second argument and $\boldsymbol{\sigma} \in \mathcal{C}_b^{1,3}([0, T] \times \mathbb{R}^{d_x}; \mathcal{L}(\mathbb{R}^{d_w}, \mathbb{R}^{d_x}))$ (Friz & Hairer, 2020, Theorem 9.1).[17] Fix an $\omega \in \Omega$, then almost surely $\mathcal{W}(\omega)$ admits a unique solution to the RDE $(\boldsymbol{X}_t(\omega), \boldsymbol{\sigma}(t, \boldsymbol{X}_t(\omega)))$ and $\boldsymbol{X}_t = \boldsymbol{X}_t(\omega)$ is a strong solution to the Stratonovich SDE[18] started at $\boldsymbol{X}_0 = \boldsymbol{x}_0$. To elucidate, consider the commutative diagram below

$$
\boldsymbol{W} \xmapsto{\Psi} (\boldsymbol{W}, \mathbb{W}) \xmapsto{S} \boldsymbol{X},
\tag{62}
$$

where $\Psi$ is a map which merely lifts Brownian motion into a rough path (could be Itô or Stratonovich), the second map, $S$, is known as the *Itô-Lyons map* (Lyons, 1998); this map is purely deterministic and is also a *continuous map* w.r.t. to initial

---

[16] See, Friz & Hairer (2020, Chapter 3) for more details.

[17] Here $\mathcal{L}(V, W)$ denotes the set of continuous maps from $V$ to $W$, a Banach space.

[18] If $\boldsymbol{X}_t$ and $\partial_{\boldsymbol{x}} \boldsymbol{X}_t$ are adapted and $\langle \boldsymbol{X}, \boldsymbol{W} \rangle_t$ exists, then almost surely

$$
\int_0^T \boldsymbol{X}\,\mathrm{d}\mathcal{W}_t = \int_0^T \boldsymbol{X} \circ \mathrm{d}\boldsymbol{W}_t.
$$

condition and driving signal. Thus for a fixed realization of the Brownian motion we have a pathwise interpretation of the Stratonovich SDE.

## B.5. Time-Changed Brownian Motion

In this subsection we develop the time-changed Brownian motion machinery used throughout the SDE derivations in Appendix C.2. We review some necessary preliminary results about continuous local martingales and Brownian motion, and show that we can simplify the stochastic integrals in Equation (7) and the corresponding reparameterization with data/noise prediction models.

**Dambis-Dubins-Schwarz Representation Theorem.** We restate the Dambis-Dubins-Schwarz representation theorem (Dubins & Schwarz, 1965) which shows that continuous local martingales can be represented as time-changed Brownian motions.

> **Theorem B.1** (Dambis-Dubins-Schwarz representation theorem). *Let $M$ be a continuous local martingale adapted to a filtration $\{\mathcal{F}_t\}_{t \geq 0}$ beginning at 0* (i.e., $M_0 = 0$) *such that $\langle M \rangle_\infty = \infty$ almost surely. Define the random variables $\{\tau_t\}_{t \geq 0}$ by*
>
> $$\tau_t = \inf \{s \geq 0 : \langle M \rangle_s > t\} = \sup \{s \geq 0 : \langle M \rangle_s = t\}. \tag{63}$$
>
> *Then for any given $t$ the random variable $\tau_t$ is an almost surely finite stopping time, and the process[a] $B_t = M_{\tau_t}$ is a Brownian motion w.r.t. the filtration $\{\mathcal{G}_t\}_{t \geq 0} = \{\mathcal{F}_{\tau_t}\}_{t \geq 0}$. Moreover,*
>
> $$M_t = B_{\langle M \rangle_t}. \tag{64}$$
>
> ___________
> [a]Defined up to a null set.

**A Multi-Dimensional Version of the Dambis-Dubins-Schwarz Representation Theorem.** In our work we are interested in a $d$-dimensional local martingale $\boldsymbol{M} \coloneqq (M^1, \ldots M^d)$. As such we discuss a multi-dimensional extension of Theorem B.1 which requires that the $d$-dimensional continuous local martingale if the quadratic (covariation) matrix $\langle \boldsymbol{M} \rangle_t^{ij} = \langle M^i, M^j \rangle_t$ is proportional to the identity matrix. We adapt the following theorem from Lowther (2010, Theorem 2) and Bourgade (2010, Theorem 4.13) (*cf*. Revuz & Yor, 2013).

> **Theorem B.2** (Multi-dimensional Dambis-Dubins-Schwarz representation theorem). *Let $\boldsymbol{M} = (M^1, \ldots, M^d)$ be a collection of continuous local martingales with $\boldsymbol{M}_0 = \boldsymbol{0}$ such that for any $1 \leq 1 \leq d, \langle \boldsymbol{M} \rangle_\infty^{ii} = \infty$ almost surely. Suppose, furthermore, that $\langle M^i, M^j \rangle_t = \delta_{ij} A_t$, where $\delta$ denotes the Kronecker delta, for some process $A$ and all $1 \leq i, j \leq d$ and $t \geq 0$. Then there is a $d$-dimensional Brownian motion $\boldsymbol{B}$ w.r.t. a filtration $\{\mathcal{G}_t\}_{t \geq 0}$ such that for each $t \geq 0$, $\omega \mapsto A_t(\omega)$ is a $\mathcal{G}$-stopping time and*
>
> $$\boldsymbol{M}_t = \boldsymbol{B}_{A_t}. \tag{65}$$

**Enlargement of the Probability Space.** Recall that in Theorems B.1 and B.2 we stated that quadratic variation of the continuous local martingale needed to tend towards infinity as $t \to \infty$. What when $\langle M \rangle_\infty$ has a nonzero probability of being finite? It can be shown that Theorems B.1 and B.2 holds under an enlargement of the probability space (not the filtration). Consider both our original probability space $(\Omega, \mathcal{F}, P)$ and another probability space $(\Omega', \mathcal{F}', P')$ along with a measurable surjection $f : \Omega' \to \Omega$ preserving probabilities such that $P(A) = P'(f^{-1}(A))$ for all $A \in \mathcal{F}$, *i.e.*, $f_* P'$ is a pushforward measure. Thus any process on the original probability space can be *lifted* to $(\Omega', \mathcal{F}', P')$ and likewise the filtration is also lifted to $\mathcal{F}'_t = \{f^{-1}(A) : A \in \mathcal{F}_t\}$. Therefore, it is possible to enlarge the probability space so that Brownian motion is defined. *E.g.*, if $(\Omega'', \mathcal{F}'', P'')$ is probability space on which there is a Brownian motion defined, we can take $\Omega' = \Omega \times \Omega''$, $\mathcal{F}' = \mathcal{F} \otimes \mathcal{F}''$, and $P' = P \otimes P''$ for the enlargement, and $f :' \Omega \to \Omega$ is just the projection onto $\Omega$.

**Data Prediction.** We now present a lemma for rewriting the stochastic differential in Equation (139) using the Dambis-Dubins-Schwarz representation theorem. Recall that in Equation (139) we denote the reverse-time $d$-dimensional Brownian motion as $\overline{\boldsymbol{W}}_t$, *i.e.*, by Lévy's characterization we have $\overline{\boldsymbol{W}}_T = \boldsymbol{0}$ and

$$\overline{\boldsymbol{W}}_t - \overline{\boldsymbol{W}}_s \sim -\mathcal{N}(\boldsymbol{0}, (t-s)\boldsymbol{I}) = \mathcal{N}(\boldsymbol{0}, (t-s)\boldsymbol{I}), \tag{66}$$

for $0 \leq t < s \leq T$. With this in mind we present Lemma B.3 below.

**Lemma B.3.** *The stochastic differential $\sqrt{-\frac{d\varrho_t}{dt}}\,d\overline{\boldsymbol{W}}_t$ can be rewritten as a time-changed Brownian motion of the form*

$$\sqrt{-\frac{d\varrho_t}{dt}}\,d\overline{\boldsymbol{W}}_t = d\boldsymbol{W}_\varrho, \tag{67}$$

*where $\varrho_t = \gamma_t^2$.*

*Proof.* To simplify the stochastic integral term we first define a continuous local martingale $\boldsymbol{M}_t$ via the stochastic integral

$$\boldsymbol{M}_t := \int_T^t \sqrt{-\frac{d\varrho}{dt}}\,d\overline{\boldsymbol{W}}_t. \tag{68}$$

We choose time $T$ as our starting point for the martingale rather than 0 and then integrate in *reverse-time*. However, due to the negative sign within the square root term it is more convenient to work with $\boldsymbol{W}_t$, *i.e.*, the standard $d$-dimensional Brownian motion defined in forward time. Recall that the standard $d$-dimensional Brownian motion in *reverse-time* with starting point $T$ is defined as

$$\overline{\boldsymbol{W}}_t = \boldsymbol{W}_T - \boldsymbol{W}_t \tag{69}$$

which is distributed like $\boldsymbol{W}_t$ in time $T - t$. Define the function $\boldsymbol{f}(t, \boldsymbol{W}_t) = \overline{\boldsymbol{W}}_t$. Then by Itô's lemma we have

$$d\boldsymbol{f}(t, \boldsymbol{W}_t) = \partial_t \boldsymbol{f}(t, \boldsymbol{W}_t)\,dt + \sum_{i=1}^d \partial_{\boldsymbol{x}_i} \boldsymbol{f}(t, \boldsymbol{W}_t)\,d\boldsymbol{W}_t^i + \frac{1}{2}\sum_{i,j=1}^d \partial_{\boldsymbol{x}_i, \boldsymbol{x}_j}\boldsymbol{f}(t, \boldsymbol{W}_t)\,d\left\langle \boldsymbol{W}^i, \boldsymbol{W}^j \right\rangle_t, \tag{70}$$

which simplifies to

$$d\boldsymbol{f}(t, \boldsymbol{W}_t) = d\overline{\boldsymbol{W}}_t = -d\boldsymbol{W}_t. \tag{71}$$

Thus we can rewrite Equation (68) as

$$\boldsymbol{M}_t = -\int_T^t \sqrt{-\frac{d\varrho}{dt}}\,d\boldsymbol{W}_t. \tag{72}$$

Next we establish a few properties of this martingale. First, $\boldsymbol{M}_T = \boldsymbol{0}$ by construction. Second, since the integral consists of scalar noise we have that $\langle M^i, M^j \rangle_t = 0$ for all $i \neq j$. Thus, the quadratic variation of $\langle \boldsymbol{M}_t \rangle^{ii}$ for each $i$ is found to be

$$\langle \boldsymbol{M} \rangle_t^{ii} = A_t = -\int_T^t \left(\sqrt{-\frac{d\varrho_\tau}{d\tau}}\right)^2 d\tau, \tag{73}$$

$$= \int_T^t \frac{d\varrho_\tau}{d\tau}\,d\tau, \tag{74}$$

$$= \varrho_t - \varrho_T = \frac{\alpha_t^2}{\sigma_t^2} - \frac{\alpha_T^2}{\sigma_T^2}. \tag{75}$$

Now we have a deterministic mapping from the original time to our new time via $A_t$. Now in general for any valid choice of $(\alpha_t, \sigma_t)$ we don't necessarily have that $\langle \boldsymbol{M} \rangle_\infty^{ii} = \infty$ almost surely and as such we may need to enlarge the underlying probability space. Our constructed martingale can be expressed as time-changed Brownian motion, see Theorem B.2, such that $\boldsymbol{M}_t = \boldsymbol{W}_{A_t}$ were $\boldsymbol{W}_\varrho$ is the standard $d$-dimensional Brownian motion with time variable $\varrho$.

Now we can rewrite Equation (72) in differential form as

$$d\boldsymbol{M}_t = d\boldsymbol{W}_{A_t}. \tag{76}$$

Because Brownian motion is time-shift invariant we can then write

$$d\boldsymbol{M}_t = d\boldsymbol{W}_{\varrho_t}. \tag{77}$$

$\square$

**Remark B.2.** Lemma B.3 can similarly be found via Øksendal (2003, Theorem 8.5.7) and Kobayashi (2011, Lemma 2.3); however, do to the oddness of the *reverse-time* integration we found it easier to tackle the problem via the Dambis-Dubins-Schwarz theorem.

**Remark B.3.** Under the common scenario where $\sigma_0 = 0$ then we have that $\langle \boldsymbol{M} \rangle_\infty^{ii} = \infty$ almost surely.

**Noise Prediction.** Next we discuss the stochastic integral used in the noise prediction formulation.

> **Lemma B.4.** *Let $\alpha_T > 0$. Then the stochastic differential $\sqrt{\frac{\mathrm{d}}{\mathrm{d}t}(\chi_t^2)}\,\mathrm{d}\overline{W}_t$ can be rewritten as a time-changed Brownian motion of the form*
>
> $$\sqrt{\frac{\mathrm{d}}{\mathrm{d}t}(\chi_t^2)}\,\mathrm{d}\overline{W}_t = \mathrm{d}\overline{W}_{\chi^2}, \tag{78}$$
>
> *where $\chi_t = \frac{\sigma_t}{\alpha_t}$.*

*Proof.* To simplify the stochastic integral term we first define a continuous local martingale $M_t$ via the stochastic integral

$$M_t := \int_T^t \sqrt{\frac{\mathrm{d}}{\mathrm{d}t}(\chi_t^2)}\,\mathrm{d}\overline{W}_t. \tag{79}$$

We choose time $T$ as our starting point for the martingale rather than 0 and then integrate in *reverse-time*, hence the negative sign. Next we establish a few properties of this martingale. First, $M_T = 0$ by construction. Second, since the integral consists of scalar noise we have that $\langle M^i, M^j \rangle_t = 0$ for all $i \neq j$. Thus, the quadratic variation of $\langle M_t \rangle^{ii}$ for each $i$ is found to be

$$\langle M \rangle_t^{ii} = A_t = \int_T^t \left(\sqrt{\frac{\mathrm{d}}{\mathrm{d}\tau}(\chi_\tau^2)}\right)^2 \mathrm{d}\tau, \tag{80}$$

$$= \int_T^t \frac{\mathrm{d}}{\mathrm{d}\tau}(\chi_t^2)\,\mathrm{d}\tau, \tag{81}$$

$$= \chi_t^2 - \chi_T^2 = \frac{\sigma_t^2}{\alpha_t^2} - \frac{\sigma_T^2}{\alpha_T^2}. \tag{82}$$

Now we have a deterministic mapping from the original time to our new time via $A_t$. Now in general for any valid choice of $(\alpha_t, \sigma_t)$ we don't necessarily have that $\langle M \rangle_\infty^{ii} = \infty$ almost surely and as such we may need to enlarge the underlying probability space. Our constructed martingale can be expressed as time-changed Brownian motion, see Theorem B.2, such that $M_t = \overline{W}_{A_t}$ were $\overline{W}_{\chi^2}$ is the standard $d$-dimensional Brownian motion with time variable $\chi^2$ in reverse-time.

Now we can rewrite Equation (68) in differential form as

$$\mathrm{d}M_t = \mathrm{d}\overline{W}_{A_t}. \tag{83}$$

Because Brownian motion is time-shift invariant we can then write

$$\mathrm{d}M_t = \mathrm{d}\overline{W}_{\chi_t^2}. \tag{84}$$

$\square$

**Remark B.4.** The constraint of $\alpha_T > 0$ is important to ensure that $\chi_T$ is finite which is necessary due

$$\overline{W}_{\chi_t^2} = W_{\chi_T^2} - W_{\chi_t^2}. \tag{85}$$

In practice this is satisfied with a number of noise schedules of diffusion models (see Appendix F.1).

## C. Derivation of Rex

We derive the *Rex* scheme presented in Proposition 3.2 in the main paper.

**Nomenclature.** As alluded to earlier, there exist two popular reparameterizations of the score function which are used widely in practice, namely the noise prediction (Ho et al., 2020) and data prediction (Kingma et al., 2021) formulations. Following the conventions of Lipman et al. (2024) we write noise prediction model as $x_{T|t}(x) = \mathbb{E}[X_T|X_t = x]$ and write data prediction model as $x_{0|t}(x) = \mathbb{E}[X_0|X_t = x]$. Throughout this paper we will assume the existence of *sufficiently trained* diffusion models denoted $x_{T|t}^\theta(x) \approx x_{T|t}(x)$ and $x_{0|t}^\theta(x) \approx x_{0|t}(x)$.

## C.1. Rex (ODE)

In this section we derive the *Rex* scheme for the probability flow ODE. We present derivations for both the data prediction and noise prediction formulations. It is well known (Blasingame & Liu, 2025, Equation (19); *cf*. Domingo-Enrich et al., 2025, Equation (8)) that the ODE in Equation (6) can be rewritten as

$$\frac{\mathrm{d}\boldsymbol{x}_t}{\mathrm{d}t} = \frac{\dot{\beta}_t}{\beta_t}\boldsymbol{x}_t + \frac{\sigma_t\dot{\alpha}_t - \dot{\sigma}_t\alpha_t}{\beta_t}\boldsymbol{f}_\theta(t, \boldsymbol{x}_t), \tag{86}$$

where $\beta_t = -\alpha_t$ for noise prediction and $\beta_t = \sigma_t$ for target prediction. This choice of $\beta$ and $\boldsymbol{f}_\theta$ thus depends on the particulars of the noise or data reparameterization.

**Remark C.1.** Without loss of generality any of the results for the probability flow ODE apply to any arbitrary flow model which models an *affine probability path* (Lipman et al., 2024) with the correct conversions to the flow matching conventions.[19]

It is well observed that the structure of the ODE in Equation (86) can be greatly simplified via *exponential integrators* (Lu et al., 2022b; Zhang & Chen, 2023; Blasingame & Liu, 2024a). We make use of this insight to rewrite the ODE in a form which eliminates the discretization error in the $f(t)\boldsymbol{x}_t$ linear term along with a time reparameterization which will simplify the construction of the reversible solver. To accomplish this, we use exponential integrators in the form of a change-of-variables with $\boldsymbol{y}_t = \exp(-\int_T^t \frac{\dot{\beta}_u}{\beta_u}\,\mathrm{d}u)\boldsymbol{x}_t = \frac{\beta_T}{\beta_t}\boldsymbol{x}_t$. *N.B.*, we integrate from time $T$ to $t$ because the ODE in Equation (86) is defined in *reverse-time*. To achieve the time reparametrization we introduce a new variable $\varsigma_t$ defined as the *signal-to-noise ratio* (SNR) $\alpha_t/\sigma_t$ for the data prediction formulation and defined as the inverse SNR $\sigma_t/\alpha_t$ for the noise prediction formulation. Using this time change we find Proposition C.1, in Appendix C.1.1 we provide the full derivation of this result.

### C.1.1. PROOF OF PROPOSITION C.1

We state Proposition C.1 below.

**Proposition C.1.** *The probability flow ODE in Equation (86) can be rewritten in $\varsigma_t$ as*

$$\frac{\mathrm{d}\boldsymbol{y}_\varsigma}{\mathrm{d}\varsigma} = |\beta_T|\boldsymbol{f}_\theta\left(\varsigma, \frac{\beta_\varsigma}{\beta_T}\boldsymbol{y}_\varsigma\right), \tag{87}$$

*where $\boldsymbol{y}_t = \frac{\beta_T}{\beta_t}\boldsymbol{x}_t$ and $\varsigma_t = \frac{\alpha_t\sigma_t}{\beta_t^2}$.*

*Proof.* Recall that from Equation (86) we have that the ODE is given by

$$\frac{\mathrm{d}\boldsymbol{x}_t}{\mathrm{d}t} = \frac{\dot{\beta}_t}{\beta_t}\boldsymbol{x}_t + \frac{\sigma_t\dot{\alpha}_t - \dot{\sigma}_t\alpha_t}{\beta_t}\boldsymbol{f}_\theta(t, \boldsymbol{x}_t). \tag{88}$$

We can use the technique of exponential integrators to rewrite the ODE as

$$\frac{\mathrm{d}}{\mathrm{d}t}\left[e^{\int_T^t -\frac{\dot{\beta}_u}{\beta_u}\,\mathrm{d}u}\boldsymbol{x}_t\right] = e^{\int_T^t -\frac{\dot{\beta}_u}{\beta_u}\,\mathrm{d}u}\frac{\sigma_t\dot{\alpha}_t - \dot{\sigma}_t\alpha_t}{\beta_t}\boldsymbol{f}_\theta(t, \boldsymbol{x}_t), \tag{89}$$

recalling that we integrate from initial time $T$ in reverse-time. Then the exponential terms simplify to

$$e^{\int_T^t -\frac{\dot{\beta}_u}{\beta_u}\,\mathrm{d}u} = \frac{\beta_T}{\beta_t}. \tag{90}$$

We introduce a *change-of-variables* $\boldsymbol{y}_t = \frac{\beta_T}{\beta_t}\boldsymbol{x}_t$ to rewrite the ODE as

$$\frac{\mathrm{d}\boldsymbol{y}_t}{\mathrm{d}t} = \underbrace{\frac{\beta_T}{\beta_t}\frac{\sigma_t\dot{\alpha}_t - \dot{\sigma}_t\alpha_t}{\beta_t}}_{=\upsilon_t}\boldsymbol{f}_\theta\left(t, \frac{\beta_t}{\beta_T}\boldsymbol{y}_t\right). \tag{91}$$

---

[19]*I.e.*, sampling in forward-time such that $\boldsymbol{X}_1 \sim q(\boldsymbol{X})$ and $\boldsymbol{X}_0 \sim p(\boldsymbol{X})$.

Next we define

$$\dot{\varsigma}_t = \operatorname{sgn}(\beta_T) \frac{\sigma_t \dot{\alpha}_t - \dot{\sigma}_t \alpha_t}{\beta_t^2}, \tag{92}$$

which we will now justify. Now recall that $\beta_t$ is either $-\alpha_t$ or $\sigma_t$ depending on the whether $\boldsymbol{f}_\theta$ denotes the data or noise prediction model. Moreover we know that $\alpha_t$ is a strictly monotonically decreasing in $t$ and that $\sigma_t$ is a strictly monotonically increasing in $t$. We will now prove that there exists an inverse function for $\varsigma_t$ such that $t_\varsigma(\varsigma_t) = t$ for both cases.

**Case $\beta_t = -\alpha_t$.** We can write $\upsilon_t$ as

$$\upsilon_t = \alpha_T \frac{\dot{\sigma}_t \alpha_t - \sigma_t \dot{\alpha}_t}{\alpha_t^2}, \tag{93}$$

$$\stackrel{(i)}{=} \alpha_T \frac{\mathrm{d}}{\mathrm{d}t}\left(\frac{\sigma_t}{\alpha_t}\right), \tag{94}$$

where (i) holds by the quotient rule. Clearly, we have that

$$\dot{\varsigma}_t = \frac{\mathrm{d}}{\mathrm{d}t}\left(\frac{\sigma_t}{\alpha_t}\right), \tag{95}$$

$$\varsigma_t = \frac{\sigma_t}{\alpha_t}, \tag{96}$$

It follows from $(\alpha_t, \sigma_t)$ that $\varsigma_t$ is strictly monotonically increasing in $t$ and thus we can construct its inverse.

**Case $\beta_t = \sigma_t$.** We can write $\upsilon_t$ as

$$\upsilon_t = \sigma_T \frac{\sigma_t \dot{\alpha}_t - \dot{\sigma}_t \alpha_t}{\sigma_t^2}, \tag{97}$$

$$\stackrel{(i)}{=} \sigma_T \frac{\mathrm{d}}{\mathrm{d}t}\left(\frac{\alpha_t}{\sigma_t}\right), \tag{98}$$

where (i) holds by the quotient rule. Clearly, we have that

$$\dot{\varsigma}_t = \frac{\mathrm{d}}{\mathrm{d}t}\left(\frac{\alpha_t}{\sigma_t}\right), \tag{99}$$

$$\varsigma_t = \frac{\alpha_t}{\sigma_t}, \tag{100}$$

It follows from $(\alpha_t, \sigma_t)$ that $\varsigma_t$ is strictly monotonically decreasing in $t$ and thus we can construct its inverse.

Thus we can rewrite the ODE via a time-change to find

$$\frac{\mathrm{d}\boldsymbol{y}_\varsigma}{\mathrm{d}\varsigma} = |\beta_T| \boldsymbol{f}_\theta\left(\varsigma, \frac{\beta_\varsigma}{\beta_T}\boldsymbol{y}_\varsigma\right), \tag{101}$$

with the usual *abuse-of-notation* $\boldsymbol{y}_\varsigma := \boldsymbol{y}_{t_\varsigma(\varsigma)}$, $\beta_\varsigma := \beta_{t_\varsigma(\varsigma)}$, &c. $\qquad\square$

**Remark C.2.** When in the noise prediction formulation with Proposition C.1 we recover the following reparameterization of Equation (86)

$$\frac{\mathrm{d}\boldsymbol{z}_\chi}{\mathrm{d}\chi} = \alpha_T \boldsymbol{x}_{T|\chi}^\theta\left(\frac{\alpha_\chi}{\alpha_T}\boldsymbol{z}_\chi\right), \tag{102}$$

where $\alpha_T > 0$, $\boldsymbol{z}_t = \frac{\alpha_T}{\alpha_t}\boldsymbol{x}_t$ and $\chi_t = \frac{\sigma_t}{\alpha_t}$, which has been observed by numerous prior works (see Song et al., 2021a, Equation (14); Pan et al., 2023, Equation (11); Wang et al., 2024, Equation (6)).

**Remark C.3.** When in the data prediction formulation, Proposition C.1 recovers Blasingame & Liu (2025, Proposition D.2) which states that Equation (86) can be written as

$$\frac{\mathrm{d}\boldsymbol{y}_\gamma}{\mathrm{d}\gamma} = \sigma_T \boldsymbol{x}_{0|\gamma}^\theta\left(\frac{\sigma_\gamma}{\sigma_T}\boldsymbol{y}_\gamma\right), \tag{103}$$

where $\boldsymbol{y}_t = \frac{\sigma_T}{\sigma_t}\boldsymbol{x}_t$ and $\gamma_t = \frac{\alpha_t}{\sigma_t}$.

C.1.2. DATA PREDICTION

We present this derivation in the form of Lemma C.2 below.

---

**Lemma C.2** (*Rex* (ODE) for data prediction models)**.** *Let* $\boldsymbol{\Phi}$ *be an explicit Runge-Kutta solver for the ODE in Equation* (103) *with Butcher tableau* $a_{ij}$, $b_i$, $c_i$. *The reversible solver for* $\boldsymbol{\Phi}$ *in terms of the original state* $\boldsymbol{x}_t$ *is given by the forward step*

$$
\begin{aligned}
\boldsymbol{x}_{n+1} &= \frac{\sigma_{n+1}}{\sigma_n} \left( \zeta \boldsymbol{x}_n + (1 - \zeta)\hat{\boldsymbol{x}}_n \right) + \sigma_{n+1}\boldsymbol{\Psi}_h(\gamma_n, \hat{\boldsymbol{x}}_n), \\
\hat{\boldsymbol{x}}_{n+1} &= \frac{\sigma_{n+1}}{\sigma_n}\hat{\boldsymbol{x}}_n - \sigma_{n+1}\boldsymbol{\Psi}_{-h}(\gamma_{n+1}, \boldsymbol{x}_{n+1}),
\end{aligned}
\tag{104}
$$

*and backward step*

$$
\begin{aligned}
\hat{\boldsymbol{x}}_n &= \frac{\sigma_n}{\sigma_{n+1}}\hat{\boldsymbol{x}}_{n+1} + \sigma_n\boldsymbol{\Psi}_{-h}(\gamma_{n+1}, \boldsymbol{x}_{n+1}), \\
\boldsymbol{x}_n &= \frac{\sigma_n}{\sigma_{n+1}}\zeta^{-1}\boldsymbol{x}_{n+1} + (1 - \zeta^{-1})\hat{\boldsymbol{x}}_n - \sigma_n\zeta^{-1}\boldsymbol{\Psi}_h(\gamma_n, \hat{\boldsymbol{x}}_n),
\end{aligned}
\tag{105}
$$

*with step size* $h := \gamma_{n+1} - \gamma_n$ *and where* $\boldsymbol{\Psi}$ *denotes the following scheme*

$$
\begin{aligned}
\hat{\boldsymbol{z}}_i &= \frac{1}{\sigma_n}\boldsymbol{x}_n + h\sum_{j=1}^{i-1} a_{ij}\boldsymbol{x}_{0|\gamma_n+c_j h}^\theta(\sigma_{\gamma_n+c_j h}\hat{\boldsymbol{z}}_j), \\
\boldsymbol{\Psi}_h(\gamma_n, \boldsymbol{x}_n) &= h\sum_{i=1}^{s} b_i\boldsymbol{x}_{0|\gamma_n+c_i h}^\theta(\sigma_{\gamma_n+c_i h}\hat{\boldsymbol{z}}_i),
\end{aligned}
\tag{106}
$$

---

*Proof.* Recall that the forward step of the McCallum-Foster method for Equation (103) given $\boldsymbol{\Phi}$ is given as

$$
\begin{aligned}
\boldsymbol{y}_{n+1} &= \zeta \boldsymbol{y}_n + (1 - \zeta)\hat{\boldsymbol{y}}_n + \boldsymbol{\Phi}_h(\gamma_n, \hat{\boldsymbol{y}}_n), \\
\hat{\boldsymbol{y}}_{n+1} &= \hat{\boldsymbol{y}}_n - \boldsymbol{\Phi}_{-h}(\gamma_{n+1}, \boldsymbol{y}_{n+1}),
\end{aligned}
\tag{107}
$$

with step size $h = \gamma_{n+1} - \gamma_n$. We use the definition of $\boldsymbol{y}_t = \frac{\sigma_T}{\sigma_t}\boldsymbol{x}_t$ to rewrite the forward pass as

$$
\begin{aligned}
\boldsymbol{x}_{n+1} &= \frac{\sigma_{n+1}}{\sigma_n}\left(\zeta \boldsymbol{x}_n + (1 - \zeta)\hat{\boldsymbol{x}}_n\right) + \frac{\sigma_{n+1}}{\sigma_T}\boldsymbol{\Phi}_h\left(\gamma_n, \frac{\sigma_T}{\sigma_n}\hat{\boldsymbol{x}}_n\right), \\
\hat{\boldsymbol{x}}_{n+1} &= \frac{\sigma_{n+1}}{\sigma_n}\hat{\boldsymbol{x}}_n - \frac{\sigma_{n+1}}{\sigma_T}\boldsymbol{\Phi}_{-h}\left(\gamma_{n+1}, \frac{\sigma_T}{\sigma_{n+1}}\boldsymbol{x}_{n+1}\right).
\end{aligned}
\tag{108}
$$

*Mutatis mutandis* we find the backward step in $\boldsymbol{x}_t$ to be given as

$$
\begin{aligned}
\hat{\boldsymbol{x}}_n &= \frac{\sigma_n}{\sigma_{n+1}}\hat{\boldsymbol{x}}_{n+1} + \frac{\sigma_n}{\sigma_T}\boldsymbol{\Phi}_{-h}\left(\gamma_{n+1}, \frac{\sigma_T}{\sigma_{n+1}}\boldsymbol{x}_{n+1}\right), \\
\boldsymbol{x}_n &= \frac{\sigma_n}{\sigma_{n+1}}\zeta^{-1}\boldsymbol{x}_{n+1} + (1 - \zeta^{-1})\hat{\boldsymbol{x}}_n - \frac{\sigma_n}{\sigma_T}\zeta^{-1}\boldsymbol{\Phi}_h\left(\gamma_n, \frac{\sigma_T}{\sigma_n}\hat{\boldsymbol{x}}_n\right),
\end{aligned}
\tag{109}
$$

Next we simplify the explicit RK scheme $\boldsymbol{\Phi}(\gamma_n, \boldsymbol{y}_n)$ for the time-changed probability flow ODE in Equation (103). Recall that the RK scheme can be written as

$$
\begin{aligned}
\boldsymbol{z}_i &= \boldsymbol{y}_n + h\sum_{j=1}^{i-1} a_{ij}\sigma_T\boldsymbol{x}_{0|\gamma_n+c_j h}\left(\frac{\sigma_{\gamma_n+c_j h}}{\sigma_T}\boldsymbol{z}_j\right), \\
\boldsymbol{\Phi}_h(\gamma_n, \boldsymbol{y}_n) &= h\sum_{i=1}^{s} b_i\sigma_T\boldsymbol{x}_{0|\gamma_n+c_i h}\left(\frac{\sigma_{\gamma_n+c_i h}}{\sigma_T}\boldsymbol{z}_i\right).
\end{aligned}
\tag{110}
$$

Next, we replace $\boldsymbol{y}_t$ back with $\boldsymbol{x}_t$ which yields

$$\boldsymbol{z}_i = \sigma_T \left( \frac{1}{\sigma_n} \boldsymbol{x}_n + h \sum_{j=1}^{i-1} a_{ij} \boldsymbol{x}_{0|\gamma_n+c_j h} \left( \frac{\sigma_{\gamma_n+c_j h}}{\sigma_T} \boldsymbol{z}_j \right) \right),$$

$$\boldsymbol{\Phi}_h \left( \gamma_n, \frac{\sigma_T}{\sigma_n} \boldsymbol{x}_n \right) = \sigma_T h \sum_{i=1}^{s} b_i \boldsymbol{x}_{0|\gamma_n+c_i h} \left( \frac{\sigma_{\gamma_n+c_i h}}{\sigma_T} \boldsymbol{z}_i \right). \tag{111}$$

To further simplify let $\sigma_T \hat{\boldsymbol{z}}_i = \boldsymbol{z}_i$ and define $\boldsymbol{\Psi}_h(\gamma_n, \boldsymbol{x}_n) := \sigma_T \boldsymbol{\Phi}(\gamma_n, \frac{\sigma_T}{\sigma_n} \boldsymbol{x}_n)$.

Thus we can write the following reversible scheme with forward step

$$\boldsymbol{x}_{n+1} = \frac{\sigma_{n+1}}{\sigma_n} (\zeta \boldsymbol{x}_n + (1-\zeta)\hat{\boldsymbol{x}}_n) + \sigma_{n+1} \boldsymbol{\Psi}_h(\gamma_n, \hat{\boldsymbol{x}}_n),$$

$$\hat{\boldsymbol{x}}_{n+1} = \frac{\sigma_{n+1}}{\sigma_n} \hat{\boldsymbol{x}}_n - \sigma_{n+1} \boldsymbol{\Psi}_{-h}(\gamma_{n+1}, \boldsymbol{x}_{n+1}), \tag{112}$$

and the backward step

$$\hat{\boldsymbol{x}}_n = \frac{\sigma_n}{\sigma_{n+1}} \hat{\boldsymbol{x}}_{n+1} + \sigma_n \boldsymbol{\Psi}_{-h}(\gamma_{n+1}, \boldsymbol{x}_{n+1}),$$

$$\boldsymbol{x}_n = \frac{\sigma_n}{\sigma_{n+1}} \zeta^{-1} \boldsymbol{x}_{n+1} + (1-\zeta^{-1})\hat{\boldsymbol{x}}_n - \sigma_n \zeta^{-1} \boldsymbol{\Psi}_h(\gamma_n, \hat{\boldsymbol{x}}_n), \tag{113}$$

with the numerical scheme

$$\hat{\boldsymbol{z}}_i = \frac{1}{\sigma_n} \boldsymbol{x}_n + h \sum_{j=1}^{i-1} a_{ij} \boldsymbol{x}_{0|\gamma_n+c_j h}^\theta (\sigma_{\gamma_n+c_j h} \hat{\boldsymbol{z}}_j),$$

$$\boldsymbol{\Psi}_h(\gamma_n, \boldsymbol{x}_n) = h \sum_{i=1}^{s} b_i \boldsymbol{x}_{0|\gamma_n+c_i h}^\theta (\sigma_{\gamma_n+c_i h} \hat{\boldsymbol{z}}_i). \tag{114}$$

$\square$

### C.1.3. NOISE PREDICTION

We present this derivation in the form of Lemma C.3 below.

**Lemma C.3** (*Rex* (ODE) for noise prediction models). *Let $\boldsymbol{\Phi}$ be an explicit Runge-Kutta solver for the ODE in Equation (102) with Butcher tableau $a_{ij}, b_i, c_i$. The reversible solver for $\boldsymbol{\Phi}$ in terms of the original state $\boldsymbol{x}_t$ is given by the forward step*

$$\boldsymbol{x}_{n+1} = \frac{\alpha_{n+1}}{\alpha_n} (\zeta \boldsymbol{x}_n + (1-\zeta)\hat{\boldsymbol{x}}_n) + \alpha_{n+1} \boldsymbol{\Psi}_h(\chi_n, \hat{\boldsymbol{x}}_n),$$

$$\hat{\boldsymbol{x}}_{n+1} = \frac{\alpha_{n+1}}{\alpha_n} \hat{\boldsymbol{x}}_n - \alpha_{n+1} \boldsymbol{\Psi}_{-h}(\chi_{n+1}, \boldsymbol{x}_{n+1}), \tag{115}$$

*and backward step*

$$\hat{\boldsymbol{x}}_n = \frac{\alpha_n}{\alpha_{n+1}} \hat{\boldsymbol{x}}_{n+1} + \alpha_n \boldsymbol{\Psi}_{-h}(\chi_{n+1}, \boldsymbol{x}_{n+1}),$$

$$\boldsymbol{x}_n = \frac{\alpha_n}{\alpha_{n+1}} \zeta^{-1} \boldsymbol{x}_{n+1} + (1-\zeta^{-1})\hat{\boldsymbol{x}}_n - \alpha_n \zeta^{-1} \boldsymbol{\Psi}_h(\chi_n, \hat{\boldsymbol{x}}_n), \tag{116}$$

*with step size $h := \chi_{n+1} - \chi_n$ and where $\boldsymbol{\Psi}$ denotes the following scheme*

$$\hat{\boldsymbol{z}}_i = \frac{1}{\alpha_n} \boldsymbol{x}_n + h \sum_{j=1}^{i-1} a_{ij} \boldsymbol{x}_{T|\chi_n+c_j h}^\theta (\alpha_{\chi_n+c_j h} \hat{\boldsymbol{z}}_j),$$

$$\boldsymbol{\Psi}_h(\chi_n, \boldsymbol{x}_n) = h \sum_{i=1}^{s} b_i \boldsymbol{x}_{T|\chi_n+c_i h}^\theta (\alpha_{\chi_n+c_i h} \hat{\boldsymbol{z}}_i), \tag{117}$$

*Proof.* Recall that the forward step of the McCallum-Foster method for Equation (102) given $\boldsymbol{\Phi}$ is given as

$$
\begin{aligned}
\boldsymbol{z}_{n+1} &= \zeta \boldsymbol{z}_n + (1 - \zeta)\hat{\boldsymbol{z}}_n + \boldsymbol{\Phi}_h(\chi_n, \hat{\boldsymbol{z}}_n), \\
\hat{\boldsymbol{z}}_{n+1} &= \hat{\boldsymbol{z}}_n - \boldsymbol{\Phi}_{-h}(\chi_{n+1}, \boldsymbol{z}_{n+1}),
\end{aligned}
\tag{118}
$$

with step size $h = \chi_{n+1} - \chi_n$. We use the definition of $\boldsymbol{z}_t = \frac{\alpha_T}{\alpha_t}\boldsymbol{x}_t$ to rewrite the forward pass as

$$
\begin{aligned}
\boldsymbol{x}_{n+1} &= \frac{\alpha_{n+1}}{\alpha_n} \left(\zeta \boldsymbol{x}_n + (1 - \zeta)\hat{\boldsymbol{x}}_n\right) + \frac{\alpha_{n+1}}{\alpha_T} \boldsymbol{\Phi}_h\left(\chi_n, \frac{\alpha_T}{\alpha_n}\hat{\boldsymbol{x}}_n\right), \\
\hat{\boldsymbol{x}}_{n+1} &= \frac{\alpha_{n+1}}{\alpha_n}\hat{\boldsymbol{x}}_n - \frac{\alpha_{n+1}}{\alpha_T} \boldsymbol{\Phi}_{-h}\left(\chi_{n+1}, \frac{\alpha_T}{\alpha_{n+1}}\boldsymbol{x}_{n+1}\right).
\end{aligned}
\tag{119}
$$

*Mutatis mutandis* we find the backward step in $\boldsymbol{x}_t$ to be given as

$$
\begin{aligned}
\hat{\boldsymbol{x}}_n &= \frac{\alpha_n}{\alpha_{n+1}}\hat{\boldsymbol{x}}_{n+1} + \frac{\alpha_n}{\alpha_T} \boldsymbol{\Phi}_{-h}\left(\chi_{n+1}, \frac{\alpha_T}{\alpha_{n+1}}\boldsymbol{x}_{n+1}\right), \\
\boldsymbol{x}_n &= \frac{\alpha_n}{\alpha_{n+1}}\zeta^{-1}\boldsymbol{x}_{n+1} + (1 - \zeta^{-1})\hat{\boldsymbol{x}}_n - \frac{\alpha_n}{\alpha_T}\zeta^{-1}\boldsymbol{\Phi}_h\left(\chi_n, \frac{\alpha_T}{\alpha_n}\hat{\boldsymbol{x}}_n\right),
\end{aligned}
\tag{120}
$$

Next we simplify the explicit RK scheme $\boldsymbol{\Phi}(\chi_n, \boldsymbol{z}_n)$ for the time-changed probability flow ODE in Equation (103). Recall that the RK scheme can be written as

$$
\begin{aligned}
\boldsymbol{z}_i &= \boldsymbol{z}_n + h\sum_{j=1}^{i-1} a_{ij}\alpha_T \boldsymbol{x}_{0|\chi_n+c_j h}\left(\frac{\alpha_{\chi_n+c_j h}}{\alpha_T}\boldsymbol{z}_j\right), \\
\boldsymbol{\Phi}_h(\chi_n, \boldsymbol{z}_n) &= h\sum_{i=1}^{s} b_i\alpha_T \boldsymbol{x}_{0|\chi_n+c_i h}\left(\frac{\alpha_{\chi_n+c_i h}}{\alpha_T}\boldsymbol{z}_i\right).
\end{aligned}
\tag{121}
$$

Next, we replace $\boldsymbol{z}_t$ back with $\boldsymbol{x}_t$ which yields

$$
\begin{aligned}
\boldsymbol{z}_i &= \alpha_T \left(\frac{1}{\alpha_n}\boldsymbol{x}_n + h\sum_{j=1}^{i-1} a_{ij}\boldsymbol{x}_{0|\chi_n+c_j h}\left(\frac{\alpha_{\chi_n+c_j h}}{\alpha_T}\boldsymbol{z}_j\right)\right), \\
\boldsymbol{\Phi}_h\left(\chi_n, \frac{\alpha_T}{\alpha_n}\boldsymbol{x}_n\right) &= \alpha_T h\sum_{i=1}^{s} b_i\boldsymbol{x}_{0|\chi_n+c_i h}\left(\frac{\alpha_{\chi_n+c_i h}}{\alpha_T}\boldsymbol{z}_i\right).
\end{aligned}
\tag{122}
$$

To further simplify let $\alpha_T \hat{\boldsymbol{z}}_i = \boldsymbol{z}_i$ and define $\boldsymbol{\Psi}_h(\chi_n, \boldsymbol{x}_n) := \alpha_T \boldsymbol{\Phi}(\chi_n, \frac{\alpha_T}{\alpha_n}\boldsymbol{x}_n)$.

Thus we can write the following reversible scheme with forward step

$$
\begin{aligned}
\boldsymbol{x}_{n+1} &= \frac{\alpha_{n+1}}{\alpha_n} \left(\zeta \boldsymbol{x}_n + (1 - \zeta)\hat{\boldsymbol{x}}_n\right) + \alpha_{n+1}\boldsymbol{\Psi}_h(\chi_n, \hat{\boldsymbol{x}}_n), \\
\hat{\boldsymbol{x}}_{n+1} &= \frac{\alpha_{n+1}}{\alpha_n}\hat{\boldsymbol{x}}_n - \alpha_{n+1}\boldsymbol{\Psi}_{-h}(\chi_{n+1}, \boldsymbol{x}_{n+1}),
\end{aligned}
\tag{123}
$$

and the backward step

$$
\begin{aligned}
\hat{\boldsymbol{x}}_n &= \frac{\alpha_n}{\alpha_{n+1}}\hat{\boldsymbol{x}}_{n+1} + \alpha_n\boldsymbol{\Psi}_{-h}(\chi_{n+1}, \boldsymbol{x}_{n+1}), \\
\boldsymbol{x}_n &= \frac{\alpha_n}{\alpha_{n+1}}\zeta^{-1}\boldsymbol{x}_{n+1} + (1 - \zeta^{-1})\hat{\boldsymbol{x}}_n - \alpha_n\zeta^{-1}\boldsymbol{\Psi}_h(\chi_n, \hat{\boldsymbol{x}}_n),
\end{aligned}
\tag{124}
$$

with the numerical scheme

$$
\begin{aligned}
\hat{\boldsymbol{z}}_i &= \frac{1}{\alpha_n}\boldsymbol{x}_n + h\sum_{j=1}^{i-1} a_{ij}\boldsymbol{x}^\theta_{T|\chi_n+c_j h}(\alpha_{\chi_n+c_j h}\hat{\boldsymbol{z}}_j), \\
\boldsymbol{\Psi}_h(\chi_n, \boldsymbol{x}_n) &= h\sum_{i=1}^{s} b_i\boldsymbol{x}^\theta_{T|\chi_n+c_i h}(\alpha_{\chi_n+c_i h}\hat{\boldsymbol{z}}_i).
\end{aligned}
\tag{125}
$$

$\square$

## C.2. Rex (SDE)

In this section we derive the *Rex* scheme for the reverse-time diffusion SDE along with several helper derivations. We rely on the time-changed Brownian motion machinery developed in Appendix B.5, and proceed by reparameterizing Equation (7) (Appendix C.2.1), specializing to the data prediction scenario (Appendix C.2.2), and then performing an analogous derivation for the noise prediction scenario (Appendix C.2.3).

### C.2.1. PROOF OF REPARAMETRIZED SEMI-LINEAR SDE WITH ADDITIVE NOISE

Recall our general semi-linear SDE with additive noise from Equation (7) which represents an abstraction (*mutatis mutandis*) of diffusion SDEs with the form,

$$\mathrm{d}\boldsymbol{X}_t = [a(t)\boldsymbol{X}_t + b(t)\boldsymbol{f}_\theta(t, \boldsymbol{X}_t)]\ \mathrm{d}t + g(t)\ \mathrm{d}\boldsymbol{W}_t. \tag{126}$$

We now restate Proposition 3.1 below.

**Proposition 3.1.** *Assume that* $g(t) = \sqrt{a(t)b(t)}$, $b(t)/a(t) \to \infty$ *as* $t \to \infty$, *and* $b(t) > 0$. *Let* $\Xi(t) := \exp\left(\int_0^t a(\tau)\ \mathrm{d}\tau\right)$ *be the reciprocal of the integrating factor of Equation* (7). *Then the SDE in Equation* (7) *can be rewritten as*

$$\mathrm{d}\boldsymbol{Y}_\varsigma = \boldsymbol{f}_\theta(\varsigma, \Xi(\varsigma)\boldsymbol{Y}_\varsigma)\ \mathrm{d}\varsigma + \mathrm{d}\boldsymbol{W}_\varsigma, \tag{8}$$

*where* $\boldsymbol{Y}_t = \Xi^{-1}(t)\boldsymbol{X}_t$ *and* $\varsigma_t = \int \Xi^{-1}(t)b(t)\ \mathrm{d}t$.

*Proof.* Let $\Xi(t)$ be defined as the reciprocal of the integrating factor of Equation (7), *i.e.*,

$$\Xi(t) = \exp\left(\int_0^t a(s)\ \mathrm{d}s\right). \tag{127}$$

We can now write Equation (7) as

$$\mathrm{d}\left[\Xi^{-1}(t)\boldsymbol{X}_t\right] = \frac{b(t)}{\Xi(t)}\boldsymbol{f}_\theta(t, \boldsymbol{X}_t)\ \mathrm{d}t + \frac{g(t)}{\Xi(t)}\ \mathrm{d}\boldsymbol{W}_t, \tag{128}$$

$$\mathrm{d}\boldsymbol{Y}_t \overset{(i)}{=} \frac{b(t)}{\Xi(t)}\boldsymbol{f}_\theta(t, \Xi(t)\boldsymbol{Y}_t)\ \mathrm{d}t + \frac{g(t)}{\Xi(t)}\ \mathrm{d}\boldsymbol{W}_t, \tag{129}$$

where (i) holds via $\boldsymbol{Y}_t := \Xi^{-1}(t)\boldsymbol{X}_t$. Now we wish to introduce a time-changed variable $\varsigma_t$, to simplify the calculation into the form

$$\mathrm{d}\boldsymbol{Y}_t = \frac{\mathrm{d}\varsigma_t}{\mathrm{d}t}\boldsymbol{f}_\theta(t, \Xi(t)\boldsymbol{Y}_t)\ \mathrm{d}t + \sqrt{\frac{\mathrm{d}\varsigma_t}{\mathrm{d}t}}\ \mathrm{d}\boldsymbol{W}_t, \tag{130}$$

thus we have the equalities

$$\frac{\mathrm{d}\varsigma_t}{\mathrm{d}t} = \frac{b(t)}{\Xi(t)}, \tag{131}$$

$$\sqrt{\frac{\mathrm{d}\varsigma_t}{\mathrm{d}t}} = \frac{g(t)}{\Xi(t)}, \tag{132}$$

which means that since $\Xi, b, g$ are all positive functions we have

$$\frac{b(t)}{\Xi(t)} \overset{(i)}{=} \frac{g^2(t)}{\Xi^2(t)}, \tag{133}$$

$$\Xi(t) = \frac{g^2(t)}{b(t)}, \tag{134}$$

where (i) holds by the squaring Equation (132) and setting equal to $\dot{\varsigma}_t$. Then,

$$\int \frac{\mathrm{d}\varsigma_t}{\mathrm{d}t} \; \mathrm{d}t = \int \frac{b(t)}{\Xi(t)} \; \mathrm{d}t, \tag{135}$$

$$\varsigma_t = \int \frac{b^2(t)}{g^2(t)} \; \mathrm{d}t. \tag{136}$$

Now we need to perform the time-change of the Brownian motion. For this we can just follow Lemma B.3 *mutatis mutandis* to find

$$\sqrt{\frac{\mathrm{d}\varsigma_t}{\mathrm{d}t}} \; \mathrm{d}\boldsymbol{W}_t = \mathrm{d}\boldsymbol{W}_\varsigma, \tag{137}$$

with $\sigma_t \to \infty$ as $t \to \infty$ since $|\frac{g(t)}{b(t)}| = \mathcal{O}(\sqrt{t})$. Thus the SDE in Equation (7) becomes

$$\mathrm{d}\boldsymbol{Y}_\varsigma = \boldsymbol{f}_\theta(\varsigma, \Xi(\varsigma)\boldsymbol{Y}_\varsigma) \; \mathrm{d}\varsigma + \mathrm{d}\boldsymbol{W}_\varsigma, \tag{138}$$

with abuse of notation for functions of $\varsigma^{-1}(\varsigma_t) = t$ being denoted by $\varsigma$. *N.B.*, this inverse exists since $b(t) \neq 0$ for all $t \geq 0$. $\qquad\square$

### C.2.2. PROOF OF REPARAMETRIZED SDE FOR DATA PREDICTION MODELS

It is well known (Lu et al., 2022a) that the reverse-time diffusion SDE in Equation (5) can be rewritten in terms of the data prediction model as

$$\mathrm{d}\boldsymbol{X}_t = \left[ \left( f(t) + \frac{g^2(t)}{\sigma_t^2} \right) \boldsymbol{X}_t - \frac{\alpha_t g^2(t)}{\sigma_t^2} \boldsymbol{x}_{0|t}^\theta(\boldsymbol{X}_t) \right] \; \mathrm{d}t + g(t) \, \mathrm{d}\overline{\boldsymbol{W}}_t. \tag{139}$$

Remarkably, following a similar derivation to the one above for the probability flow ODE yields a time-changed SDE with a very similar form to the one above, sans the Brownian motion term and different weighting terms.

**Lemma C.4** (Time reparametrization of the reverse-time diffusion SDE). *The reverse-time SDE in Equation* (139) *can be rewritten in terms of the data prediction model as*

$$\mathrm{d}\boldsymbol{Y}_\varrho = \frac{\sigma_T}{\gamma_T} \boldsymbol{x}_{0|\varrho}^\theta \left( \frac{\gamma_T \sigma_\varrho}{\sigma_T \gamma_\varrho} \boldsymbol{Y}_\varrho \right) \; \mathrm{d}\varrho + \frac{\sigma_T}{\gamma_T} \, \mathrm{d}\boldsymbol{W}_\varrho, \tag{140}$$

*where $\boldsymbol{Y}_t = \frac{\sigma_T^2 \alpha_t}{\sigma_t^2 \alpha_T} \boldsymbol{X}_t$ and $\varrho_t := \frac{\alpha_t^2}{\sigma_t^2}$.*

*Proof.* We rewrite Equation (5) in terms of the data prediction model, using the identity

$$\nabla_{\boldsymbol{x}} \log p_t(\boldsymbol{x}) = -\frac{1}{\sigma_t^2} \boldsymbol{x} + \frac{\alpha_t}{\sigma_t^2} \boldsymbol{x}_{0|t}(\boldsymbol{x}), \tag{141}$$

to find

$$\mathrm{d}\boldsymbol{X}_t = \left[ \underbrace{\left( f(t) + \frac{g^2(t)}{\sigma_t^2} \right)}_{=a(t)} \boldsymbol{X}_t + \underbrace{\left( -\frac{\alpha_t g^2(t)}{\sigma_t^2} \right)}_{=b(t)} \boldsymbol{x}_{0|t}(\boldsymbol{X}_t) \right] \; \mathrm{d}t + g(t) \, \mathrm{d}\overline{\boldsymbol{W}}_t, \tag{142}$$

where

$$f(t) = \frac{\dot{\alpha}_t}{\alpha_t}, \quad g^2(t) = \dot{\sigma}_t^2 - 2\frac{\dot{\alpha}_t}{\alpha_t}\sigma_t^2 = -2\sigma_t^2 \frac{\mathrm{d}\log\gamma_t}{\mathrm{d}t}. \tag{143}$$

Next we find the integrating factor $\Xi_t = \exp\left(-\int_T^t a(u)\,\mathrm{d}u\right)$,

$$\Xi_t = \exp\left(\int_t^T \frac{\mathrm{d}\log\alpha_u}{\mathrm{d}u} + \frac{g^2(u)}{\sigma_u^2}\,\mathrm{d}u\right), \tag{144}$$

$$= \exp\left(\int_t^T \frac{\mathrm{d}\log\alpha_u}{\mathrm{d}u} - 2\frac{\mathrm{d}\log\gamma_u}{\mathrm{d}u}\,\mathrm{d}u\right), \tag{145}$$

$$= \exp\left(\int_t^T \frac{\mathrm{d}\log\alpha_u}{\mathrm{d}u} - 2\left[\frac{\mathrm{d}\log\alpha_u}{\mathrm{d}u} - \frac{\mathrm{d}\log\sigma_u}{\mathrm{d}u}\right]\mathrm{d}u\right), \tag{146}$$

$$= \exp\left(\int_t^T \frac{\mathrm{d}\log\sigma_u^2}{\mathrm{d}u} - \frac{\mathrm{d}\log\alpha_u}{\mathrm{d}u}\,\mathrm{d}u\right), \tag{147}$$

$$= \exp\left(\log\sigma_T^2 - \log\sigma_t^2 - (\log\alpha_T - \log\alpha_t)\right), \tag{148}$$

$$= \frac{\sigma_T^2\alpha_t}{\sigma_t^2\alpha_T}. \tag{149}$$

We can write the integrating factor in terms of $\gamma_t$ as

$$\Xi_t = \frac{\sigma_T\gamma_t}{\sigma_t\gamma_T}. \tag{150}$$

Moreover we can further simplify $b(t)$ as

$$b(t) = \frac{-\alpha_t g^2(t)}{\sigma_t^2}, \tag{151}$$

$$= 2\alpha_t\frac{\mathrm{d}\log\gamma_t}{\mathrm{d}t}. \tag{152}$$

Thus we can rewrite the SDE in Equation (142) as

$$\mathrm{d}\left[\frac{\sigma_T}{\gamma_T}\frac{\gamma_t}{\sigma_t}\boldsymbol{X}_t\right] = 2\frac{\sigma_T}{\gamma_T}\frac{\alpha_t}{\sigma_t}\gamma_t\frac{\mathrm{d}\log\gamma_t}{\mathrm{d}t}\boldsymbol{x}_{0|t}(\boldsymbol{X}_t)\,\mathrm{d}t + \frac{\sigma_T}{\gamma_T}\frac{\gamma_t}{\sigma_t}\sqrt{-2\sigma_t^2\frac{\mathrm{d}\log\gamma_t}{\mathrm{d}t}}\,\mathrm{d}\overline{\boldsymbol{W}}_t, \tag{153}$$

$$\mathrm{d}\boldsymbol{Y}_t \stackrel{(i)}{=} 2\frac{\sigma_T}{\gamma_T}\frac{\alpha_t}{\sigma_t}\gamma_t\frac{\mathrm{d}\log\gamma_t}{\mathrm{d}t}\boldsymbol{x}_{0|t}\left(\frac{\gamma_T\sigma_t}{\sigma_T\gamma_t}\boldsymbol{Y}_t\right)\,\mathrm{d}t + \frac{\sigma_T}{\gamma_T}\frac{\gamma_t}{\sigma_t}\sqrt{-2\sigma_t^2\frac{\mathrm{d}\log\gamma_t}{\mathrm{d}t}}\,\mathrm{d}\overline{\boldsymbol{W}}_t, \tag{154}$$

$$\mathrm{d}\boldsymbol{Y}_t = \frac{\sigma_T}{\gamma_T}\frac{\mathrm{d}\gamma_t^2}{\mathrm{d}t}\boldsymbol{x}_{0|t}\left(\frac{\gamma_T\sigma_t}{\sigma_T\gamma_t}\boldsymbol{Y}_t\right)\,\mathrm{d}t + \frac{\sigma_T}{\gamma_T}\sqrt{-\gamma_t^2\frac{\mathrm{d}\log\gamma_t^2}{\mathrm{d}t}}\,\mathrm{d}\overline{\boldsymbol{W}}_t, \tag{155}$$

$$\mathrm{d}\boldsymbol{Y}_t = \frac{\sigma_T}{\gamma_T}\frac{\mathrm{d}\gamma_t^2}{\mathrm{d}t}\boldsymbol{x}_{0|t}\left(\frac{\gamma_T\sigma_t}{\sigma_T\gamma_t}\boldsymbol{Y}_t\right)\,\mathrm{d}t + \frac{\sigma_T}{\gamma_T}\sqrt{-\frac{\mathrm{d}\gamma_t^2}{\mathrm{d}t}}\,\mathrm{d}\overline{\boldsymbol{W}}_t, \tag{156}$$

$$\mathrm{d}\boldsymbol{Y}_\varrho \stackrel{(ii)}{=} \frac{\sigma_T}{\gamma_T}\boldsymbol{x}_{0|\varrho}\left(\frac{\gamma_T\sigma_\varrho}{\sigma_T\gamma_\varrho}\boldsymbol{Y}_\varrho\right)\,\mathrm{d}\varrho + \frac{\sigma_T}{\gamma_T}\,\mathrm{d}\boldsymbol{W}_\varrho, \tag{157}$$

where (i) holds by the change-of-variables $\boldsymbol{Y}_t = \frac{\sigma_T\gamma_t}{\gamma_T\sigma_t}\boldsymbol{X}_t$ and (ii) holds by Lemma B.3. $\qquad\square$

### C.2.3. PROOF OF REPARAMETRIZED SDE FOR NOISE PREDICTION MODELS

**Lemma C.5** (Time reparametrization of the reverse-time diffusion SDE for noise prediction models). *The reverse-time SDE in Equation (5) can be rewritten in terms of the noise prediction model as*

$$\mathrm{d}\boldsymbol{Y}_\chi = 2\alpha_T\boldsymbol{x}_{T|\chi}^\theta\left(\frac{\alpha_\chi}{\alpha_T}\boldsymbol{Y}_\chi\right)\,\mathrm{d}\chi + \alpha_T\,\mathrm{d}\overline{\boldsymbol{W}}_{\chi^2}, \tag{158}$$

*where $\boldsymbol{Y}_t = \frac{\alpha_t}{\alpha_T}\boldsymbol{X}_t$ and $\chi_t := \frac{\sigma_t}{\alpha_t}$.*

*Proof.* We rewrite Equation (5) in terms of the noise prediction model to find

$$d\boldsymbol{X}_t = \left[ f(t)\boldsymbol{X}_t + \frac{g^2(t)}{\sigma_t}\boldsymbol{x}_{T|t}^\theta(\boldsymbol{X}_t) \right] dt + g(t) \, d\overline{\boldsymbol{W}}_t, \tag{159}$$

where

$$f(t) = \frac{\dot{\alpha}_t}{\alpha_t}, \quad g^2(t) = \dot{\sigma}_t^2 - 2\frac{\dot{\alpha}_t}{\alpha_t}\sigma_t^2 = -2\sigma_t^2 \frac{d\log\gamma_t}{dt}. \tag{160}$$

Next we find the integrating factor to be $\exp\left( -\int_T^t f(u) \, du \right) = \frac{\alpha_T}{\alpha_t}$. Moreover, we can further simplify $\frac{g^2(t)}{\sigma_t}$ as

$$\frac{g^2(t)}{\sigma_t} = -2\sigma_t \frac{d\log\gamma_t}{dt}, \tag{161}$$

$$= -2\sigma_t \frac{\dot{\gamma}_t}{\gamma_t}, \tag{162}$$

$$= -2\frac{\sigma_t}{\gamma_t}\frac{\dot{\alpha}_t\sigma_t - \alpha_t\dot{\sigma}_t}{\sigma_t^2}, \tag{163}$$

$$= -2\frac{\sigma_t^2}{\alpha_t}\frac{\dot{\alpha}_t\sigma_t - \alpha_t\dot{\sigma}_t}{\sigma_t^2}, \tag{164}$$

$$= 2\frac{\sigma_t^2}{\alpha_t}\frac{\alpha_t\dot{\sigma}_t - \dot{\alpha}_t\sigma_t}{\sigma_t^2}, \tag{165}$$

$$= 2\frac{\alpha_t\dot{\sigma}_t - \dot{\alpha}_t\sigma_t}{\alpha_t}, \tag{166}$$

$$\tag{167}$$

Let $\chi_t := \frac{\sigma_t}{\alpha_t} = \frac{1}{\gamma_t}$. Thus we can rewrite the SDE in Equation (159) as

$$d\left[ \frac{\alpha_T}{\alpha_t}\boldsymbol{X}_t \right] = \frac{\alpha_T}{\alpha_t}\frac{g^2(t)}{\sigma_t^2}\boldsymbol{x}_{T|t}^\theta(\boldsymbol{X}_t) \, dt + \frac{\alpha_T}{\alpha_t}\sqrt{-2\sigma_t^2\frac{d\log\gamma_t}{dt}} \, d\overline{\boldsymbol{W}}_t, \tag{168}$$

$$d\boldsymbol{Y}_t \overset{(i)}{=} \frac{\alpha_T}{\alpha_t}\frac{g^2(t)}{\sigma_t^2}\boldsymbol{x}_{T|t}^\theta\left( \frac{\alpha_t}{\alpha_T}\boldsymbol{Y}_t \right) dt + \frac{\alpha_T}{\alpha_t}\sqrt{-2\sigma_t^2\frac{d\log\gamma_t}{dt}} \, d\overline{\boldsymbol{W}}_t, \tag{169}$$

$$d\boldsymbol{Y}_t = 2\alpha_T\frac{\alpha_t\dot{\sigma}_t - \dot{\alpha}_t\sigma_t}{\alpha_t^2}\boldsymbol{x}_{T|t}^\theta\left( \frac{\alpha_t}{\alpha_T}\boldsymbol{Y}_t \right) dt + \frac{\alpha_T}{\alpha_t}\sqrt{-2\sigma_t^2\frac{d\log\gamma_t}{dt}} \, d\overline{\boldsymbol{W}}_t, \tag{170}$$

$$d\boldsymbol{Y}_t \overset{(ii)}{=} 2\alpha_T\dot{\chi}_t\boldsymbol{x}_{T|t}^\theta\left( \frac{\alpha_t}{\alpha_T}\boldsymbol{Y}_t \right) dt + \alpha_T\sqrt{-2\frac{\sigma_t^2}{\alpha_t^2}\frac{d\log\gamma_t}{dt}} \, d\overline{\boldsymbol{W}}_t, \tag{171}$$

$$d\boldsymbol{Y}_t = 2\alpha_T\dot{\chi}_t\boldsymbol{x}_{T|t}^\theta\left( \frac{\alpha_t}{\alpha_T}\boldsymbol{Y}_t \right) dt + \alpha_T\sqrt{\dot{\chi}_t^2} \, d\overline{\boldsymbol{W}}_t, \tag{172}$$

$$d\boldsymbol{Y}_\chi \overset{(iii)}{=} 2\alpha_T\boldsymbol{x}_{T|\chi}^\theta\left( \frac{\alpha_\chi}{\alpha_T}\boldsymbol{Y}_\chi \right) d\chi + \alpha_T \, d\overline{\boldsymbol{W}}_{\chi^2}, \tag{173}$$

$$\tag{174}$$

where (i) holds by the change-of-variables $\boldsymbol{Y}_t = \frac{\alpha_T}{\alpha_t}\boldsymbol{X}_t$, (ii) holds by

$$-2\frac{\sigma_t^2}{\alpha_t^2}\frac{d\log\gamma_t}{dt} = \frac{\sigma_t^2}{\alpha_t^2}\frac{d(-2\log\gamma_t)}{dt}, \tag{175}$$

$$= \frac{\sigma_t^2}{\alpha_t^2}\frac{d\log\chi_t^2}{dt}, \tag{176}$$

$$= \frac{\sigma_t^2}{\alpha_t^2}\frac{\dot{\chi}_t^2}{\chi_t^2}, \tag{177}$$

$$= \dot{\chi}_t^2, \tag{178}$$

and (iii) holds by Lemma B.3 *mutatis mutandis* for $\chi_t$. □

C.2.4. DERIVATION OF REX (SDE)

We present derivations for both the data prediction and noise prediction formulations.

**Data Prediction.** We present this derivation in the form of Lemma C.6 below.

> **Lemma C.6** (*Rex* (SDE) for data prediction models)**.** *Let $\boldsymbol{\Phi}$ be an explicit stochastic Runge-Kutta solver for the additive noise SDE in Equation* (140)*, we construct the following reversible scheme for diffusion models*
>
> $$
> \begin{aligned}
> \boldsymbol{X}_{n+1} &= \frac{\sigma_{n+1}\gamma_n}{\gamma_{n+1}\sigma_n}(\zeta \boldsymbol{X}_n + (1-\zeta)\hat{\boldsymbol{X}}_n) + \frac{\sigma_{n+1}}{\gamma_{n+1}}\boldsymbol{\Psi}_h(\varrho_n, \hat{\boldsymbol{X}}_n, \boldsymbol{W}_\varrho(\omega)), \\
> \hat{\boldsymbol{X}}_{n+1} &= \frac{\sigma_{n+1}\gamma_n}{\gamma_{n+1}\sigma_n}\hat{\boldsymbol{X}}_n - \frac{\sigma_{n+1}}{\gamma_{n+1}}\boldsymbol{\Psi}_{-h}(\varrho_{n+1}, \boldsymbol{X}_{n+1}, \boldsymbol{W}_\varrho(\omega)),
> \end{aligned}
> \tag{179}
> $$
>
> *and the backward step is given as*
>
> $$
> \begin{aligned}
> \hat{\boldsymbol{X}}_n &= \frac{\sigma_n\gamma_{n+1}}{\gamma_n\sigma_{n+1}}\hat{\boldsymbol{X}}_{n+1} + \frac{\sigma_n}{\gamma_n}\boldsymbol{\Psi}_{-h}(\varrho_{n+1}, \boldsymbol{X}_{n+1}, \boldsymbol{W}_\varrho(\omega)), \\
> \boldsymbol{X}_n &= \frac{\sigma_n\gamma_{n+1}}{\gamma_n\sigma_{n+1}}\zeta^{-1}\boldsymbol{X}_{n+1} + (1-\zeta^{-1})\hat{\boldsymbol{X}}_n - \frac{\sigma_n}{\gamma_n}\zeta^{-1}\boldsymbol{\Psi}_h(\varrho_n, \hat{\boldsymbol{X}}_n, \boldsymbol{W}_\varrho(\omega)),
> \end{aligned}
> \tag{180}
> $$
>
> *with step size $h := \varrho_{n+1} - \varrho_n$ and where $\boldsymbol{\Psi}$ denotes the following scheme*
>
> $$
> \begin{aligned}
> \hat{\boldsymbol{Z}}_i &= \frac{\gamma_n}{\sigma_n}\boldsymbol{X}_n + h\sum_{j=1}^{i-1}\left[a_{ij}\boldsymbol{x}_{0|\varrho_n+c_jh}^\theta\left(\frac{\sigma_{\varrho_n+c_jh}}{\gamma_{\varrho_n+c_jh}}\hat{\boldsymbol{Z}}_j\right)\right] + a_i^W\boldsymbol{W}_n + a_i^H\boldsymbol{H}_n, \\
> \boldsymbol{\Psi}_h(\varrho_n, \boldsymbol{X}_n, \boldsymbol{W}_\varrho(\omega)) &= h\sum_{j=1}^{s}\left[b_i\boldsymbol{x}_{0|\varrho_n+c_ih}^\theta\left(\frac{\sigma_{\varrho_n+c_ih}}{\gamma_{\varrho_n+c_ih}}\hat{\boldsymbol{Z}}_i\right)\right] + b^W\boldsymbol{W}_n + b^H\boldsymbol{H}_n.
> \end{aligned}
> \tag{181}
> $$

*Proof.* We write the SRK scheme for the time-changed reverse-time SDE in Equation (140) to construct the following SRK scheme

$$
\begin{aligned}
\boldsymbol{Z}_i &= \boldsymbol{Y}_n + h\sum_{j=1}^{i-1}\left[a_{ij}\frac{\sigma_T}{\gamma_T}\boldsymbol{x}_{0|\varrho_n+c_jh}\left(\frac{\gamma_T\sigma_{\varrho_n+c_jh}}{\sigma_T\gamma_{\varrho_n+c_jh}}\boldsymbol{Z}_j\right)\right] + \frac{\sigma_T}{\gamma_T}(a_i^W\boldsymbol{W}_n + a_i^H\boldsymbol{H}_n), \\
\boldsymbol{Y}_{n+1} &= \boldsymbol{Y}_n + h\sum_{i=1}^{s}\left[b_i\frac{\sigma_T}{\gamma_T}\boldsymbol{x}_{0|\varrho_n+c_ih}\left(\frac{\gamma_T\sigma_{\varrho_n+c_ih}}{\sigma_T\gamma_{\varrho_n+c_ih}}\boldsymbol{Z}_i\right)\right] + \frac{\sigma_T}{\gamma_T}(b^W\boldsymbol{W}_n + b^H\boldsymbol{H}_n),
\end{aligned}
\tag{182}
$$

with step size $h = \varrho_{n+1} - \varrho_n$. Next, we replace $\boldsymbol{Y}_t$ back with $\boldsymbol{X}_t$ which yields

$$
\begin{aligned}
\boldsymbol{Z}_i &= \frac{\sigma_T}{\gamma_T}\left(\frac{\gamma_n}{\sigma_n}\boldsymbol{X}_n + h\sum_{j=1}^{i-1}\left[a_{ij}\boldsymbol{x}_{0|\varrho_n+c_jh}\left(\frac{\gamma_T\sigma_{\varrho_n+c_jh}}{\sigma_T\gamma_{\varrho_n+c_jh}}\boldsymbol{Z}_j\right)\right]\right) \\
&\quad + \frac{\sigma_T}{\gamma_T}(a_i^W\boldsymbol{W}_n + a_i^H\boldsymbol{H}_n), \\
\frac{\sigma_T\gamma_{n+1}}{\gamma_T\sigma_{n+1}}\boldsymbol{X}_{n+1} &= \frac{\sigma_T\gamma_n}{\gamma_T\sigma_n}\boldsymbol{X}_n \\
&\quad + \frac{\sigma_T}{\gamma_T}\underbrace{h\sum_{i=1}^{s}\left[b_i\frac{\sigma_T}{\gamma_T}\boldsymbol{x}_{0|\varrho_n+c_ih}\left(\frac{\gamma_T\sigma_{\varrho_n+c_ih}}{\sigma_T\gamma_{\varrho_n+c_ih}}\boldsymbol{Z}_i\right)\right] + \frac{\sigma_T}{\gamma_T}(b^W\boldsymbol{W}_n + b^H\boldsymbol{H}_n)}_{=\boldsymbol{\Psi}_h(\varrho_n, \boldsymbol{X}_n, \boldsymbol{W}_\varrho)}.
\end{aligned}
\tag{183}
$$

To further simplify let $\frac{\sigma_T}{\gamma_T}\hat{\boldsymbol{Z}}_i = \boldsymbol{Z}_i$, then we construct the reversible scheme with forward pass:

$$
\begin{aligned}
\boldsymbol{X}_{n+1} &= \frac{\sigma_{n+1}\gamma_n}{\gamma_{n+1}\sigma_n}(\zeta \boldsymbol{X}_n + (1-\zeta)\hat{\boldsymbol{X}}_n) + \frac{\sigma_{n+1}}{\gamma_{n+1}}\boldsymbol{\Psi}_h(\varrho_n, \hat{\boldsymbol{X}}_n, \boldsymbol{W}_\varrho(\omega)), \\
\hat{\boldsymbol{X}}_{n+1} &= \frac{\sigma_{n+1}\gamma_n}{\gamma_{n+1}\sigma_n}\hat{\boldsymbol{X}}_n - \frac{\sigma_{n+1}}{\gamma_{n+1}}\boldsymbol{\Psi}_{-h}(\varrho_{n+1}, \boldsymbol{X}_{n+1}, \boldsymbol{W}_\varrho(\omega)),
\end{aligned}
\tag{184}
$$

and backward pass

$$\hat{\boldsymbol{X}}_n = \frac{\sigma_n \gamma_{n+1}}{\gamma_n \sigma_{n+1}} \hat{\boldsymbol{X}}_{n+1} + \frac{\sigma_n}{\gamma_n} \boldsymbol{\Psi}_{-h}(\varrho_{n+1}, \boldsymbol{X}_{n+1}, \boldsymbol{W}_{\varrho}(\omega)),$$

$$\boldsymbol{X}_n = \frac{\sigma_n \gamma_{n+1}}{\gamma_n \sigma_{n+1}} \zeta^{-1} \boldsymbol{X}_{n+1} + (1 - \zeta^{-1}) \hat{\boldsymbol{X}}_n - \frac{\sigma_n}{\gamma_n} \zeta^{-1} \boldsymbol{\Psi}_h(\varrho_n, \hat{\boldsymbol{X}}_n, \boldsymbol{W}_{\varrho}(\omega)),$$

(185)

with step size $h \coloneqq \varrho_{n+1} - \varrho_n$ and where

$$\hat{\boldsymbol{Z}}_i = \frac{\gamma_n}{\sigma_n} \boldsymbol{X}_n + h \sum_{j=1}^{i-1} \left[ a_{ij} \boldsymbol{x}^{\theta}_{0|\varrho_n + c_j h} \left( \frac{\sigma_{\varrho_n + c_j h}}{\gamma_{\varrho_n + c_j h}} \hat{\boldsymbol{Z}}_j \right) \right] + a_i^W \boldsymbol{W}_n + a_i^H \boldsymbol{H}_n,$$

$$\boldsymbol{\Psi}_h(\varrho_n, \boldsymbol{X}_n, \boldsymbol{W}_{\varrho}(\omega)) = h \sum_{j=1}^{s} \left[ b_i \boldsymbol{x}^{\theta}_{0|\varrho_n + c_i h} \left( \frac{\sigma_{\varrho_n + c_i h}}{\gamma_{\varrho_n + c_i h}} \hat{\boldsymbol{Z}}_i \right) \right] + b^W \boldsymbol{W}_n + b^H \boldsymbol{H}_n.$$

(186)

$\square$

**Noise Prediction.** We present this derivation in the form of Lemma C.7 below.

**Lemma C.7** (*Rex* (SDE) for noise prediction models)**.** *Let $\boldsymbol{\Phi}$ be an explicit stochastic Runge-Kutta solver for the additive noise SDE in Equation* (158)*, we construct the following reversible scheme for diffusion models*

$$\boldsymbol{X}_{n+1} = \frac{\alpha_{n+1}}{\alpha_n} (\zeta \boldsymbol{X}_n + (1 - \zeta) \hat{\boldsymbol{X}}_n) + \alpha_{n+1} \boldsymbol{\Psi}_h(\chi_n, \hat{\boldsymbol{X}}_n, \boldsymbol{W}_{\chi^2}(\omega)),$$

$$\hat{\boldsymbol{X}}_{n+1} = \frac{\alpha_{n+1}}{\alpha_n} \hat{\boldsymbol{X}}_n - \alpha_{n+1} \boldsymbol{\Psi}_{-h}(\chi_{n+1}, \boldsymbol{X}_{n+1}, \boldsymbol{W}_{\chi^2}(\omega)),$$

(187)

*and the backward step is given as*

$$\hat{\boldsymbol{X}}_n = \frac{\alpha_n}{\alpha_{n+1}} \hat{\boldsymbol{X}}_{n+1} + \alpha_n \boldsymbol{\Psi}_{-h}(\chi_{n+1}, \boldsymbol{X}_{n+1}, \boldsymbol{W}_{\chi^2}(\omega)),$$

$$\boldsymbol{X}_n = \frac{\alpha_n}{\alpha_{n+1}} \zeta^{-1} \boldsymbol{X}_{n+1} + (1 - \zeta^{-1}) \hat{\boldsymbol{X}}_n - \alpha_n \zeta^{-1} \boldsymbol{\Psi}_h(\chi_n, \hat{\boldsymbol{X}}_n, \boldsymbol{W}_{\chi^2}(\omega)),$$

(188)

*with step size $h \coloneqq \chi_{n+1} - \chi_n$ and where $\boldsymbol{\Psi}$ denotes the following scheme*

$$\hat{\boldsymbol{Z}}_i = \frac{1}{\alpha_n} \boldsymbol{X}_n + h \sum_{j=1}^{i-1} \left[ 2a_{ij} \boldsymbol{x}^{\theta}_{T|\chi_n + c_j h} \left( \alpha_{\chi_n + c_j h} \hat{\boldsymbol{Z}}_j \right) \right] + a_i^W \boldsymbol{W}_n + a_i^H \boldsymbol{H}_n,$$

$$\boldsymbol{\Psi}_h(\chi_n, \boldsymbol{X}_n, \boldsymbol{W}_{\chi}(\omega)) = h \sum_{i=1}^{s} \left[ 2b_i \boldsymbol{x}^{\theta}_{T|\chi_n + c_i h} \left( \alpha_{\chi_n + c_i h} \hat{\boldsymbol{Z}}_i \right) \right] + b^W \boldsymbol{W}_n + b^H \boldsymbol{H}_n.$$

(189)

*Proof.* We write the SRK scheme for the time-changed reverse-time SDE in Equation (158) to construct the following SRK scheme

$$\boldsymbol{Z}_i = \boldsymbol{Y}_n + h \sum_{j=1}^{i-1} \left[ 2a_{ij} \alpha_T \boldsymbol{x}_{T|\chi_n + c_j h} \left( \frac{\alpha_{\chi_n + c_j h}}{\alpha_T} \boldsymbol{Z}_j \right) \right] + \alpha_T (a_i^W \boldsymbol{W}_n + a_i^H \boldsymbol{H}_n),$$

$$\boldsymbol{Y}_{n+1} = \boldsymbol{Y}_n + h \sum_{i=1}^{s} \left[ 2b_i \alpha_T \boldsymbol{x}_{T|\chi_n + c_i h} \left( \frac{\alpha_{\chi_n + c_i h}}{\alpha_T} \boldsymbol{Z}_i \right) \right] + \alpha_T (b^W \boldsymbol{W}_n + b^H \boldsymbol{H}_n),$$

(190)

with step size $h = \chi_{n+1} - \chi_n$. Next, we replace $\boldsymbol{Y}_t$ back with $\boldsymbol{X}_t$ which yields

$$
\begin{aligned}
\boldsymbol{Z}_i &= \alpha_T \left( \frac{1}{\alpha_n} \boldsymbol{X}_n + h \sum_{j=1}^{i-1} \left[ 2a_{ij} \boldsymbol{x}_{T|\chi_n + c_j h} \left( \frac{\alpha_{\chi_n + c_j h}}{\alpha_T} \boldsymbol{Z}_j \right) \right] \right) \\
&\quad + \alpha_T (a_i^W \boldsymbol{W}_n + a_i^H \boldsymbol{H}_n), \\
\frac{\alpha_{n+1}}{\alpha_T} \boldsymbol{X}_{n+1} &= \frac{\alpha_T}{\alpha_n} \boldsymbol{X}_n \\
&\quad + \underbrace{\alpha_T h \sum_{i=1}^{s} \left[ 2b_i \alpha_T \boldsymbol{x}_{T|\chi_n + c_i h} \left( \frac{\alpha_{\chi_n + c_i h}}{\alpha_T} \boldsymbol{Z}_i \right) \right] + \alpha_T (b^W \boldsymbol{W}_n + b^H \boldsymbol{H}_n)}_{= \boldsymbol{\Psi}_h(\chi_n, \boldsymbol{X}_n, \boldsymbol{W}_\chi)}.
\end{aligned}
\tag{191}
$$

To further simplify let $\alpha_T \hat{\boldsymbol{Z}}_i = \boldsymbol{Z}_i$, then we construct the reversible scheme with forward pass:

$$
\begin{aligned}
\boldsymbol{X}_{n+1} &= \frac{\alpha_{n+1}}{\alpha_n} (\zeta \boldsymbol{X}_n + (1 - \zeta) \hat{\boldsymbol{X}}_n) + \alpha_{n+1} \boldsymbol{\Psi}_h(\chi_n, \hat{\boldsymbol{X}}_n, \boldsymbol{W}_\chi(\omega)), \\
\hat{\boldsymbol{X}}_{n+1} &= \frac{\alpha_{n+1}}{\alpha_n} \hat{\boldsymbol{X}}_n - \alpha_{n+1} \boldsymbol{\Psi}_{-h}(\chi_{n+1}, \boldsymbol{X}_{n+1}, \boldsymbol{W}_\chi(\omega)),
\end{aligned}
\tag{192}
$$

and backward pass

$$
\begin{aligned}
\hat{\boldsymbol{X}}_n &= \frac{\alpha_n}{\alpha_{n+1}} \hat{\boldsymbol{X}}_{n+1} + \alpha_n \boldsymbol{\Psi}_{-h}(\chi_{n+1}, \boldsymbol{X}_{n+1}, \boldsymbol{W}_\chi(\omega)), \\
\boldsymbol{X}_n &= \frac{\alpha_n}{\alpha_{n+1}} \zeta^{-1} \boldsymbol{X}_{n+1} + (1 - \zeta^{-1}) \hat{\boldsymbol{X}}_n - \alpha_n \zeta^{-1} \boldsymbol{\Psi}_h(\chi_n, \hat{\boldsymbol{X}}_n, \boldsymbol{W}_\chi(\omega)),
\end{aligned}
\tag{193}
$$

with step size $h := \chi_{n+1} - \chi_n$ and where

$$
\begin{aligned}
\hat{\boldsymbol{Z}}_i &= \frac{1}{\alpha_n} \boldsymbol{X}_n + h \sum_{j=1}^{i-1} \left[ 2a_{ij} \boldsymbol{x}_{T|\chi_n + c_j h}^\theta \left( \alpha_{\chi_n + c_j h} \hat{\boldsymbol{Z}}_j \right) \right] + a_i^W \boldsymbol{W}_n + a_i^H \boldsymbol{H}_n, \\
\boldsymbol{\Psi}_h(\chi_n, \boldsymbol{X}_n, \boldsymbol{W}_\chi(\omega)) &= h \sum_{i=1}^{s} \left[ 2b_i \boldsymbol{x}_{T|\chi_n + c_i h}^\theta \left( \alpha_{\chi_n + c_i h} \hat{\boldsymbol{Z}}_i \right) \right] + b^W \boldsymbol{W}_n + b^H \boldsymbol{H}_n.
\end{aligned}
\tag{194}
$$

*N.B.*, $\boldsymbol{W}_n = \overline{\boldsymbol{W}}_{\chi_{n+1}^2} - \overline{\boldsymbol{W}}_{\chi_n^2}$. $\qquad\qquad\square$

### C.3. Proof of Proposition 3.2

We now can construct *Rex*.

**Proposition C.8** (*Rex*)**.** *Without loss of generality let $\boldsymbol{\Phi}$ denote an explicit SRK scheme for the SDE in Equation* (140) *with extended Butcher tableau $a_{ij}, b_i, c_i, a_i^W, a_i^H, b^W, b^H$. Fix an $\omega \in \Omega$ and let $\boldsymbol{W}$ be the Brownian motion over time variable $\varsigma$. Then the reversible solver constructed from $\boldsymbol{\Phi}$ in terms of the underlying state variable $\boldsymbol{X}_t$ is given by the forward step*

$$
\begin{aligned}
\boldsymbol{X}_{n+1} &= \frac{w_{n+1}}{w_n}\left(\zeta \boldsymbol{X}_n + (1-\zeta)\hat{\boldsymbol{X}}_n\right) + w_{n+1}\boldsymbol{\Psi}_h(\varsigma_n, \hat{\boldsymbol{X}}_n, \boldsymbol{W}_n(\omega)), \\
\hat{\boldsymbol{X}}_{n+1} &= \frac{w_{n+1}}{w_n}\hat{\boldsymbol{X}}_n - w_{n+1}\boldsymbol{\Psi}_{-h}(\varsigma_{n+1}, \boldsymbol{X}_{n+1}, \boldsymbol{W}_n(\omega)),
\end{aligned}
\tag{195}
$$

*and backward step*

$$
\begin{aligned}
\hat{\boldsymbol{X}}_n &= \frac{w_n}{w_{n+1}}\hat{\boldsymbol{X}}_{n+1} + w_n\boldsymbol{\Psi}_{-h}(\varsigma_{n+1}, \boldsymbol{X}_{n+1}, \boldsymbol{W}_n(\omega)), \\
\boldsymbol{X}_n &= \frac{w_n}{w_{n+1}}\zeta^{-1}\boldsymbol{X}_{n+1} + (1-\zeta^{-1})\hat{\boldsymbol{X}}_n - w_n\zeta^{-1}\boldsymbol{\Psi}_h(\varsigma_n, \hat{\boldsymbol{X}}_n, \boldsymbol{W}_n(\omega)),
\end{aligned}
\tag{196}
$$

*with step size $h := \varsigma_{n+1} - \varsigma_n$ and where $\boldsymbol{\Psi}$ denotes the following scheme*

$$
\begin{aligned}
\hat{\boldsymbol{Z}}_i &= \frac{1}{w_n}\boldsymbol{X}_n + h\sum_{j=1}^{i-1}\left[a_{ij}\boldsymbol{f}_\theta\left(\varsigma_n + c_j h, w_{\varsigma_n+c_j h}\hat{\boldsymbol{Z}}_j\right)\right] + a_i^W \boldsymbol{W}_n(\omega) + a_i^H \boldsymbol{H}_n(\omega), \\
\boldsymbol{\Psi}_h(\varsigma_n, \boldsymbol{X}_n, \boldsymbol{W}_n(\omega)) &= h\sum_{i=1}^{s}\left[b_i\boldsymbol{f}_\theta\left(\varsigma_n + c_i h, w_{\varsigma_n+c_i h}\hat{\boldsymbol{Z}}_i\right)\right] + b^W \boldsymbol{W}_n(\omega) + b^H \boldsymbol{H}_n(\omega),
\end{aligned}
\tag{197}
$$

*where $\boldsymbol{f}_\theta$ denotes the data prediction model, $w_n = \frac{\sigma_n}{\gamma_n}$ and $\varsigma_t = \varrho_t$. The ODE case is recovered for an explicit RK scheme $\boldsymbol{\Phi}$ for the ODE in Equation* (103) *with $w_n = \sigma_n$ and $\varsigma_t = \gamma_t$. For noise prediction models we have $\boldsymbol{f}_\theta$ denoting the noise prediction model with $w_n = \alpha_n$ and $\varsigma_t = \frac{\sigma_n}{\alpha_n}$.*

*Proof.* This follows by Lemmas C.2, C.3, C.6 and C.7 *mutatis mutandis*. □

# D. Convergence Order Proofs

## D.1. Assumptions

Beyond the general regularity conditions imposed on the learned diffusion model itself (see Lu et al., 2022b; Blasingame & Liu, 2024a; 2025) we also assert that in the noise prediction setting that $\alpha_T > 0$. In practice most commonly used diffusion noise schedules like the linear or scaled linear schedule satisfy this, (see Appendix F.1; *cf*. Lin et al., 2024).

## D.2. Proof of Theorem 3.3

**Theorem 3.3** (*Rex* is a $k$-th order solver)**.** *Let $\boldsymbol{\Phi}$ be a $k$-th order explicit Runge-Kutta scheme for the reparameterized probability flow ODE in Equation* (87) *with variance preserving noise schedule $(\alpha_t, \sigma_t)$. Then Rex constructed from $\boldsymbol{\Phi}$ is a $k$-th order solver, i.e., given the reversible solution $\{\boldsymbol{x}_n, \hat{\boldsymbol{x}}_n\}_{n=1}^{N}$ and true solution $\boldsymbol{x}_{t_n}$ we have*

$$
\|\boldsymbol{x}_n - \boldsymbol{x}_{t_n}\| \le Ch^k,
\tag{14}
$$

*for constants $C, h_{max} > 0$ and for step sizes $h \in [0, h_{max}]$.*

*Proof.* We will prove this for both the data prediction and noise prediction formulations.

**Data prediction.** By Theorem A.1 we have that reversible $\boldsymbol{\Phi}$ is a $k$-th order solver, and thus

$$
\|\boldsymbol{y}_n - \boldsymbol{y}_{t_n}\| \le Ch^k.
\tag{198}
$$

We use the change of variables from Equation (103) to find

$$\left\| \frac{\sigma_T}{\sigma_n} \boldsymbol{x}_n - \frac{\sigma_T}{\sigma_n} \boldsymbol{x}_{t_n} \right\| \leq Ch^k, \tag{199}$$

which simplifies to

$$\| \boldsymbol{x}_n - \boldsymbol{x}_{t_n} \| \leq \frac{\sigma_n}{\sigma_T} Ch^k. \tag{200}$$

Now by definition for variance preserving type diffusion SDEs we have that $\sigma_t \leq 1$ for all $t$. Thus we can write

$$\| \boldsymbol{x}_n - \boldsymbol{x}_{t_n} \| \leq C_1 h^k, \tag{201}$$

where $C_1 = \frac{C}{\sigma_T}$.

**Noise prediction.** By Theorem A.1 we have that reversible $\boldsymbol{\Phi}$ is a $k$-th order solver, and thus

$$\| \boldsymbol{y}_n - \boldsymbol{y}_{t_n} \| \leq Ch^k. \tag{202}$$

We use the change of variables from Equation (102) to find

$$\left\| \frac{\alpha_T}{\alpha_n} \boldsymbol{x}_n - \frac{\alpha_T}{\alpha_n} \boldsymbol{x}_{t_n} \right\| \leq Ch^k, \tag{203}$$

which simplifies to

$$\| \boldsymbol{x}_n - \boldsymbol{x}_{t_n} \| \leq \frac{\alpha_n}{\alpha_T} Ch^k. \tag{204}$$

Now by definition we have $\alpha_t \leq 1$ for all $t$ and we assume that $\alpha_T > 0$. Thus we can write

$$\| \boldsymbol{x}_n - \boldsymbol{x}_{t_n} \| \leq C_1 h^k, \tag{205}$$

where $C_1 = \frac{C}{\sigma_T}$. $\qquad\square$

## D.3. Proof of Theorem 3.4

**Definition D.1** (Strong order of convergence). Suppose an SDE solver admits a numerical solution $\boldsymbol{X}_n$ and we have a true solution $\boldsymbol{X}_{t_n}$. If

$$\sup_{0 \leq n \leq N} \mathbb{E} \| \boldsymbol{X}_n - \boldsymbol{X}_{t_n} \|^2 \leq Ch^{2\alpha}, \tag{206}$$

where $C > 0$ is a constant and $h$ is the step size, then the SDE solver strongly converges with order $\alpha$.

> **Theorem 3.4** (Convergence order for *Princeps*). *Let $\boldsymbol{\Phi}$ be a SRK scheme with strong order of convergence $\xi > 0$ for the reparameterized reverse-time diffusion SDE in Equation (8) with variance preserving noise schedule $(\alpha_t, \sigma_t)$ and $\alpha_T > 0$. Then $\boldsymbol{\Psi}$ constructed from $\boldsymbol{\Phi}$ has strong order of convergence $\xi$.*

*Proof.* We will prove this for both the data prediction and noise prediction formulations.

**Data prediction.** By definition we have $\boldsymbol{\Phi}$ has strong order of convergence $\xi$ and thus,

$$\sup_{0 \leq n \leq N} \mathbb{E} \| \boldsymbol{Y}_n - \boldsymbol{Y}_{t_n} \|^2 \leq Ch^{2\xi}, \tag{207}$$

where $h = \frac{\sigma_{n+1}^2}{\alpha_{n+1}} - \frac{\sigma_n^2}{\alpha_n}$. We use the change of variables from Equation (140) to find

$$\sup_{0 \leq n \leq N} \mathbb{E} \left\| \frac{\sigma_T^2 \alpha_n}{\sigma_n^2 \alpha_T} \boldsymbol{X}_n - \frac{\sigma_T^2 \alpha_n}{\sigma_n^2 \alpha_T} \boldsymbol{X}_{t_n} \right\|^2 \leq Ch^{2\xi}, \tag{208}$$

which simplifies to

$$\sup_{0 \leq n \leq N} \mathbb{E} \| \boldsymbol{X}_n - \boldsymbol{X}_{t_n} \|^2 \leq \frac{\sigma_n \sqrt{\alpha_T}}{\sigma_T \sqrt{\alpha_n}} Ch^{2\xi}. \tag{209}$$

Since by definition of $\alpha_n$ is a monotonically decreasing function, $\sigma_n$ is a monotonically increasing function, $\alpha_T > 0$, and $\sigma_T \leq 1$ we can write

$$\sup_{0 \leq n \leq N} \mathbb{E}\|\boldsymbol{X}_n - \boldsymbol{X}_{t_n}\|^2 \leq Ch^{2\xi}, \tag{210}$$

as

$$\frac{\sigma_n \sqrt{\alpha_T}}{\sigma_T \sqrt{\alpha_n}} \leq 1. \tag{211}$$

**Noise prediction.** By definition we have $\boldsymbol{\Phi}$ has strong order of convergence $\xi$ and thus,

$$\sup_{0 \leq n \leq N} \mathbb{E}\|\boldsymbol{Y}_n - \boldsymbol{Y}_{t_n}\|^2 \leq Ch^{2\xi}, \tag{212}$$

where $h = \frac{\sigma_{n+1}}{\alpha_{n+1}} - \frac{\sigma_n}{\alpha_n}$. We use the change of variables from Equation (158) to find

$$\sup_{0 \leq n \leq N} \mathbb{E}\left\|\frac{\alpha_n}{\alpha_T}\boldsymbol{X}_n - \frac{\alpha_n}{\alpha_T}\boldsymbol{X}_{t_n}\right\|^2 \leq Ch^{2\xi}, \tag{213}$$

which simplifies to

$$\sup_{0 \leq n \leq N} \mathbb{E}\|\boldsymbol{X}_n - \boldsymbol{X}_{t_n}\|^2 \leq \frac{\sqrt{\alpha_T}}{\sqrt{\alpha_n}}Ch^{2\xi}. \tag{214}$$

Since by definition of $\alpha_n$ is a monotonically decreasing function strictly less than 1 and $\alpha_T > 0$ we can write

$$\sup_{0 \leq n \leq N} \mathbb{E}\|\boldsymbol{X}_n - \boldsymbol{X}_{t_n}\|^2 \leq Ch^{2\xi}. \tag{215}$$

$\square$

# E. Relation to Other Solvers for Diffusion Models

While this paper primarily focused on *Rex* and the family of reversible solvers created by it, we wish to discuss the relation between the underlying scheme $\boldsymbol{\Psi}$ constructed from our method and other existing solvers for diffusion models. The solvers considered here are all *non-reversible*; for a discussion of prior *reversible* solvers for diffusion models (EDICT, BDIA, O-BELM, CycleDiffusion) see Appendix A.3.

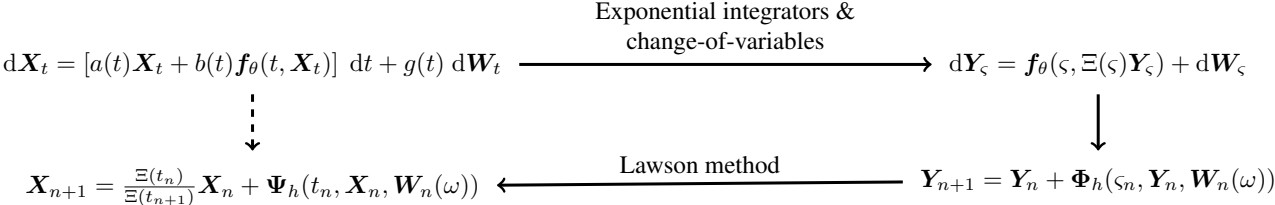

*Figure 10.* Overview of the construction of $\boldsymbol{\Psi}$ for the probability flow ODE from an underlying RK scheme $\boldsymbol{\Phi}$ for the reparameterized ODE. This graph holds for the SDE case *mutatis mutandis*.

Surprisingly, we discover that using Lawson methods outlined in Figure 10 (*cf*. Figure 3 from the main paper) is a surprisingly generalized methodology for construing numerical schemes for diffusion modes, and that it subsumes previous works. This means that several of the reversible schemes we presented here are reversible variants of well known schemes in the literature in diffusion models.

**Theorem 3.5** (*Princeps* subsumes previous solvers). *Princeps subsumes the following solvers for diffusion models*
1. *DDIM (Song et al., 2021a),*
2. *DPM-Solver-1, DPM-Solver-2, DPM-Solver-12 (Lu et al., 2022b),*
3. *DPM-Solver++1, DPM-Solver++(2S), SDE-DPM-Solver-1, SDE-DPM-Solver++1 (Lu et al., 2022a),*
4. *SEEDS-1 (Gonzalez et al., 2024), and*
5. *gDDIM (Zhang et al., 2023).*

*Proof.* We prove the connection to each solver in the list within a set of separate propositions for easier readability. The statement holds true via Propositions E.1 to E.8 and Corollaries E.1.1 to E.6.1. □

**Corollary E.0.1.** *Rex is the reversible version of the well-known solvers for diffusion models in Theorem 3.5.*

**Remark E.1.** The SDE solvers constructed from Foster-Reis-Strange SRK schemes are wholly unique (with the exception of the trivial Euler-Maruyama scheme) and have no existing counterpart in the literature in diffusion models. Thus *Rex* (ShARK) is not only a novel reversible solver, but a novel solver for diffusion models in general.

### E.1. Rex as Reversible ODE Solvers

Here we discuss *Rex* as reversible versions for well-known numerical schemes for diffusion models. Recall that the general Butcher tableau for a $s$-stage explicit RK scheme (Stewart, 2022, Section 6.1.4) is written as

$$
\begin{array}{c|ccccc}
c_1 & \\
c_2 & a_{21} \\
c_3 & a_{31} & a_{32} \\
\vdots & \vdots & \vdots & \ddots \\
c_s & a_{s1} & a_{s2} & \cdots & a_{(s-1)s} \\
\hline
& b_1 & b_2 & \cdots & b_{s-1} & b_s
\end{array}
= 
\begin{array}{c|c}
c & a \\
\hline
& b
\end{array}.
\tag{216}
$$

Embedded methods for adaptive step sizing are of the form

$$
\begin{array}{c|ccccc}
c_1 & \\
c_2 & a_{21} \\
c_3 & a_{31} & a_{32} \\
\vdots & \vdots & \vdots & \ddots \\
c_s & a_{s1} & a_{s2} & \cdots & a_{(s-1)s} \\
\hline
& b_1 & b_2 & \cdots & b_{s-1} & b_s \\
& b_1^* & b_2^* & \cdots & b_{s-1}^* & b_s^*
\end{array},
\tag{217}
$$

where the lower-order step is given by the coefficients $b_i^*$.

#### E.1.1. EULER

In this section we explore the numerical schemes produced by choosing the Euler scheme for $\boldsymbol{\Phi}$. The Butcher tableau for the Euler method is

$$
\begin{array}{c|c}
0 & 0 \\
\hline
& 1
\end{array}.
\tag{218}
$$

**Proposition E.1** (*Rex* (Euler) is reversible DPM-Solver++1)**.** *The underlying scheme of Rex (Euler) for the data prediction parameterization of diffusion models in Equation* (103) *is the DPM-Solver++1 from Lu et al. (2022a).*

*Proof.* Apply in the Butcher tableau for the Euler scheme to $\boldsymbol{\Psi}$ constructed from Equation (102) to find

$$
\boldsymbol{x}_{n+1} = \frac{\sigma_{n+1}}{\sigma_n} \boldsymbol{x}_n + \sigma_{n+1} h \boldsymbol{x}_{0|\gamma_n}^{\theta}(\boldsymbol{x}_n),
\tag{219}
$$

with $h = \gamma_{n+1} - \gamma_n$. We can rewrite the step size as

$$\sigma_{n+1}h = \sigma_{n+1}\left(\frac{\alpha_{n+1}}{\sigma_{n+1}} - \frac{\alpha_n}{\sigma_n}\right), \tag{220}$$

$$= \left(\alpha_{n+1} - \alpha_n\frac{\sigma_{n+1}}{\sigma_n}\right), \tag{221}$$

$$= \left(\alpha_{n+1}\frac{\alpha_{n+1}}{\alpha_{n+1}} - \frac{\alpha_n}{\alpha_{n+1}}\frac{\sigma_{n+1}}{\sigma_n}\right), \tag{222}$$

$$= -\alpha_{n+1}\left(\frac{\alpha_n}{\alpha_{n+1}}\frac{\sigma_{n+1}}{\sigma_n} - 1\right), \tag{223}$$

$$= -\alpha_{n+1}\left(\frac{\gamma_n}{\gamma_{n+1}} - 1\right), \tag{224}$$

$$= -\alpha_{n+1}\left(e^{\log\frac{\gamma_n}{\gamma_{n+1}}} - 1\right), \tag{225}$$

$$= -\alpha_{n+1}\left(e^{\log\gamma_n - \log\gamma_{n+1}} - 1\right), \tag{226}$$

$$\overset{(i)}{=} -\alpha_{n+1}\left(e^{\lambda_n - \lambda_{n+1}} - 1\right), \tag{227}$$

$$\overset{(ii)}{=} -\alpha_{n+1}\left(e^{-h_\lambda} - 1\right), \tag{228}$$

where (i) holds by the letting $\lambda_t = \log\gamma_t$ following the notation of Lu et al. (2022b;a) and (ii) holds by letting $h_\lambda = \lambda_{n+1} - \lambda_n$. Plugging this back into Equation (219) yields

$$\boldsymbol{x}_{n+1} = \frac{\sigma_{n+1}}{\sigma_n}\boldsymbol{x}_n - \alpha_{n+1}\left(e^{-h_\lambda} - 1\right)\boldsymbol{x}_{0|t_n}^\theta(\boldsymbol{x}_n), \tag{229}$$

which is the DPM-Solver++1 from Lu et al. (2022a). □

**Corollary E.1.1** (*Rex* (Euler) is reversible deterministic DDIM for data prediction models). *The underlying scheme of Rex (Euler) for the data prediction parameterization of diffusion models in Equation (103) is the deterministic DDIM solver from Song et al. (2021a).*

*Proof.* This holds because DPM-Solver++1 is DDIM see Lu et al. (2022a, Equation (21)) with $\eta = 0$. □

**Proposition E.2** (*Rex* (Euler) is reversible DPM-Solver-1). *The underlying scheme of Rex (Euler) for the data prediction parameterization of diffusion models in Equation (102) is the DPM-Solver-1 from Lu et al. (2022b, Equation (3.7)).*

*Proof.* Apply in the Butcher tableau for the Euler scheme to $\boldsymbol{\Psi}$ from *Rex* (see Proposition 3.2) to find

$$\boldsymbol{x}_{n+1} = \frac{\alpha_{n+1}}{\alpha_n}\boldsymbol{x}_n + \alpha_{n+1}h\boldsymbol{x}_{T|\chi_n}^\theta(\boldsymbol{x}_n), \tag{230}$$

with $h = \chi_{n+1} - \chi_n$. We can rewrite step size as

$$\alpha_{n+1}h = \alpha_{n+1}\left(\frac{\sigma_{n+1}}{\alpha_{n+1}} - \frac{\sigma_n}{\alpha_n}\right), \tag{231}$$

$$= \left(\sigma_{n+1} - \sigma_n\frac{\alpha_{n+1}}{\alpha_n}\right), \tag{232}$$

$$= \left(\sigma_{n+1}\frac{\sigma_{n+1}}{\sigma_{n+1}} - \frac{\sigma_n}{\sigma_{n+1}}\frac{\alpha_{n+1}}{\alpha_n}\right), \tag{233}$$

$$= -\sigma_{n+1}\left(\frac{\sigma_n}{\sigma_{n+1}}\frac{\alpha_{n+1}}{\alpha_n} - 1\right), \tag{234}$$

$$= -\sigma_{n+1}\left(\frac{\chi_n}{\chi_{n+1}} - 1\right), \tag{235}$$

$$= -\sigma_{n+1}\left(e^{\log\frac{\chi_n}{\chi_{n+1}}} - 1\right), \tag{236}$$

$$= -\sigma_{n+1}\left(e^{\log\chi_n - \log\chi_{n+1}} - 1\right), \tag{237}$$

$$\overset{(i)}{=} -\sigma_{n+1}\left(e^{-\lambda_n + \lambda_{n+1}} - 1\right), \tag{238}$$

$$\overset{(ii)}{=} -\sigma_{n+1}\left(e^{h_\lambda} - 1\right), \tag{239}$$

where (i) holds by the letting $\lambda_t = \log\gamma_t = -\log\chi_t$ following the notation of Lu et al. (2022b;a) and (ii) holds by letting $h_\lambda = \lambda_{n+1} - \lambda_n$. Plugging this back into Equation (219) yields

$$\boldsymbol{x}_{n+1} = \frac{\alpha_{n+1}}{\alpha_n}\boldsymbol{x}_n - \sigma_{n+1}\left(e^{h_\lambda} - 1\right)\boldsymbol{x}^\theta_{T|t_n}(\boldsymbol{x}_n), \tag{240}$$

which is the DPM-Solver-1 from Lu et al. (2022b). $\qquad\square$

**Corollary E.2.1** (*Rex* (Euler) is reversible deterministic DDIM for noise prediction models)**.** *The underlying scheme of Rex (Euler) for the noise prediction parameterization of diffusion models in Equation* (102) *is the deterministic DDIM solver from Song et al. (2021a).*

*Proof.* This holds because DPM-Solver-1 is DDIM see Lu et al. (2022b, Equation (4.1)). $\qquad\square$

### E.1.2. SECOND-ORDER METHODS

In this section we explore the numerical schemes produced by choosing the explicit midpoint method for $\boldsymbol{\Phi}$. We can write a generic second-order method as

$$\begin{array}{c|cc}
0 & & \\
\eta & \eta & \\
\hline
& 1 - \frac{1}{2\eta} & \frac{1}{2\eta}
\end{array}, \tag{241}$$

for $\eta \neq 0$ (Butcher, 2016). The choice of $\eta = \frac{1}{2}$ yields the explicit midpoint, $\eta = \frac{2}{3}$ gives Ralston's second-order method, and $\eta = 1$ gives Heun's second-order method.

**Proposition E.3** (*Rex* (generic second-order) is reversible DPM-Solver++(2S))**.** *The underlying scheme of Rex (generic second-order) for the data prediction parameterization of diffusion models in Equation* (103) *is the DPM-Solver++(2S) from Lu et al. (2022a, Algorithm 1).*

*Proof.* The DPM-Solver++(2S) (Lu et al., 2022a, Algorithm 1) is defined as

$$\boldsymbol{u} = \frac{\sigma_p}{\sigma_n}\boldsymbol{x}_n - \alpha_p\left(e^{-r_\lambda h_\lambda} - 1\right)\boldsymbol{x}^\theta_{0|t_n}(\boldsymbol{x}_n),$$

$$\boldsymbol{D} = \left(1 - \frac{1}{2r_\lambda}\right)\boldsymbol{x}^\theta_{0|t_n}(\boldsymbol{x}_n) + \frac{1}{2r_\lambda}\boldsymbol{x}^\theta_{0|t_p}(\boldsymbol{u}), \tag{242}$$

$$\boldsymbol{x}_{n+1} = \frac{\sigma_{n+1}}{\sigma_n}\boldsymbol{x}_n - \alpha_{n+1}\left(e^{-h_\lambda} - 1\right)\boldsymbol{D},$$

for some intermediate timestep $t_n > t_p > t_{n+1}$ and with $r_\lambda = \frac{\lambda_p - \lambda_n}{\lambda_{n+1} - \lambda_n}$. Notice that $r_\lambda$ describes the location of the midpoint time in the $\lambda$-domain as a ratio, *i.e.*, we could say

$$\lambda_p = \lambda_n + r_\lambda h_\lambda, \tag{243}$$

where $r_\lambda \in (0, 1)$ denotes the interpolation point between the initial timestep $\lambda_n$ and terminal timestep $\lambda_{n+1}$. Thus we fix $\eta = r_\lambda$ as the step size ratio of the intermediate point.

Now we return to the underlying scheme of *Rex* applied to the generic second-order scheme, see Equation (241), Apply in the Butcher tableau for generic second-order scheme to $\mathbf{\Psi}$ constructed from Equation (102) to find

$$
\begin{aligned}
\mathbf{z} &= \frac{1}{\sigma_n} \mathbf{x}_n + \eta h \mathbf{x}^\theta_{0|\gamma_n}(\mathbf{x}_n), \\
\mathbf{x}_{n+1} &= \frac{\sigma_{n+1}}{\sigma_n} \mathbf{x}_n + \sigma_{n+1} h \left( \left(1 - \frac{1}{2\eta}\right) \mathbf{x}^\theta_{0|\gamma_n}(\mathbf{x}_n) + \frac{1}{2\eta} \mathbf{x}^\theta_{0|\gamma_n + \eta h}(\sigma_p \mathbf{z}) \right),
\end{aligned}
\tag{244}
$$

with $h = \gamma_{n+1} - \gamma_n$ and $\sigma_p = \sigma_{\gamma_n + \eta h}$ with $\gamma_p = \gamma_n + \eta h$. We can write

$$\sigma_p \mathbf{z} = \frac{\sigma_p}{\sigma_n} \mathbf{x}_n + \sigma_p \eta h \mathbf{x}^\theta_{0|\gamma_n}(\mathbf{x}_n). \tag{245}$$

Plugging this back into Equation (244) yields

$$
\begin{aligned}
\sigma_p \mathbf{z} &= \frac{\sigma_p}{\sigma_n} \mathbf{x}_n + \sigma_p \eta h \mathbf{x}^\theta_{0|\gamma_n}(\mathbf{x}_n), \\
\mathbf{x}_{n+1} &= \frac{\sigma_{n+1}}{\sigma_n} \mathbf{x}_n + \sigma_{n+1} h \underbrace{\left( \left(1 - \frac{1}{2\eta}\right) \mathbf{x}^\theta_{0|\gamma_n}(\mathbf{x}_n) + \frac{1}{2\eta} \mathbf{x}^\theta_{0|\gamma_n + \eta h}(\sigma_p \mathbf{z}) \right)}_{= \hat{\mathbf{D}}},
\end{aligned}
\tag{246}
$$

which is the DPM-Solver++1 from Lu et al. (2022a). Now recall from Proposition E.1 that

$$\sigma_{n+1} h = -\alpha_{n+1} \left( e^{-h_\lambda} - 1 \right), \tag{247}$$

it follows that

$$\sigma_p \eta h = -\alpha_p \left( e^{-r_\lambda h_\lambda} - 1 \right), \tag{248}$$

due to $\lambda_p - \lambda_n = r_\lambda h_\lambda$ and $\eta h = \lambda_p - \lambda_n$. Thus by letting $\sigma_p \mathbf{z} = \mathbf{u}$ and $\hat{\mathbf{D}} = \mathbf{D}$ we recover the DPM-Solver++(2S) solver. $\qquad\square$

**Proposition E.4** (*Rex* (generic second-order) is reversible DPM-Solver-2))**.** *The underlying scheme of Rex (generic second-order) for the noise prediction parameterization of diffusion models in Equation (102) is the DPM-Solver-2 from Lu et al. (2022b, Algorithm 4 cf. Algorithm 1).*

*Proof.* This follows as straightforward derivation from Proposition E.2 and Proposition E.3. $\qquad\square$

**Proposition E.5** (*Rex* (Euler-Midpoint) is DPM-Solver-12))**.** *The underlying scheme of Rex (Euler-Midpoint) for the noise prediction parameterization of diffusion models in Equation (102) is the DPM-Solver-12 from Lu et al. (2022b).*

*Proof.* The explicit midpoint method with embedded Euler method for adaptive step sizing is given by the Butcher tableau

$$
\begin{array}{c|cc}
0 & & \\
\frac{1}{2} & \frac{1}{2} & \\
\hline
 & 0 & 1 \\
 & 1 & 0
\end{array}
. \tag{249}
$$

From Proposition E.2 and Proposition E.4 we have shown that *Rex* (Euler) and *Rex* (Midpoint) correspond to DPM-Solver-1 and DPM-Solver-2 respectively. Thus the Butcher tableau above outlines DPM-Solver-12. $\qquad\square$

### E.1.3. THIRD-ORDER METHODS

For third-order solvers like DPM-Solver-3 (Lu et al., 2022b, Algorithm 5) our constructed scheme differs from solvers derived using ETD methods due to the presence of $\varphi_2$ terms where

$$\varphi_{k+1}(t) = \int_0^1 e^{(1-\delta)t} \frac{\delta^k}{k!} \, \mathrm{d}\delta, \tag{250}$$

this also reasoning extends to the DPM-Solver-4 from Gonzalez et al. (2024, Algorithm 7).

### E.2. Rex as Reversible SDE Solvers

In this section we discuss the connections between *Rex* and preexisting SDE solvers for diffusion models.

### E.2.1. EULER-MARUYAMA

The extended Butcher tableau for the Euler-Maruyama scheme is given by

$$
\begin{array}{c|ccc}
0 & 0 & 0 & 0 \\
\hline
 & 1 & 1 & 0
\end{array}. \tag{251}
$$

> **Proposition E.6** (*Rex* (Euler-Maruyama) is reversible SDE-DPM-Solver++1). *The underlying scheme of Rex (Euler-Maruyama) for the data prediction parameterization of diffusion models in Equation* (140) *is the SDE-DPM-Solver++1 from Lu et al. (2022a, Equation (18)).*

*Proof.* Apply in the Butcher tableau for the Euler-Maruyama scheme to $\mathbf{\Psi}$ constructed from Equation (158) to find

$$\boldsymbol{x}_{n+1} = \frac{\sigma_{n+1}^2 \alpha_n}{\sigma_n^2 \alpha_{n+1}} \boldsymbol{x}_n + \frac{\sigma_{n+1}^2}{\alpha_{n+1}} h \boldsymbol{x}_{0|\varrho_n}^\theta(\boldsymbol{x}_n) + \frac{\sigma_{n+1}^2}{\alpha_{n+1}} \boldsymbol{W}_n, \tag{252}$$

with $h = \varrho_{n+1} - \varrho_n$. We can rewrite the step size as

$$\frac{\sigma_{n+1}^2}{\alpha_{n+1}} h = \frac{\sigma_{n+1}^2}{\alpha_{n+1}} \left( \frac{\alpha_{n+1}^2}{\sigma_{n+1}^2} - \frac{\alpha_n^2}{\sigma_n^2} \right), \tag{253}$$

$$= \left( \alpha_{n+1} - \frac{\alpha_n^2}{\alpha_{n+1}} \frac{\sigma_{n+1}^2}{\sigma_n^2} \right), \tag{254}$$

$$= \alpha_{n+1} \left( 1 - \frac{\alpha_n^2}{\alpha_{n+1}^2} \frac{\sigma_{n+1}^2}{\sigma_n^2} \right), \tag{255}$$

$$= \alpha_{n+1} \left( 1 - \frac{\varrho_n}{\varrho_{n+1}} \right), \tag{256}$$

$$= \alpha_{n+1} \left( 1 - e^{2 \log \frac{\gamma_n}{\gamma_{n+1}}} \right), \tag{257}$$

$$= \alpha_{n+1} \left( 1 - e^{2 \log \gamma_n - 2 \log \gamma_{n+1}} \right), \tag{258}$$

$$\overset{(i)}{=} \alpha_{n+1} \left( 1 - e^{2\lambda_n - 2\lambda_{n+1}} \right), \tag{259}$$

$$\overset{(ii)}{=} \alpha_{n+1} \left( 1 - e^{-2h_\lambda} \right), \tag{260}$$

where (i) holds by the letting $\lambda_t = \log \gamma_t$ following the notation of Lu et al. (2022b;a) and (ii) holds by letting $h_\lambda = \lambda_{n+1} - \lambda_n$. Now recall that

$$\frac{\sigma_{n+1}^2 \alpha_n}{\sigma_n^2 \alpha_{n+1}} = \frac{\sigma_{n+1}}{\sigma_n} e^{-h_\lambda}. \tag{261}$$

Plugging these back into Equation (252) yields

$$\boldsymbol{x}_{n+1} = \frac{\sigma_{n+1}}{\sigma_n} e^{-h_\lambda} \boldsymbol{x}_n + \alpha_{n+1} \left( 1 - e^{-2h_\lambda} \right) \boldsymbol{x}_{0|t_n}^\theta(\boldsymbol{x}_n) + \frac{\sigma_{n+1}^2}{\alpha_n} \boldsymbol{W}_n. \tag{262}$$

Now recall that the Brownian increment $\boldsymbol{W}_n := \boldsymbol{W}_{\varrho_{n+1}} - \boldsymbol{W}_{\varrho_n}$ has variance $h$. Thus via the Itô isometry we can write

$$\boldsymbol{W}_n \sim \sqrt{h}\boldsymbol{\epsilon}, \tag{263}$$

with $\boldsymbol{\epsilon} \sim \mathcal{N}(\boldsymbol{0}, \boldsymbol{I})$. Then we have

$$\frac{\sigma_{n+1}^2}{\alpha_{n+1}}\sqrt{h} = \frac{\sigma_{n+1}^2}{\alpha_{n+1}}\sqrt{\frac{\alpha_{n+1}^2}{\sigma_{n+1}^2} - \frac{\alpha_n^2}{\sigma_n^2}}, \tag{264}$$

$$= \sqrt{\sigma_{n+1}^2 - \frac{\alpha_n^2}{\alpha_{n+1}^2}\frac{\sigma_{n+1}^4}{\sigma_n^2}}, \tag{265}$$

$$= \sigma_{n+1}\sqrt{1 - \frac{\alpha_n^2}{\alpha_{n+1}^2}\frac{\sigma_{n+1}^2}{\sigma_n^2}}, \tag{266}$$

$$= \sigma_{n+1}\sqrt{1 - \frac{\varrho_n}{\varrho_{n+1}}}, \tag{267}$$

$$= \sigma_{n+1}\sqrt{1 - e^{-2h_\lambda}}. \tag{268}$$

Thus we have re-derived the noise term of the SDE-DPM-Solver++1, and putting everything together we have obtained the SDE-DPM-Solver++1 from Lu et al. (2022a) which is

$$\boldsymbol{x}_{n+1} = \frac{\sigma_{n+1}}{\sigma_n}e^{-h_\lambda}\boldsymbol{x}_n + \alpha_{n+1}\left(1 - e^{-2h_\lambda}\right)\boldsymbol{x}_{0|t_n}^\theta(\boldsymbol{x}_n) + \sigma_{n+1}\sqrt{1 - e^{-2h_\lambda}}\boldsymbol{\epsilon}. \tag{269}$$

Thus we have shown that the SDE-DPM-Solver++1 is the same as the underlying scheme of *Rex* (Euler-Maruyama). □

**Corollary E.6.1** (*Rex* (Euler-Maruyama) is reversible stochastic DDIM). *The underlying scheme of Rex (Euler-Maruyama) for the data prediction parameterization of diffusion models in Equation* (140) *is the stochastic DDIM solver from Song et al. (2021a) with* $\eta = \sigma_t\sqrt{1 - e^{-2h_\lambda}}$.

*Proof.* This holds because SDE-DPM-Solver-1 is DDIM see Lu et al. (2022a, Section 6.1). □

**Proposition E.7** (*Rex* (Euler-Maruyama) is reversible SDE-DPM-Solver-1). *The underlying scheme of Rex (Euler-Maruyama) for the noise prediction parameterization of diffusion models in Equation* (158) *is the SDE-DPM-Solver-1 from Lu et al. (2022a, Equation (17)).*

*Proof.* Apply in the Butcher tableau for the Euler scheme to $\boldsymbol{\Psi}$ from *Rex* (see Proposition 3.2) to find

$$\boldsymbol{x}_{n+1} = \frac{\alpha_{n+1}}{\alpha_n}\boldsymbol{x}_n + 2\alpha_{n+1}h\boldsymbol{x}_{T|\chi_n}^\theta(\boldsymbol{x}_n) + \alpha_{n+1}\boldsymbol{W}_n, \tag{270}$$

with $h = \chi_{n+1} - \chi_n$. Recall from Proposition E.2 that we can rewrite the step size

$$\alpha_{n+1}h = -\sigma_{n+1}\left(e^{h_\lambda} - 1\right). \tag{271}$$

Now recall that the Brownian increment $\boldsymbol{W}_n := \overline{\boldsymbol{W}}_{\chi_{n+1}^2} - \overline{\boldsymbol{W}}_{\chi_n^2}$ has variance $\chi_n^2 - \chi_{n+1}^2$.[20] Thus via the Itô isometry we can write

$$\boldsymbol{W}_n \sim \sqrt{\chi_n^2 - \chi_{n+1}^2}\boldsymbol{\epsilon}, \tag{272}$$

---

[20]This is because $\overline{\boldsymbol{W}}_\chi^2$ is defined in reverse-time.

with $\epsilon \sim \mathcal{N}(\mathbf{0}, \boldsymbol{I})$. Then we have

$$\alpha_{n+1}\sqrt{\chi_n^2 - \chi_{n+1}^2} = \alpha_{n+1}\sqrt{\frac{\sigma_n^2}{\alpha_n^2} - \frac{\sigma_{n+1}^2}{\alpha_{n+1}^2}}, \tag{273}$$

$$= \sqrt{\frac{\sigma_n^2\alpha_{n+1}^2}{\alpha_n^2} - \sigma_{n+1}^2}, \tag{274}$$

$$= \sigma_{n+1}\sqrt{\frac{\sigma_n^2\alpha_{n+1}^2}{\sigma_{n+1}^2\alpha_n^2} - 1}, \tag{275}$$

$$= \sigma_{n+1}\sqrt{\frac{\chi_n^2}{\chi_{n+1}^2} - 1}, \tag{276}$$

$$= \sigma_{n+1}\sqrt{e^{\log\frac{\chi_n^2}{\chi_{n+1}^2}} - 1}, \tag{277}$$

$$= \sigma_{n+1}\sqrt{e^{\log\chi_n^2 - \log\chi_{n+1}^2} - 1}, \tag{278}$$

$$= \sigma_{n+1}\sqrt{e^{-2\log\gamma_n + 2\log\gamma_{n+1}} - 1}, \tag{279}$$

$$= \sigma_{n+1}\sqrt{e^{2\log\lambda_{n+1} - 2\log\lambda_n} - 1}, \tag{280}$$

$$= \sigma_{n+1}\sqrt{e^{2h_\lambda} - 1}. \tag{281}$$

Plugging Equations (271) and (281) back into Equation (270) yields

$$\boldsymbol{x}_{n+1} = \frac{\alpha_{n+1}}{\alpha_n}\boldsymbol{x}_n - 2\sigma_{n+1}\left(e^{h_\lambda} - 1\right)\boldsymbol{x}_{T|\chi_n}^\theta(\boldsymbol{x}_n) + \sigma_{n+1}\sqrt{e^{2h_\lambda} - 1}\boldsymbol{\epsilon}, \tag{282}$$

which is the SDE-DPM-Solver-1 from Lu et al. (2022a). □

**Corollary E.7.1** (*Rex* (Euler-Maruyama) is reversible stochastic DDIM for noise prediction models). *The underlying scheme of Rex (Euler-Maruyama) for the noise prediction parameterization of diffusion models in Equation* (158) *is the stochastic DDIM solver from Song et al. (2021a) with* $\eta = \sigma_t\sqrt{e^{-2h_\lambda} - 1}$.

*Proof.* This follows from a straightforwardly from Corollary E.6.1 and Lu et al. (2022b, Equation (4.1)). □

### E.3. Rex as Reversible SEEDS-1

> **Proposition E.8** (*Rex* is reversible SEEDS-1). *The choice of Euler or Euler-Maruyama for the underlying scheme of Rex with either the noise prediction parameterization of diffusion models in Equations* (102) *and* (158) *or data prediction in Equations* (102) *and* (140) *yields the four variants of SEEDS-1 outlined in Gonzalez et al. (2024, Equations (28-31)).*

*Proof.* This follows straightforwardly from Propositions E.1, E.2, E.6 and E.7 by definition of SEEDS-1. □

**Corollary E.8.1** (*Rex* (Euler-Maruyama) is reversible gDDIM). *The underlying scheme of Rex (Euler-Maruyama) for the data prediction parameterization of diffusion models in Equation* (140) *is the gDDIM solver in Zhang et al. (2023, Theorem 1) for* $\ell = 1$.

*Proof.* This follows as an immediate consequence of Proposition E.8 since by Gonzalez et al. (2024, Proposition 4.5) gDDIM is SEEDS-1. □

As mentioned earlier in Appendix A.4.1 high-order variants of SEEDS use a Markov-preserving noise decomposition to approximate the iterated stochastic integrals. However, we follow Foster et al. (2024) and use the space-time Lévy area resulting in numerical schemes that are quite different beyond the first-order case, albeit that *Rex* exhibits better convergence properties.

# F. Implementation Details

## F.1. Closed Form Expressions of the Noise Schedule

### F.1.1. VARIANCE PRESERVING SDEs

In practice, popular libraries like the `diffusers` library define the noise schedule for diffusion models as a discrete schedule $\{\beta_n\}_{n=1}^N$ following Ho et al. (2020); Song et al. (2021a) as an arithemetic sequence of the form

$$\beta_n = \frac{\beta_0}{N} + \frac{n-1}{N(N-1)}(\beta_1 - \beta_0), \tag{283}$$

with hyperparameters $\beta_0, \beta_1 \in \mathbb{R}_{\geq 0}$. Song et al. (2021b) defines the continuous-time schedule as

$$\beta_t = \beta_0 + t(\beta_1 - \beta_0), \tag{284}$$

for all $t \in [0, 1]$ in the limit of $N \to \infty$. Thus one can write the forward-time diffusion (variance preserving) SDE as

$$d\boldsymbol{X}_t = -\frac{1}{2}\beta_t \boldsymbol{X}_t \, dt + \sqrt{\beta_t} \, d\boldsymbol{W}_t. \tag{285}$$

Thus we can express the noise schedule $(\alpha_t, \sigma_t)$ as

$$\alpha_t = \exp\left(-\frac{1}{2}\int \beta_t \, dt\right), \tag{286a}$$

$$\sigma_t = \sqrt{1 - \alpha_t^2}. \tag{286b}$$

*N.B.*, often the hyperparmeters in libraries like `diffusers` are expressed as $\hat{\beta}_0 = \frac{\beta_0}{N}$ and $\hat{\beta}_1 = \frac{\beta_1}{N}$, often with $N = 1000$.

**Linear Noise Schedule.** For the linear noise schedule in Equation (284) used by DDPMs (Ho et al., 2020), the schedule $(\alpha_t, \sigma_t)$ is written as

$$\alpha_t = \exp\left(-\frac{\beta_1 - \beta_0}{4}t^2 - \frac{\beta_0}{2}t\right),$$
$$\sigma_t = \sqrt{1 - \alpha_t^2}, \tag{287}$$

for $t \in [0, 1]$ with hyperparameters $\beta_0$ and $\beta_1$.

> **Proposition F.1** (Inverse function of $\gamma_t$ for linear noise schedule). *For the linear noise schedule used by DDPMs (Ho et al., 2020) the inverse function of $\gamma_t$ denoted $t_\gamma$ can be expressed in closed form as*
>
> $$t_\gamma(\gamma) = \frac{-\beta_0 + \sqrt{\beta_0^2 + 2(\beta_1 - \beta_0)\log(\gamma^{-2} + 1)}}{\beta_1 - \beta_0}. \tag{288}$$

*Proof.* Let $\alpha_t$ be denoted by $\alpha_t = e^{a_t}$ where

$$a_t = -\frac{\beta_1 - \beta_0}{4}t^2 - \frac{\beta_0}{2}t. \tag{289}$$

Then by definition of $\gamma_t$ we can write

$$\gamma_t = \frac{e^{a_t}}{\sqrt{1 - e^{2a_t}}}, \tag{290}$$

and with a little more algebra we find

$$\sqrt{1 - e^{2a_t}} = \frac{e^{a_t}}{\gamma_t}, \tag{291}$$

$$1 - e^{2a_t} = \frac{e^{2a_t}}{\gamma_t^2}, \tag{292}$$

$$e^{-2a_t} - 1 = \gamma_t^{-2}, \tag{293}$$

$$e^{-2a_t} = \gamma_t^{-2} + 1, \tag{294}$$

$$-2a_t = \log(\gamma_t^{-2} + 1). \tag{295}$$

Then by substituting in the definition of $a_t$ and letting $\gamma$ denote the variable produced by $\gamma_t$ we have

$$\frac{\beta_1 - \beta_0}{2} t^2 + \beta_0 t - \log(\gamma^{-2} + 1) = 0. \tag{296}$$

We then use the quadratic formula to find the roots of the polynomial of $t$ to find

$$t = \frac{-\beta_0 \pm \sqrt{\beta_0^2 + 2(\beta_1 - \beta_0) \log(\gamma^{-2} + 1)}}{\beta_1 - \beta_0}. \tag{297}$$

Since $t \in [0, 1]$ we only take the positive root and thus

$$t = \frac{-\beta_0 + \sqrt{\beta_0^2 + 2(\beta_1 - \beta_0) \log(\gamma^{-2} + 1)}}{\beta_1 - \beta_0}. \tag{298}$$

$\square$

**Corollary F.1.1** (Inverse function of $\chi_t$ for linear noise schedule). *It follows by a straightforward substitution from Proposition F.1 that $t_\chi$ can be written as*

$$t_\chi(\chi) = \frac{-\beta_0 + \sqrt{\beta_0^2 + 2(\beta_1 - \beta_0) \log(\chi^2 + 1)}}{\beta_1 - \beta_0}. \tag{299}$$

**Corollary F.1.2** (Inverse function of $\varrho_t$ for linear noise schedule). *It follows by a straightforward substitution from Proposition F.1 that $t_\varrho$ can be written as*

$$t_\varrho(\varrho) = \frac{-\beta_0 + \sqrt{\beta_0^2 + 2(\beta_1 - \beta_0) \log(\varrho^{-1} + 1)}}{\beta_1 - \beta_0}. \tag{300}$$

**Scaled Linear Schedule.** The *scaled linear schedule* is used widely by *latent diffusion models* (LDMs) (Rombach et al., 2022) and takes the discrete form of

$$\beta_n = \left( \sqrt{\hat{\beta}_0} + \frac{n-1}{N-1} \left( \sqrt{\hat{\beta}_1} - \sqrt{\hat{\beta}_0} \right) \right)^2. \tag{301}$$

Thus following a similar approach to Song et al. (2021b) we write the scaled linear schedule as a function of $t$,

$$\beta_t = (\beta_1 - 2\sqrt{\beta_1 \beta_0} + \beta_0)t^2 + 2t(\sqrt{\beta_1 \beta_0} - \beta_0) + \beta_0. \tag{302}$$

Then using Equation (286) we find the noise schedule $(\alpha_t, \sigma_t)$ to be defined as

$$\alpha_t = \exp\left( -\frac{\beta_1 - 2\sqrt{\beta_1 \beta_0} + \beta_0}{6} t^3 - \frac{\sqrt{\beta_1 \beta_0} - \beta_0}{2} t^2 - \frac{\beta_0}{2} t \right),$$
$$\sigma_t = \sqrt{1 - \alpha_t^2}. \tag{303}$$

Next we will derive the inverse function for $\gamma_t$

**Proposition F.2** (Inverse function of $\gamma_t$ for scaled linear noise schedule). *For the scaled linear noise schedule commonly used by LDMs (Rombach et al., 2022) the inverse function of $\gamma_t$ denoted $t_\gamma$ can be expressed in closed form as*

$$t_\gamma(\gamma) = \frac{\beta_0 - \sqrt{\beta_1\beta_0} - \sqrt[3]{2(\sqrt{\beta_1\beta_0} - \beta_0)^3 - 3\beta_0\Delta(\sqrt{\beta_1\beta_0} - \beta_0) - 3\Delta^2 \log(\gamma^{-2} + 1)}}{\Delta}, \tag{304}$$

*where*

$$\Delta = \beta_1 - 2\sqrt{\beta_1\beta_0} + \beta_0. \tag{305}$$

*Proof.* Let $\alpha_t$ be denoted by $\alpha_t = e^{a_t}$ where

$$a_t = -\frac{\beta_1 - 2\sqrt{\beta_1\beta_0} + \beta_0}{6}t^3 - \frac{\sqrt{\beta_1\beta_0} - \beta_0}{2}t^2 - \frac{\beta_0}{2}t. \tag{306}$$

Then by definition of $\gamma_t$ we can write

$$\gamma_t = \frac{e^{a_t}}{\sqrt{1 - e^{2a_t}}}, \tag{307}$$

and with a little more algebra we find

$$\sqrt{1 - e^{2a_t}} = \frac{e^{a_t}}{\gamma_t}, \tag{308}$$

$$1 - e^{2a_t} = \frac{e^{2a_t}}{\gamma_t^2}, \tag{309}$$

$$e^{-2a_t} - 1 = \gamma_t^{-2}, \tag{310}$$

$$e^{-2a_t} = \gamma_t^{-2} + 1, \tag{311}$$

$$-2a_t = \log(\gamma_t^{-2} + 1). \tag{312}$$

Then by substituting in the definition of $a_t$ and letting $\gamma$ denote the variable produced by $\gamma_t$ we have

$$\frac{\beta_1 - 2\sqrt{\beta_1\beta_0} + \beta_0}{3}t^3 + (\sqrt{\beta_1\beta_0} - \beta_0)t^2 + \beta_0 t - \log(\gamma^{-2} + 1) = 0. \tag{313}$$

We then use the cubic formula (Cardano, 1545) to find the roots of the polynomial of $t$. The only real root is given by

$$t_\gamma(\gamma) = \frac{\beta_0 - \sqrt{\beta_1\beta_0} - \sqrt[3]{2(\sqrt{\beta_1\beta_0} - \beta_0)^3 - 3\beta_0\Delta(\sqrt{\beta_1\beta_0} - \beta_0) - 3\Delta^2 \log(\gamma^{-2} + 1)}}{\Delta}, \tag{314}$$

where

$$\Delta = \beta_1 - 2\sqrt{\beta_1\beta_0} + \beta_0. \tag{315}$$

$\square$

**Corollary F.2.1** (Inverse function of $\chi_t$ for scaled linear noise schedule). *It follows by a straightforward substitution from Proposition F.2 that $t_\chi$ can be written as*

$$t_\chi(\chi) = \frac{\beta_0 - \sqrt{\beta_1\beta_0} - \sqrt[3]{2(\sqrt{\beta_1\beta_0} - \beta_0)^3 - 3\beta_0\Delta(\sqrt{\beta_1\beta_0} - \beta_0) - 3\Delta^2 \log(\chi^2 + 1)}}{\Delta}, \tag{316}$$

*where*

$$\Delta = \beta_1 - 2\sqrt{\beta_1\beta_0} + \beta_0. \tag{317}$$

**Corollary F.2.2** (Inverse function of $\varrho_t$ for scaled linear noise schedule). *It follows by a straightforward substitution from Proposition F.2 that $t_\varrho$ can be written as*

$$t_\varrho(\varrho) = \frac{\beta_0 - \sqrt{\beta_1\beta_0} - \sqrt[3]{2(\sqrt{\beta_1\beta_0} - \beta_0)^3 - 3\beta_0\Delta(\sqrt{\beta_1\beta_0} - \beta_0) - 3\Delta^2 \log(\varrho^{-1} + 1)}}{\Delta}, \tag{318}$$

*where*

$$\Delta = \beta_1 - 2\sqrt{\beta_1\beta_0} + \beta_0. \tag{319}$$

### F.1.2. OT FLOW MATCHING

Within OT flow matching (Tong et al., 2024) framework these expressions become much simpler.[21] Within this framework we have

$$\alpha_t = t, \tag{320a}$$

$$\sigma_t = 1 - t. \tag{320b}$$

Consequently, we have the following simple proposition.

**Proposition F.3** (Inverse function of $\gamma_t$ in OT flow matching). *Within the OT flow matching context the inverse function of $\gamma_t$ denoted $t_\gamma$ can be expressed in closed form as*

$$t_\gamma(\gamma) = \frac{\gamma}{1 + \gamma}. \tag{321}$$

*Proof.* By definition of $\gamma_t$ we write

$$\gamma_t = \frac{t}{1 - t}, \tag{322}$$

$$(1 - t)\gamma_t = t, \tag{323}$$

$$\gamma_t = (1 + \gamma_t)t, \tag{324}$$

$$t = \frac{\gamma_t}{1 + \gamma_t}. \tag{325}$$

$\square$

**Corollary F.3.1** (Inverse function of $\chi_t$ in OT flow matching). *It follows by a straightforward substitution from Proposition F.3 that $t_\chi$ can be written as*

$$t_\chi(\chi) = \frac{1}{1 + \chi}. \tag{326}$$

### F.2. Some Other Inverse Functions

$\gamma \mapsto \sigma$. Additionally, we need to be able to extract the weighting terms from the time integration variable. For the ODE case we need the function $\sigma_\gamma(\gamma)$ which describes the map $\gamma \mapsto \sigma$. By the definition of $\gamma$ we have

$$\gamma = \frac{\alpha}{\sigma}, \tag{327}$$

$$\gamma \overset{(i)}{=} \frac{\sqrt{1 - \sigma^2}}{\sigma}, \tag{328}$$

$$\sigma\gamma = \sqrt{1 - \sigma^2}, \tag{329}$$

$$\sigma^2\gamma^2 = 1 - \sigma^2, \tag{330}$$

$$\sigma^2\gamma^2 = 1 - \sigma^2, \tag{331}$$

$$\gamma^2 = \sigma^{-2} - 1, \tag{332}$$

$$\gamma^2 + 1 = \sigma^{-2}, \tag{333}$$

$$\sigma^2 = \frac{1}{\gamma^2 + 1} \tag{334}$$

$$\sigma_\gamma(\gamma) = \frac{1}{\sqrt{\gamma^2 + 1}}, \tag{335}$$

where (i) hold by $\sigma^2 = 1 - \alpha^2$ for VP type diffusion SDEs.

---

[21]We will use the flow matching conventions where $\boldsymbol{X}_1 \sim p_{\text{data}}$.

$\varrho \mapsto \frac{\sigma}{\gamma}$. Likewise, for the SDE case we need the function which maps $\varrho \mapsto \frac{\sigma}{\gamma}$. Recall that (note we drop the subscript $t$ for the derivation)

$$\varrho = \frac{\alpha^2}{\sigma^2}, \tag{336}$$

thus we have

$$\varrho \overset{(i)}{=} \frac{\alpha^2}{1 - \alpha^2}, \tag{337}$$

$$(1 - \alpha^2)\varrho = \alpha^2, \tag{338}$$

$$\alpha^{-2} - 1 = \varrho^{-1}, \tag{339}$$

$$\alpha^{-2} = \varrho^{-1} + 1, \tag{340}$$

$$\alpha = \frac{1}{\sqrt{\varrho^{-1} + 1}}, \tag{341}$$

where (i) hold by $\sigma^2 = 1 - \alpha^2$ for VP type diffusion SDEs. Then we can write

$$\frac{\sigma}{\gamma} = \frac{\sigma^2}{\alpha}, \tag{342}$$

$$= \frac{\sigma^2}{\alpha}\frac{\alpha}{\alpha}, \tag{343}$$

$$= \frac{\sigma^2}{\alpha^2}\alpha, \tag{344}$$

$$= \varrho^{-1}\alpha, \tag{345}$$

$$= \frac{1}{\rho\sqrt{\rho^{-1} + 1}}. \tag{346}$$

$\chi \mapsto \alpha$.  Lastly, for the noise prediction models we need the map $\chi \mapsto \alpha$ denoted $\alpha_\chi(\chi)$. By definition of $\chi$ we have

$$\chi = \frac{\sigma}{\alpha}, \tag{347}$$

$$\chi \overset{(i)}{=} \frac{\sqrt{1 - \alpha^2}}{\alpha}, \tag{348}$$

$$\alpha_\chi(\chi) \overset{(ii)}{=} \frac{1}{\sqrt{\chi^2 + 1}}, \tag{349}$$

where (i) hold by $\sigma^2 = 1 - \alpha^2$ for VP type diffusion SDEs and (ii) holds by the derivation for $\sigma_\gamma(\gamma)$ *mutatis mutandis*.

### F.3. Numerical Simulation of Brownian Motion

Earlier we mentioned that for reversible methods we need to be able to compute both the *same* realization of the Brownian motion. Now sampling Brownian motion is quite simple—recall Lévy's characterization of Brownian motion (Øksendal, 2003, Theorem 8.6.1)—and can be sampled by drawing independent Gaussian increments during the numerical solve of an SDE. A common choice for an adaptive solver is to use Lévy's Brownian bridge formula (Revuz & Yor, 2013).

**Definition F.1** (Lévy's Brownian bridge).  Given the standard $d_w$-dimensional Brownian motion $\{W_t : t \geq 0\}$ and for any $0 \leq s < t < u$, the Brownian bridge is defined as

$$W_t | W_s, W_u \sim \mathcal{N}\left(W_s + \frac{t - s}{u - s}(W_u - W_s), \frac{(u - t)(t - s)}{u - s}I\right), \tag{350}$$

and this quantity is conditionally independent of $W_v$ for $v < s$ or $v > u$.

Sampling the Brownian motion in reverse-time, however, is more complicated as it is only adapted to the natural filtration defined in forward time. The naïve approach to sampling Brownian motion, called the *Brownian path*, is to simply store the entire realization of the Brownian motion from the forward pass in memory and use Equation (350) when necessary (for adaptive step size methods). This results in a query time of $\mathcal{O}(1)$, but with a memory cost of $\mathcal{O}(nd_w)$, where $n$ is the number of samples.

### F.3.1. METHODS

**Virtual Brownian Tree.**    Seminal work on neural SDEs by Li et al. (2020) introduced the *Virtual Brownian Tree* which extends the concept of Brownian trees introduced by Gaines & Lyons (1997). The Brownian tree recursively applies Equation (350) to sample the Brownian motion at any midpoint, constructing a tree structure; however, storing such a tree would be memory intensive. By making use of splittable *pseudo-random number generators* PRNGs (Salmon et al., 2011; Claessen & Pałka, 2013) which can deterministically generate two random seeds given an existing seed. Then making use of a splittable PRNG one can evaluate the Brownian motion at any point by recursively applying the Brownian tree constructing to rebuild the tree until the recursive midpoint time $t_r$ is suitable *close* to the desired timestep $t$, *i.e.*, $|t - t_r| < \epsilon$ for some fixed error threshold $\epsilon > 0$. This requires constant $\mathcal{O}(1)$ memory but takes $\mathcal{O}(\log(1/\epsilon))$ time and is only *approximate*.

**Brownian Interval.**    Closely related work by Kidger et al. (2021) introduces the *Brownian Interval* which offers exact sampling with $\mathcal{O}(1)$ query times. The primary difference between this method and Virtual Brownian Trees is that this method focuses on intervals rather than particular sample points. To elucidate, let $\boldsymbol{W}_{s,t} = \boldsymbol{W}_t - \boldsymbol{W}_s$ denote an interval of Brownian motion. Then the formula for Lévy's Brownian bridge (350) can be rewritten in terms of Brownian intervals as

$$\boldsymbol{W}_{s,t}|\boldsymbol{W}_{s,u} \sim \mathcal{N}\left(\frac{t-s}{u-s}\boldsymbol{W}_{s,u}, \frac{(u-t)(s-u)}{u-s}\boldsymbol{I}\right). \tag{351}$$

Then, the method constructs a tree with stump being the global interval $[0, T]$ and a random seed for a splittable PRNG. New leaf nodes are constructed when queries over intervals are made; this provides the advantage of the tree being query-dependent unlike the Virtual Brownian Tree which has a fixed dyadic structure. Further computational improvements are made to improve implementation with the details being found in Kidger (2022, Section 5.5.3). Beyond the numerical efficiency in computing intervals over points is that we regularly need use intervals in numeric schemes and not single sample points. Often, solvers which approximate higher-order integrals (*e.g.*, stochastic Runge-Kutta) require samples of the Lévy area[22] which would require the Brownian interval to construct.[23]

**Updated Virtual Brownian Tree.**    Recent work by Jelinčič et al. (2024) improves upon the Virtual Brownian Tree (Li et al., 2020) by using an interpolation strategy between query points.[24] This enables the updated algorithm to exactly match the distribution of Brownian motion and Lévy areas at all query times as long as each query time is at least $\epsilon$ apart.

### F.3.2. IMPLEMENTATION

We used the Brownian interval (Kidger et al., 2021) provided by the `torchsde` library. In general we would recommend the virtual Brownian tree from Jelinčič et al. (2024) over the Brownian interval, an implementation of this can be found in the `diffrax` library. However, as our code base made extensive used of prior projects developed in pytorch and `diffrax` is a jax library it made more sense to use `torchsde` for this project.

## G. Experimental Details

We provide additional details for the empirical studies conducted in Section 4. *N.B.*, for all experiments we used fixed random seeds between the different software components to ensure a fair comparision. For all image experiments we built our experimental code from the `zituitui/BELM` repository which contains the implementation of BELM (Wang et al., 2024) and reimplementations of EDICT (Wallace et al., 2023) and BDIA (Zhang et al., 2024). The rest of this section is devoted to particular experimental details.

---

[22]*I.e.*, for a $d_w$-dimensional Brownian motion over $[s, t]$ the Lévy area is

$$2\boldsymbol{L}_{s,t}^{i,j} := \int_s^t \boldsymbol{W}_{s,u}^i \mathrm{d}\boldsymbol{W}_u^j - \int_s^t \boldsymbol{W}_{s,u}^j \mathrm{d}\boldsymbol{W}_u^i.$$

[23]The interested reader can find more details in James Foster's thesis (Foster, 2020).

[24]This algorithm is a part of the popular `Diffrax` library.

## G.1. Unconditional Image Generation

### G.1.1. DIFFUSION MODEL

We make use of a pre-trained DDPM (Ho et al., 2020) model trained on the CelebA-HQ $256 \times 256$ dataset (Karras et al., 2018). The linear noise schedule from (Ho et al., 2020) is given as

$$\beta_i = \frac{\hat{\beta}_0}{T} + \frac{i-1}{T(T-1)}(\hat{\beta}_1 - \hat{\beta}_0). \tag{352}$$

We convert this into a continuous time representation via the details in Appendix F.1 following Song et al. (2021b). For this experiment we used $\hat{\beta}_0 = 0.0001$ and $\hat{\beta}_1 = 0.2$. To ensure numerical stability due to $\frac{1}{\sigma_t}$ terms we solve the probability flow ODE in reverse-time on the time interval $[\epsilon, 1]$ with $\epsilon = 0.0002$. This is a common choice to make in practice see Song et al. (2023).

### G.1.2. METRICS

We use several metrics to assess the performance in unconditional image generation following Stein et al. (2023) by using a DINOv2 feature extractor (Oquab et al., 2023), all of which are calculated using the 10k generated samples and 30k real samples from the CelebA-HQ dataset. Throughout this section we will let $\{x_i\}_{i=1}^n$ denote an empirical distribution drawn from our generated distribution $\mathbb{P}_\theta$ and let $\{\hat{x}_i\}_{i=1}^m$ denote an empirical distribution drawn from the data distribution $\mathbb{P}_{data}$.

**FD.** The *Fréchet distance* (FD) (Dowson & Landau, 1982) is measured using the sample mean and covariance of the real $\mathbb{P}_{data}$ and generated $\mathbb{P}_\theta$ distributions denoted

$$\text{FD}(\mathbb{P}_{data}\|\mathbb{P}_\theta) = \|\mu_{data} - \mu_\theta\|_2^2 + \text{Tr}\left(\Sigma_{data} + \Sigma_\theta - 2(\Sigma_{data}\Sigma_\theta)^{\frac{1}{2}}\right), \tag{353}$$

where $(\mu_\cdot, \Sigma_\cdot)$ denote the sample mean and covariances. This metric corresponds two the 2-Wasserstein distance between two multivariate Gaussians and is thus a valid metric between the first two moments. Heusel et al. (2017) popularized the use of this metric within the feature layer of an Inception-V3 network (Szegedy et al., 2016) to assess the fidelity of unconditional image generation, this metric is referred to as the *Fréchet inception distance* or FID. Recent works have challenged the use of the Inception-V3 network as the feature extractor (Stein et al., 2023; Jayasumana et al., 2024; Kynkäänniemi et al., 2023) showing that the Inception-V3 network is poorly suited for capturing a semantic view of images which correlates well to human judgment. In particular, Stein et al. (2023) shows that using DINOv2 (Oquab et al., 2023) for the feature extractor results in a metric which is significantly more aligned with human judgment.

**FD$_\infty$.** FD$_\infty$ proposed by Chong & Forsyth (2020) is a modification of FD which aims to remove the inherent bias induced by using a finite number of empirical samples. The samples is determined by evaluating FD over 15 regular intervals over the number of total samples and fitting a linear trend to the 15 data points to infer a trend for FD as the number of empirical samples, $N \to \infty$.

**Precision, Recall, Density and Coverage.** The density metric (Naeem et al., 2020) is used as a proxy to measure sample fidelity and improves upon the earlier precision metric (Kynkäänniemi et al., 2019; Sajjadi et al., 2018). The metric is based upon nearest neighbours distance computed in a representation space and counts how many real-sample neighbourhood balls contain the generated sample. Likewise to quantify sample diversity we use the coverage metric (Naeem et al., 2020) which improves upon the earlier recall metric (Kynkäänniemi et al., 2019; Sajjadi et al., 2018). The density metric is given by

$$\text{density}(\mathbb{P}_{data}, \mathbb{P}_\theta) = \frac{1}{kn}\sum_{i=1}^n\sum_{j=1}^m \mathbb{1}_{B(\hat{x}_j, \delta^k(\hat{x}_j))}(x_i), \tag{354}$$

where $\mathbb{1}_A(\cdot)$ denotes the indicator function for set $A$, $B(x, r)$ constructs a Euclidean ball centered at $x$ with radius $r$, and $\delta^k(\hat{x}_j)$ is the distance to the $k$-th nearest neighbour in $\{\hat{x}_i\}_{i=1}^m$, excluding itself. The precision metric is given by

$$\text{precision}(\mathbb{P}_{data}, \mathbb{P}_\theta) = \frac{1}{n}\sum_{i=1}^n \mathbb{1}_{\bigcup_{j=1}^m B(\hat{x}_j, \delta^k(\hat{x}_j))}(x_i). \tag{355}$$

Similarly, coverage is given by

$$\text{coverage}(\mathbb{P}_{data}, \mathbb{P}_\theta) = \frac{1}{m} \sum_{j=1}^{m} \max_{i=1,\dots,n} 1_{B(\hat{\boldsymbol{x}}_j, \delta^k(\hat{\boldsymbol{x}}_j))}(\boldsymbol{x}_i). \tag{356}$$

Likewise, the recall metric is given by

$$\text{recall}(\mathbb{P}_{data}, \mathbb{P}_\theta) = \frac{1}{m} \sum_{j=1}^{m} 1_{\bigcup_{i=1}^{n} B(\boldsymbol{x}_i, \delta^k(\boldsymbol{x}_i))}(\hat{\boldsymbol{x}}_j). \tag{357}$$

We used $k = 5$ and 10k samples throughtout, as standard.

**On Reporting.** When reporting on these metrics like in Table 5 we use **bold font** to denote the best performance with a 1% error range. More formally, suppose we have a series of $n$ data points $\{x_i\}_{i=1}^{n}$ that is totally ordered by some relation $R$. We say will denote a query point $x_i$ with **bold font** if the *range-normalized absolute percentage error* is less than $\epsilon > 0$, *i.e.*,

$$\frac{|\max_j x_j - x_i|}{\max_j x_j - \min_k x_k} < \epsilon. \tag{358}$$

In our experiments we report $\epsilon = 0.01$.

### G.1.3. HYPERPARAMETERS

We follow the suggestion of Wallace et al. (2023) and report results with EDICT using the hyperparameter $p = 0.93$. For BDIA, the original paper recommends $\gamma = 1.0$ for unconditional image generation (Zhang et al., 2024, Section 6.1). However, we found $\gamma = 0.5$ to yield better performance, this corroborates with the findings of Wang et al. (2024).

### G.2. Conditional Image Generation

#### G.2.1. DIFFUSION MODEL

We make use of Stable Diffusion v1.5 (Rombach et al., 2022) a pre-trained *latent diffusion model* (LDM) model. We also use the scaled linear noise schedule given as

$$\beta_i = \left( \sqrt{\frac{\hat{\beta}_0}{T}} + \frac{i - 1}{\sqrt{T}(T - 1)} \left( \sqrt{\hat{\beta}_1} - \sqrt{\hat{\beta}_0} \right) \right)^2. \tag{359}$$

We convert this into a continuous time representation via the details in Appendix F.1 following Song et al. (2021b). For this experiment we used $\hat{\beta}_0 = 0.00085$ and $\hat{\beta}_1 = 0.012$. To ensure numerical stability due to $\frac{1}{\sigma_t}$ terms we solve the probability flow ODE in reverse-time on the time interval $[\epsilon, 1]$ with $\epsilon = 0.0002$. This is a common choice to make in practice see Song et al. (2023).

**Numerical Schemes.** We set the last two steps of *Rex* schemes to be either Euler or Euler-Maruyama for better stability near time 0.

#### G.2.2. METRICS

As mentioned in the main paper we use the CLIP Score (Hessel et al., 2021) PickScore (Kirstain et al., 2023), and Image Reward metrics (Xu et al., 2023) to asses the ability of the text-to-image conditional generation task. We calculate each by comparing the sampled image and the given text prompt used to produce the image. We then report the average over the 1000 samples.

**CLIP Score.** The CLIP score measures the cosine similarity between the text and visual embeddings with pretrained CLIP model (Radford et al., 2021) denoted as

$$\text{CLIPScore}(\boldsymbol{x}, \boldsymbol{c}) = \max \left\{ \frac{\langle \mathcal{E}_I(\boldsymbol{x}), \mathcal{E}_C(\boldsymbol{c}) \rangle}{\|\mathcal{E}_I(\boldsymbol{x})\| \|\mathcal{E}_C(\boldsymbol{c})\|}, 0 \right\}, \tag{360}$$

where $\mathcal{E}_I : \mathbb{R}^d \to V$ is the image embedder and $\mathcal{E}_C : \mathbb{R}^{d'} \to V$ is the caption embedder; and where $x$ is the query image and $c$ is the query caption. Thus this metric aims to measure how well our generated images align with their prompt. In particular, we use the `ViT-L/14` backbone trained by OpenAI.

**PickScore.**   Similar to CLIP score, PickScore finetunes a CLIP-H model on their proposed Pick-a-Pic dataset which purportedly aligns better with human preference over CLIP score.

**Image Reward.**   Image Reward (Xu et al., 2023) is the newest of the three metrics and uses BLIP (Li et al., 2022) over CLIP as the backbone and finetunes the model using reward model training. The resulting metrics achieves state-of-the-art alignment with human preferences.

**On Reporting.**   When reporting on these metrics like in Table 6 we use **bold font** to denote the best performance with a 1% error range. In our experiments we report $\epsilon = 0.01$.

### G.2.3. HYPERPARAMETERS

We follow the suggestion of Wallace et al. (2023) and report results with EDICT using the hyperparameter $p = 0.93$. For BDIA, the original paper recommends $\gamma = 0.5$ for text-to-image generation (Zhang et al., 2024, Section 6.1). We also ran BDIA with $\gamma = 0.96$ as suggested by Wang et al. (2024).

### G.3. Image Editing

For our experiments we drew 100 text-image-instruction triples from the `InstructPix2Pix` dataset (Brooks et al., 2023) and report the mean of each metric over these. We use Stable Diffusion v1.5 with the same scaled-linear noise schedule and continuous-time conversion as in Appendix G.2. All fixed-step solvers use 100 inversion steps and 100 generation steps with CFG scale 3.0; *Rex* (Dopri5) uses the adaptive step controller of Dormand & Prince (1980) with atol = rtol = $10^{-5}$. The inversion is performed under the source caption $c_{\text{src}}$ to time $t = 0.6$, after which generation back to $t = 0$ is performed under the edit caption $c_{\text{edit}}$. We use the same EDICT and BDIA hyperparameters as in Appendix G.2.

### G.4. Interpolation

**Diffusion Model.**   We make use of a pre-trained DDPM (Ho et al., 2020) model trained on the CelebA-HQ $256 \times 256$ dataset (Karras et al., 2018). We used linear noise schedule from (Ho et al., 2020). We convert this into a continuous time representation via the details in Appendix F.1 following Song et al. (2021b). For this experiment we used $\hat{\beta}_0 = 0.0001$ and $\hat{\beta}_1 = 0.2$. For the face pairings we followed Blasingame & Liu (2024a;c) and used the FRLL (DeBruine & Jones, 2017) dataset.

Notably, we used the noise prediction parameterization rather than data prediction as we found that it performed better for editing. This is likely due to the singularity of the $\frac{1}{\sigma_t}$ terms as $t \to 0$. Within this parameterization we could use the time interval $[0, 1]$ instead of $[\epsilon, 1]$ like in previous experiments with data prediction models.

### G.5. Boltzmann Sampling

Our codebase is built upon the `transferable-samplers/transferable-samplers` repository (Tan et al., 2025b) wherein we add the *Rex* code.

### G.5.1. DATASETS

We follow the same training, validation, and test split used by Tan et al. (2025a) to evaluate *Rex* on equilibrium conformation sampling tasks, with a focus on tri-alanine. These datasets are obtained from implicit solvent molecular dynamics (MD) simulations. In particular, a single MCMC chain is decomposed into $10^5$, $2 \times 10^4$, and $10^4$ samples for training, validation, and testing. The training and validation data are each taken from contiguous regions of the chain to simulate the realistic scenario wherein a pre-existing MCMC trajectory exists and one would like to use a Boltzmann generator to continue generating samples. Earlier parts of the trajectory under-sampled specific modes enabling a biased training set which we aim to debias through access to the energy function and SNIS. All MD simulations were run for 1 μs with a timestep of 1 fs at temperatures of 300K and 310K for alanine dipeptide and tri-alanine mirroring those done by Klein & Noé (2025).

**Tri-alanine.** For the tri-alanine dataset, we follow the splitting procedure of Tan et al. (2025a). The first 100,000 datapoints after burn-in are used as the training set, the next 10,000 points in the (subsampled) chain are used for validation, and a random selection of the rest of the chain is used as test samples. This creates a biased training set relative to the test set, and is realistic in the setting where we would like to draw samples more efficiently from an existing MD chain.

### G.5.2. TRAINING DETAILS

**Architecture.** We adopt a DiT backbone (Peebles & Xie, 2023) with the details shown below in Table 3.

*Table 3.* Overview of architecture configurations.

| Parameter | DiT |
|---|---|
| Hidden size | 192 |
| Blocks | 6 |
| Heads | 6 |
| Conditional dimension | 64 |
| # of Parameters (M) | 3.1 M |

**Training Configuration.** All models were trained us an exponential moving average on the weights with a decay rate of 0.999. For evaluation we generated $10^4$ proposal samples and we used the same number of re-sampling and computing all metrics.

**Hyperparameters.** We used AdamW algorithm (Loshchilov & Hutter, 2017) to perform gradient descent with a learning rate of $5 \times 10^{-4}$, $\beta = (0.9, 0.999)$, $\epsilon = 10^{-8}$, and weight decay $10^{-4}$. A cosine annealing schedule was applied to the learning rate with a warm-up phase covering 5% of the training iterations. We trained for 3000 epochs.

### G.5.3. METRICS

We evaluate model performance using both sample-based metrics and metrics that assess energy distributions. We enumerate these in greater detail below.

**Effective Sample Size.** We compute the effective sample size (ESS) using Kish's formula (Kish, 1957), *i.e.*, given $N \in \mathbb{N}$ generated particles with unnormalized importance weights $\{w_i\}_{i=1}^N \subsetneq \mathbb{R}_{\geq 0}$ we have

$$ESS(\{w_i\}_i = 1^N) := \frac{1}{N} \frac{1}{\sum_{i=1}^N w_i^2} \left( \sum_{i=1}^N w_i \right)^2. \tag{361}$$

The ESS is a measure of how many independent and equally-weighted samples would provide equivalent statistical power to the weighted sample.

**2-Wasserstein Energy Distance ($\mathcal{E}$-$\mathcal{W}_2$).** To compare energy distributions we measure the 2-Wasserstein distance between them which for two probability measures $\mu, \nu$ on $\mathbb{R}$ over energy values is given as

$$\mathcal{E}\text{-}\mathcal{W}_2(\mu, \nu)^2 = \inf_{\gamma \in \Pi(\mu, \nu)} \int_{\mathbb{R} \times \mathbb{R}} |x - y|^2 \, \mathrm{d}\gamma(x, y), \tag{362}$$

where $\Pi(\mu, \nu)$ is the set of all couplings whose marginals are $\mu$ and $\nu$. The 2-Wasserstein distance is an integral probability metric which measures captures both the difference in *location* and *shape* of two distributions.

**Torus 2-Wasserstein Distance ($\mathbb{T}$-$\mathcal{W}_2$).** To measure structural similarity in torsional space, we compute the 2-Wasserstein distance over dihedral angles. For a molecule with $L \in \mathbb{N}$ residues, we define the dihedral vector as

$$\text{Dihedrals}(\boldsymbol{x}) = (\phi_1, \psi_1, \ldots, \phi_{L-1}, \psi_{L-1}) \in [0, 2\pi)^{2(L-1)}. \tag{363}$$

Thus given the torus geometry a natural cost function arises as the minimal signed angle difference, *i.e.*,

$$c_{\mathcal{T}}(\boldsymbol{x}, \boldsymbol{y})^2 = \sum_{i=1}^2 \left[ (\text{Dihedrals}(\boldsymbol{x})_i - \text{Dihedrals}(\boldsymbol{y})_i + \pi) \mod 2\pi - \pi \right]^2. \tag{364}$$

*Table 4.* Reconstruction error in pixel space and latent space (MSE) for Stable Diffusion v1.5 ($512 \times 512$) averaged over 100 real images, at varying number of steps and floating-point precision. A CFG scale of 1.0 was used. $^{\dagger}$ denotes runs with non-deterministic GPU operations enabled. Best results in **bold**, second best underlined.

| Precision | Solver | Steps = 10 | | Steps = 20 | | Steps = 50 | |
|---|---|---|---|---|---|---|---|
| | | **Pixel MSE** | **Latent MSE** | **Pixel MSE** | **Latent MSE** | **Pixel MSE** | **Latent MSE** |
| fp32 | DDIM | $2.81 \times 10^{-2}$ | $3.57 \times 10^{-1}$ | $1.05 \times 10^{-2}$ | $9.89 \times 10^{-2}$ | $3.36 \times 10^{-3}$ | $1.66 \times 10^{-2}$ |
| | EDICT | $\mathbf{1.86 \times 10^{-3}}$ | $1.59 \times 10^{-6}$ | $\mathbf{1.86 \times 10^{-3}}$ | $1.98 \times 10^{-7}$ | $\mathbf{1.86 \times 10^{-3}}$ | $2.23 \times 10^{-9}$ |
| | BDIA | $\mathbf{1.86 \times 10^{-3}}$ | $3.26 \times 10^{-7}$ | $\mathbf{1.86 \times 10^{-3}}$ | $1.29 \times 10^{-7}$ | $\mathbf{1.86 \times 10^{-3}}$ | $2.34 \times 10^{-8}$ |
| | O-BELM | $\mathbf{1.86 \times 10^{-3}}$ | $\underline{1.05 \times 10^{-7}}$ | $\mathbf{1.86 \times 10^{-3}}$ | $\underline{2.24 \times 10^{-7}}$ | $\mathbf{1.86 \times 10^{-3}}$ | $3.35 \times 10^{-7}$ |
| | *Rex* (Euler) | $\mathbf{1.86 \times 10^{-3}}$ | $\mathbf{3.77 \times 10^{-9}}$ | $\mathbf{1.86 \times 10^{-3}}$ | $\mathbf{1.98 \times 10^{-9}}$ | $\mathbf{1.86 \times 10^{-3}}$ | $\mathbf{8.85 \times 10^{-10}}$ |
| | *Rex* $^{\dagger}$ (Euler) | $1.86 \times 10^{-3}$ | $3.77 \times 10^{-9}$ | $1.86 \times 10^{-3}$ | $1.98 \times 10^{-9}$ | $1.86 \times 10^{-3}$ | $8.85 \times 10^{-10}$ |
| fp16 | DDIM | $2.82 \times 10^{-2}$ | $3.57 \times 10^{-1}$ | $1.05 \times 10^{-2}$ | $9.90 \times 10^{-2}$ | $3.36 \times 10^{-3}$ | $1.66 \times 10^{-2}$ |
| | EDICT | $1.96 \times 10^{-3}$ | $9.84 \times 10^{-4}$ | $\underline{1.90 \times 10^{-3}}$ | $3.94 \times 10^{-4}$ | $\mathbf{1.86 \times 10^{-3}}$ | $\mathbf{7.87 \times 10^{-6}}$ |
| | BDIA | $\underline{1.87 \times 10^{-3}}$ | $6.62 \times 10^{-5}$ | $\mathbf{1.87 \times 10^{-3}}$ | $6.80 \times 10^{-5}$ | $1.87 \times 10^{-3}$ | $6.80 \times 10^{-5}$ |
| | O-BELM | $\mathbf{1.86 \times 10^{-3}}$ | $9.03 \times 10^{-6}$ | $1.87 \times 10^{-3}$ | $\underline{8.16 \times 10^{-5}}$ | $1.93 \times 10^{-3}$ | $9.17 \times 10^{-4}$ |
| | *Rex* (Euler) | $\mathbf{1.86 \times 10^{-3}}$ | $\mathbf{1.63 \times 10^{-6}}$ | $1.87 \times 10^{-3}$ | $\mathbf{2.44 \times 10^{-6}}$ | $\underline{1.87 \times 10^{-3}}$ | $1.25 \times 10^{-5}$ |

Thus the torus 2-Wasserstein between two distributions $\mu, \nu$ on $[0, 2\pi)^{2(L-1)}$ is then

$$\mathbb{T}\text{-}\mathcal{W}_2(\mu, \nu)^2 = \inf_{\gamma \in \Pi(\mu,\nu)} \int_{\mathbb{R}^d \times \mathbb{R}^d} c_{\mathcal{T}}(\boldsymbol{x}, \boldsymbol{y})^2 \, \mathrm{d}\gamma(\boldsymbol{x}, \boldsymbol{y}). \tag{365}$$

## G.6. Hardware

All experiments were run using a single NVIDIA H100 80 GB GPU.

## G.7. Repositories

In our empirical studies we made use of the following resources and repositories:

1. `google/ddpm-celebahq-256` (DDPM Model)

2. `stable-diffusion-v1-5/stable-diffusion-v1-5` (Stable Diffusion v1.5)

3. `zituitui/BELM` (Implementation of BELM, EDICT, and BDIA)

4. `google-research/torchsde` (Brownian Interval)

5. `layer6ai-labs/dgm-eval` (FD, FD$_\infty$, KD, Density, and Coverage metrics)

6. `torchmetrics` (CLIP score)

7. `zai-org/ImageReward` (Image Reward)

8. `yuvalkirstain/pickscore` (PickScore)

9. `timbrooks/instructpix2pix-clip-filtered` (InstructPix2Pix dataset)

10. `transferable-samplers/transferable-samplers` (Boltzman sampling)

11. `transferable-samplers/many-peptides-md` (Tri-alanine data)

# H. Additional Results

This appendix collects results that could not fit into the main paper, including full quantitative tables for the radar charts and additional qualitative samples. We first report the full reconstruction-error study referenced in Section 4.1 (Appendix H.1), followed by the complete tables for unconditional generation (Appendix H.2), conditional generation (Appendix H.3), image editing (Appendix H.4), and Boltzmann sampling (Appendix H.5). We then turn to two further analyses of *Rex*: a profiling of

*Table 5.* Quantitative comparison of different reversible solvers for unconditional image generation with a pre-trained DDPM model on CelebA-HQ ($256 \times 256$) with the non-reversible DDIM as a baseline. Our reversible ODE solvers are in pink and reversible SDE solvers in mauve.

| Steps | Solver | FD ($\downarrow$) | FD$_\infty$ ($\downarrow$) | Precision ($\uparrow$) | Recall ($\uparrow$) | Density ($\uparrow$) | Coverage ($\uparrow$) |
|---|---|---|---|---|---|---|---|
| | EDICT | 1042.89 | 1034.82 | 0.49 | 0.10 | 0.19 | 0.11 |
| | BDIA | 900.95 | 894.23 | 0.61 | 0.10 | 0.28 | 0.14 |
| | O-BELM | **605.52** | **596.47** | 0.78 | 0.18 | 0.56 | 0.34 |
| | *Rex* (Midpoint) | **607.20** | **597.04** | 0.78 | 0.21 | 0.60 | **0.37** |
| 10 | *Rex* (RK4) | 633.90 | 617.11 | **0.81** | **0.22** | **0.64** | 0.36 |
| | *Rex* (Euler-Maruyama) | **610.16** | **598.56** | 0.79 | 0.10 | 0.61 | **0.37** |
| | DDIM | 727.75 | 716.41 | 0.75 | 0.14 | 0.49 | 0.27 |
| | EDICT | 752.68 | 743.89 | 0.68 | 0.15 | 0.36 | 0.21 |
| | BDIA | 611.47 | 601.37 | 0.76 | 0.19 | 0.50 | 0.30 |
| | O-BELM | 489.94 | 477.82 | 0.82 | 0.23 | 0.71 | 0.43 |
| | *Rex* (Midpoint) | 539.96 | 527.85 | 0.81 | 0.26 | 0.66 | 0.41 |
| 20 | *Rex* (RK4) | 547.24 | 533.30 | 0.82 | **0.27** | 0.71 | 0.43 |
| | *Rex* (Euler-Maruyama) | **460.42** | **447.01** | **0.86** | 0.21 | **0.91** | **0.51** |
| | DDIM | 570.11 | 555.26 | 0.79 | 0.20 | 0.62 | 0.38 |
| | EDICT | 551.13 | 534.73 | 0.78 | 0.24 | 0.60 | 0.37 |
| | BDIA | 500.79 | 489.24 | 0.82 | 0.27 | 0.70 | 0.44 |
| | O-BELM | 476.29 | 463.07 | 0.84 | **0.29** | 0.77 | 0.45 |
| | *Rex* (Midpoint) | 505.67 | 494.94 | 0.81 | **0.29** | 0.70 | 0.44 |
| 50 | *Rex* (RK4) | 511.17 | 498.94 | 0.80 | 0.27 | 0.69 | 0.44 |
| | *Rex* (Euler-Maruyama) | **391.93** | **381.01** | **0.87** | 0.28 | **0.98** | **0.56** |
| | DDIM | 490.88 | 479.87 | 0.80 | 0.26 | 0.67 | 0.45 |

the Brownian Interval overhead incurred by our SDE schemes (Appendix H.6) and an ablation of the three core ingredients of *Rex*—the reversible coupling, exponential transformation, and time reparameterization (Appendix H.7). Finally, we present qualitative material: a visualization of the inversion trajectory and the diffusion latent space (Appendix H.8), an interpolation study between real-image pairs (Appendix H.9), and a gallery of uncurated conditional generation samples (Appendix H.10).

## H.1. Reconstruction Error

This section reports the full reconstruction-error results referenced from Section 4.1 in the main paper. We measure the round-trip reconstruction error (forward solve followed by reverse solve) over 100 real images encoded with Stable Diffusion v1.5 at $512 \times 512$ resolution, evaluating at $10, 20, 50$ steps in both fp32 and fp16 with a CFG scale of $1.0$. We report the mean-squared error (MSE) in both pixel space and latent space; the latter isolates errors attributable to the numerical scheme by removing the VAE reconstruction error. For fair comparison we disabled non-deterministic GPU operations across all seed-matched runs; enabling them does not meaningfully change the inversion error of *Rex* (*cf*. the *Rex* $^\dagger$ row in Table 4).

Table 4 reports the full results. The fp32 latent-space MSEs are visualized in Figure 4 of the main paper. *Rex* (Euler) outperforms every baseline w.r.t. inversion error in nearly every tested setting, and often by a wide margin. In fp32, *Rex* achieves a latent-space MSE that is on average roughly three orders of magnitude smaller than O-BELM and eight orders of magnitude smaller than DDIM. The advantage is particularly pronounced in the few-step regime where EDICT fails (note EDICT is competitive with *Rex* at 50 fp32 steps, but degrades sharply at fewer steps). DDIM is not algebraically reversible, so its error simply reflects the truncation error of the underlying solver. EDICT and BDIA both improve with the number of steps but plateau in the $10^{-7}$–$10^{-9}$ range due to numerical instability of the coupling. O-BELM exhibits a more striking failure mode: the error *grows* with the number of steps, since the iteration is nowhere linearly stable and round-off error accumulates (see Appendix A.2). *Rex* sits closest to the floating-point round-off floor and improves monotonically with step count.

In fp16, *Rex* remains the best or close-second across all step budgets, with the advantage particularly pronounced in the few-step regime where EDICT fails. Pixel-space MSEs are dominated by the VAE reconstruction error and are essentially identical across reversible solvers; we report them for completeness.

*Table 6.* Quantitative comparison of different reversible solvers in terms of average CLIP score, Image Reward, and PickScore for conditional text-to-image generation with Stable Diffusion v1.5 ($512 \times 512$) with the non-reversible DDIM as a baseline. Our reversible ODE solvers are in pink and reversible SDE solvers in mauve.

| Solver / Steps | CLIP score ($\uparrow$) | | | Image Reward ($\uparrow$) | | | PickScore ($\uparrow$) | | |
|---|---|---|---|---|---|---|---|---|---|
| | 10 | 20 | 50 | 10 | 20 | 50 | 10 | 20 | 50 |
| DDIM | 31.78 | 31.76 | 31.24 | 0.033 | 0.136 | 0.247 | 21.06 | 21.29 | 21.04 |
| EDICT | 27.97 | 31.04 | 31.17 | -1.219 | -0.134 | -0.055 | 19.52 | 20.84 | 21.05 |
| BDIA $\gamma = 0.96$ | 31.11 | 31.52 | 31.54 | -0.111 | 0.067 | 0.087 | 20.52 | 21.01 | 21.19 |
| BDIA $\gamma = 0.5$ | 31.57 | 31.48 | 31.48 | -0.006 | 0.055 | 0.066 | 20.98 | 21.16 | 21.21 |
| O-BELM | 31.47 | 31.43 | 31.51 | 0.051 | 0.105 | 0.160 | 20.88 | 21.00 | 21.16 |
| *Rex* (Midpoint) | 31.62 | **31.64** | **31.60** | 0.119 | 0.179 | 0.198 | 21.28 | 21.38 | 21.41 |
| *Rex* (RK4) | **31.69** | 31.60 | 31.57 | 0.156 | 0.187 | 0.195 | 21.35 | 21.40 | 21.41 |
| *Rex* (Euler-Maruyama) | **31.68** | 31.56 | 31.33 | 0.222 | 0.239 | **0.264** | **21.50** | **21.66** | 21.70 |
| *Rex* (ShARK) | 31.55 | 31.56 | 31.39 | **0.239** | **0.249** | 0.263 | **21.51** | **21.66** | **21.72** |

## H.2. Unconditional Image Generation

In Table 5 we report the raw metric values corresponding to the radar visualization in Figure 5. Across all three step budgets we observe that *Rex* variants consistently match or exceed the non-reversible DDIM baseline as well as the reversible baselines (EDICT, BDIA, O-BELM). The stochastic *Rex* (Euler-Maruyama) scheme is particularly strong at the 20 and 50 step budgets, attaining the best FD, $FD_\infty$, Precision, Density, and Coverage at 50 steps.

**A Note on Higher-Order Schemes.** Comparing the entries of Table 5 one might at first be surprised to find that the higher-order *Rex* (RK4) performs worse than the lower-order *Rex* (Euler) and that the SDE variant *Rex* (Euler–Maruyama) outperforms both. This is consistent with prior observations within the diffusion model community: for unconditional sampling with diffusion models, higher-order ODE schemes have often been observed to be *no better*—and sometimes worse—than first-order schemes in low-step regimes (Lu et al., 2022b;a). In contrast, in flow matching and AI4Science settings, adaptive higher-order schemes such as Dormand–Prince do consistently outperform Euler (Tong et al., 2024; Rehman et al., 2026a; Tan et al., 2025a); we see this same effect for the Boltzmann sampling experiment in Table 1. The comparison between *Rex* (RK4) and *Rex* (Euler–Maruyama) is also less surprising than it may at first appear: the latter is an *SDE* scheme and benefits from a stochastic regularization effect which pulls trajectories back toward the data manifold; this can dominate any gains from higher-order discretization. We therefore emphasize that our central claim is not that higher-order schemes are universally superior, but that *Rex* affords arbitrary order of convergence, giving practitioners the freedom to select the order appropriate for their setting.

## H.3. Conditional Image Generation

In Table 6 we report the raw CLIP, Image Reward, and PickScore values corresponding to the nine-axis radar in Figure 8. The stochastic *Rex* variants (Euler-Maruyama and ShARK) consistently lead the Image Reward and PickScore axes at all three step budgets, while every *Rex* variant stays within a tenth of a point on CLIP of the non-reversible DDIM baseline.

*Table 7.* Quantitative comparison of different reversible solvers for image editing with the non-reversible DDIM as a baseline. Our reversible ODE solvers are in pink and adaptive ODE solvers are in teal. Best results in **bold**, second best under-lined.

| Solver | Image Reward ($\uparrow$) | CLIP score ($\uparrow$) | PickScore ($\uparrow$) | LPIPS ($\downarrow$) |
|---|---|---|---|---|
| DDIM | -0.564 | **19.17** | 18.367 | 0.214 |
| BDIA | -2.205 | 18.57 | 16.956 | 0.885 |
| O-BELM | -0.639 | 19.16 | 18.416 | 0.140 |
| *Rex* (Euler) | -0.551 | **19.17** | **18.721** | 0.109 |
| *Rex* (Dopri5) | **-0.547** | 19.16 | 18.698 | **0.107** |

## H.4. Image Editing

In Table 7 we report the raw Image Reward, CLIP score, PickScore, and LPIPS values for the round-trip image-editing experiments visualised in Figure 9. Both fixed-step *Rex* (Euler) and adaptive-step *Rex* (Dopri5) match or beat the non-

*Table 8.* Profiler report of a *Rex* (ShARK) sampling run with 50 steps.

| Section | Calls | Wall (s) | CPU (s) | Avg Wall | Peak CPU MB | VRAM In | VRAM Out | Δ VRAM |
|---|---|---|---|---|---|---|---|---|
| model_loading | 1 | 2.754 | 2.470 | 2.754 | 27.53 | 0.0 | 4096.8 | +4096.8 |
| data_loading | 1 | 0.002 | 0.002 | 0.002 | 1.13 | 4096.8 | 4096.8 | +0.0 |
| rex/encode_prompt | 10 | 0.367 | 0.362 | 0.037 | 0.08 | 4125.6 | 4238.7 | +113.1 |
| rex/init_noise | 10 | 0.050 | 0.050 | 0.005 | 0.06 | 4238.7 | 4238.9 | +0.1 |
| rex/brownian_interval | 10 | 0.007 | 0.007 | 0.001 | 0.06 | 4238.9 | 4238.9 | +0.0 |
| rex/inference | 10 | 104.914 | 104.697 | 10.491 | 0.35 | 4238.9 | 4239.0 | +0.1 |
| sampling/rex | 10 | 105.405 | 105.170 | 10.540 | 0.37 | 4125.6 | 4128.8 | +3.2 |
| decode_and_save | 10 | 1.438 | 1.365 | 0.144 | 3.79 | 4128.8 | 4128.8 | +0.0 |
| **__TOTAL__** | **1** | **109.665** | **109.071** | **109.665** | **3.77** | **0.0** | **4128.8** | **+4128.8** |

reversible DDIM baseline on every metric, with LPIPS roughly halved relative to DDIM. We omit EDICT from this table because the model failed entirely on this benchmark, collapsing to the identity map.

### H.5. Boltzmann Sampling

In Figure 11 we provide a qualitative comparison of the energy distributions corresponding to the quantitative results reported in Table 1. We plot the true MD energy distribution alongside the unweighted and reweighted proposal distributions obtained from the DiT model when integrated with the non-reversible Dopri5 scheme and with the reversible *Rex* (Dopri5) scheme. The reweighted proposals from *Rex* (Dopri5) more closely match the true MD energy distribution, consistent with the substantial improvement in $\mathcal{E}\text{-}\mathcal{W}_2$ reported in Table 1.

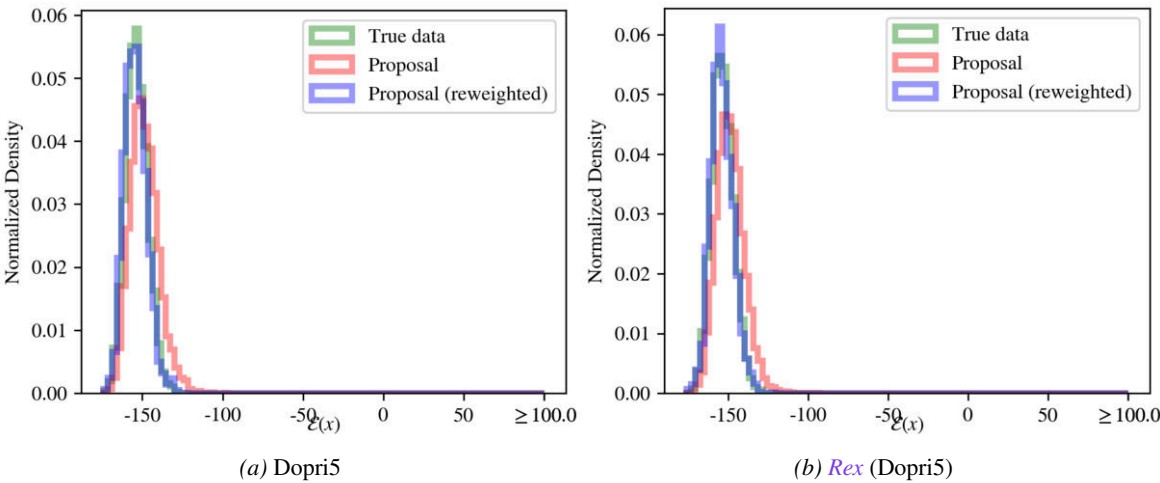

*(a) Dopri5*  *(b) Rex (Dopri5)*

*Figure 11.* True MD energy distribution with unweighted and reweighted proposals created using *Rex* (Dopri5).

### H.6. Brownian Interval Overhead

Our stochastic *Rex* schemes rely on Kidger's *Brownian Interval* (Kidger et al., 2021) to reconstruct the realizations of the Brownian motion and the associated space-time Lévy increments without storing the entire trajectory (see Appendix F.3). Since the Foster-Reis-Strange schemes only require the space-time Lévy area—which admits an exact closed-form joint distribution with the Brownian increment (Foster et al., 2020, Remark 3.6)—we avoid the notoriously difficult space-space Lévy area entirely; for further details see Jelinčič et al. (2024, Section 3). The Brownian Interval has $\mathcal{O}(\log N)$ query times and $\mathcal{O}(N)$ memory in the number of queries. For longer horizons and higher-order schemes the queries remain exact and the convergence behavior is governed by the underlying numerical scheme rather than the Brownian sampler. We emphasize that our work is agnostic to the choice of underlying mechanism for sampling the Brownian and space-time Lévy increments; we adopt the Brownian Interval because it is lightweight and well suited to our needs.

In Table 8 we profile *Rex* (ShARK) with 50 solver steps. The Brownian Interval contributes essentially *no* meaningful overhead, accounting for a negligible fraction of the total wall-clock time and adding no measurable VRAM usage compared with the cost of the network evaluations themselves.

*Table 9.* Ablations on image editing with Stable Diffusion v1.5 ($512 \times 512$) at varying number of steps; CFG scale of 3.0. Inversion error is reported as MSE in the latent space. Best results in **bold**, second best underlined.

| Steps | Variant | Inv. error ($\downarrow$) | LPIPS ($\downarrow$) | CLIP score ($\uparrow$) | ImageReward ($\uparrow$) | PickScore ($\uparrow$) |
|---|---|---|---|---|---|---|
| | *Rex* (Euler) *baseline* | $\underline{1.08 \times 10^{-6}}$ | 0.0505 | **19.1511** | **−0.4351** | 19.0827 |
| 10 | No rev. coupling (DDIM) | $4.54 \times 10^{-2}$ | 0.1552 | 19.1126 | −0.6081 | 18.9360 |
| | No exp. transform | $2.52 \times 10^{-6}$ | 0.0309 | 19.1479 | −0.4582 | **19.1999** |
| | No time reparam | $\mathbf{1.13 \times 10^{-8}}$ | **0.0275** | 19.1477 | −0.4746 | **19.2000** |
| | *Rex* (Euler) *baseline* | $\underline{8.66 \times 10^{-7}}$ | 0.0532 | **19.1552** | **−0.4142** | 19.1135 |
| 20 | No rev. coupling (DDIM) | $1.68 \times 10^{-2}$ | 0.0950 | 19.1289 | −0.4555 | 19.1270 |
| | No exp. transform | $2.17 \times 10^{-6}$ | 0.0309 | 19.1469 | −0.4600 | 19.1813 |
| | No time reparam | $\mathbf{6.38 \times 10^{-9}}$ | **0.0275** | 19.1481 | −0.4755 | **19.2010** |
| | *Rex* (Euler) *baseline* | $\underline{8.65 \times 10^{-7}}$ | 0.0577 | **19.1487** | **−0.3838** | 19.1596 |
| 50 | No rev. coupling (DDIM) | $4.21 \times 10^{-3}$ | 0.0664 | 19.1461 | −0.3906 | 19.1761 |
| | No exp. transform | $1.36 \times 10^{-6}$ | 0.0311 | **19.1476** | −0.4571 | 19.1831 |
| | No time reparam | $\mathbf{2.69 \times 10^{-9}}$ | **0.0275** | 19.1482 | −0.4755 | **19.2015** |

## H.7. Ablation Study

We provide ablations of the three core ingredients of *Rex*: the *reversible coupling*, the *exponential transformation*, and the *time reparameterization*. We run *Rex* (Euler) on the image editing task with Stable Diffusion v1.5 at $512 \times 512$ resolution and CFG scale of 3.0, and report inversion error (latent MSE), LPIPS, CLIP score, ImageReward, and PickScore. Results are reported in Table 9.

**Reversible Coupling.** Removing the reversible coupling reduces *Rex* (Euler) to (the non-reversible) DDIM (*cf*. Corollary E.2.1). The inversion error increases by at least four orders of magnitude and the downstream editing quality (LPIPS, ImageReward) degrades noticeably.

**Exponential Transformation.** Removing the exponential transform leads to a different (non-exponential) reversible scheme. Interestingly, this variant produces slightly higher LPIPS-preservation at a small cost to edit quality (ImageReward). This is consistent with the role of the exponential treatment of the linear drift, which biases the solver towards higher-fidelity edits.

**Time Reparameterization.** Removing the time reparameterization yields evenly spaced steps in $t$ rather than uniform steps in $\varsigma$. This is known to influence diffusion sampling behaviour and we observe improved inversion error at slightly worse editing performance.

## H.8. Visualization of Inversion and the Latent Space

We conduct a further qualitative study of the latent space produced by inversion and the impact various design parameters play. First in Figure 12 we show the process of inverting and then reconstructing a real sample. Notice that while the data prediction formulation worked great in sampling and still possesses the correct reconstruction, *i.e.*, it is still reversible, the latent space is all messed up. The variance of $(\boldsymbol{x}_n, \hat{\boldsymbol{x}}_n)$ tends to about $10^7$, many orders of magnitude too large! We did observe that raising $\zeta = 1 - 10^{-9}$ did help reduce this, but it was still relatively unstable. *N.B.*, these trends hold in a large number of discretization steps (we tested up to 250); however, for visualization purposes we chose fewer steps.

Conversely, the noise prediction formulation is much more stable, see Figure 13. The variance of $(\boldsymbol{x}_n, \hat{\boldsymbol{x}}_n)$ is on the right order of magnitude this time, however, there are strange artefacting and it is clear the latent variables are not normally distributed.

Moving to the SDE case with ShARK in Figure 14, we see that the data prediction formulation is so unstable in forward-time that we ran into overflow errors and can no longer achieve algebraic reversibility. However, the noise parameterization with ShARK, see Figure 15, works very well with the latent variables appearing to be close to normally distributed.

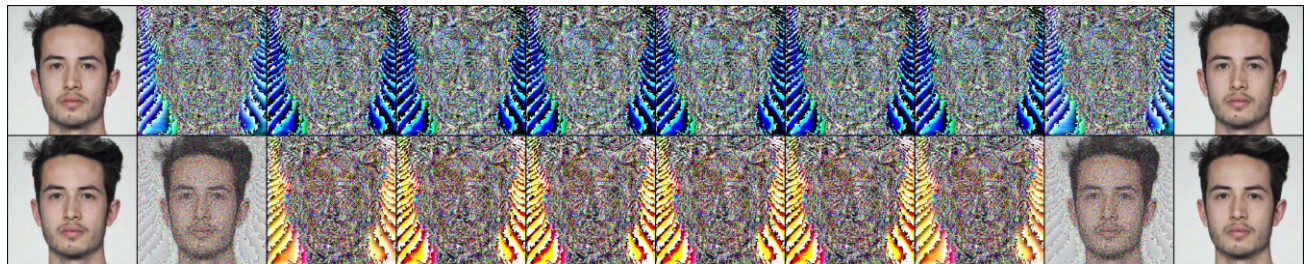

*Figure 12.* Inversion followed by sampling with *Rex* (Euler) 5 steps, $\zeta = 0.999$. Data prediction. Top row tracks $\boldsymbol{x}_n$, bottom row $\hat{\boldsymbol{x}}_n$.

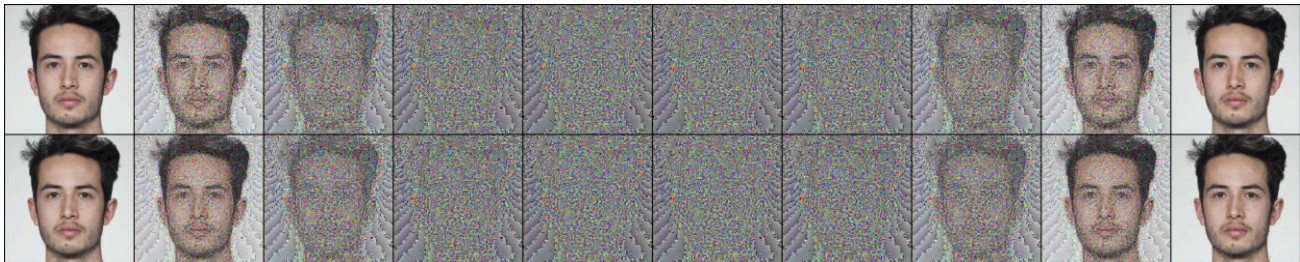

*Figure 13.* Inversion followed by sampling with *Rex* (Euler) 5 steps, $\zeta = 0.999$. Noise prediction. Top row tracks $\boldsymbol{x}_n$, bottom row $\hat{\boldsymbol{x}}_n$.

### H.9. Interpolation

We explore interpolating between the inversions of two images, a difficult problem as the inverted space is often non-Gaussian (Blasingame & Liu, 2024b). We illustrate an example of this in Figure 16 exploring interpolation with an unconditional DDPM model. We observe that stochastic *Rex* has much better interpolation properties than both ODE inversions, corroborating with Nie et al. (2024). Both ODE variants seem to fail quite noticeably, unable to smoothly interpolate between the two samples. *N.B.*, we noticed that the inverted samples with ShARK had variance much closer to one, whereas the other inverted samples had much larger variance, likely contributing to the distortions.

### H.10. Uncurated Image Generation Samples

We present uncurated text-to-image samples produced by *Rex* with Stable Diffusion v1.5 ($512 \times 512$) across several underlying solvers and discretization-step budgets. Figures 17 and 18 show samples generated by *Rex* (RK4) at 10 and 50 steps, respectively, and Figures 19 and 20 show samples generated by *Rex* (ShARK) at the same step counts.

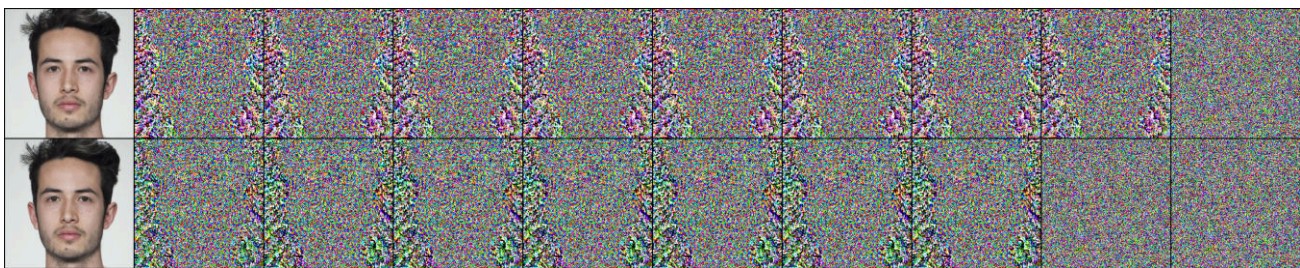

*Figure 14.* FAILURE CASE! Inversion followed by sampling with *Rex* (ShARK) 5 steps, $\zeta = 0.999$. Data prediction. Top row tracks $\boldsymbol{x}_n$, bottom row $\hat{\boldsymbol{x}}_n$.

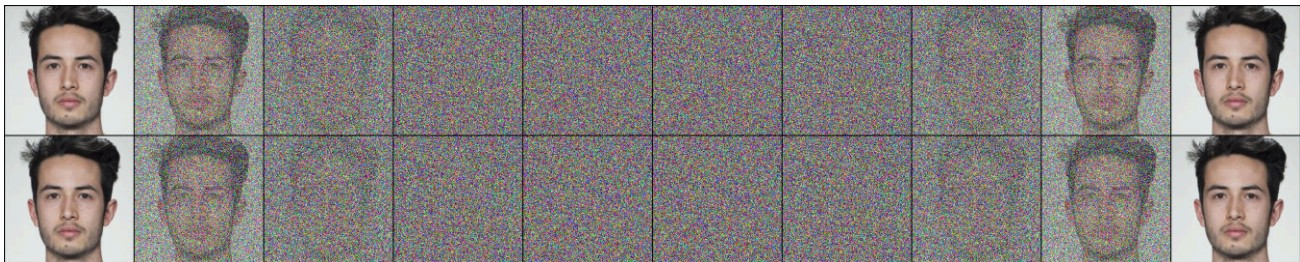

*Figure 15.* Inversion followed by sampling with *Rex* (ShARK) 5 steps, $\zeta = 0.999$. Noise prediction. Top row tracks $\boldsymbol{x}_n$, bottom row $\hat{\boldsymbol{x}}_n$.

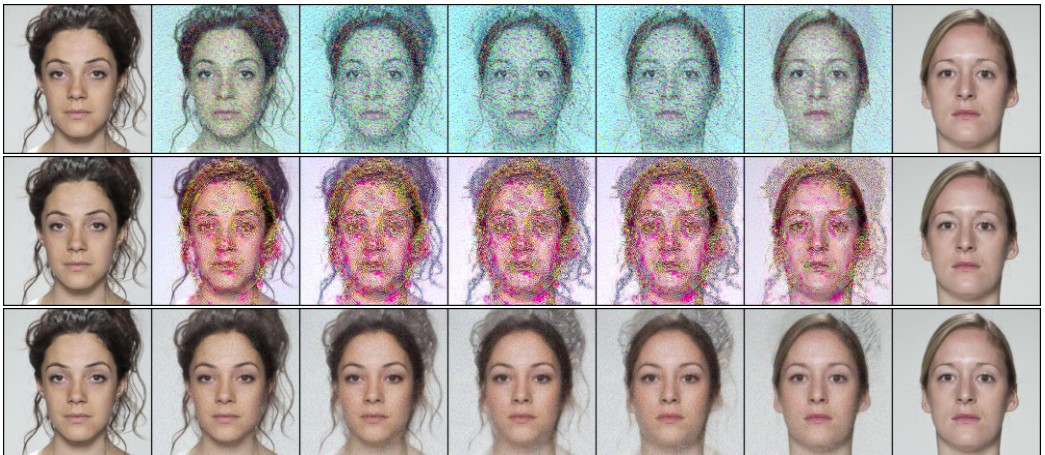

*Figure 16.* Unconditional interpolation between two real images from FRLL (DeBruine & Jones, 2017) with a DDPM model trained on CelebA-HQ. Top row is BELM, middle is *Rex* (Euler), and bottom is *Rex* (ShARK). 50 steps used for each method.

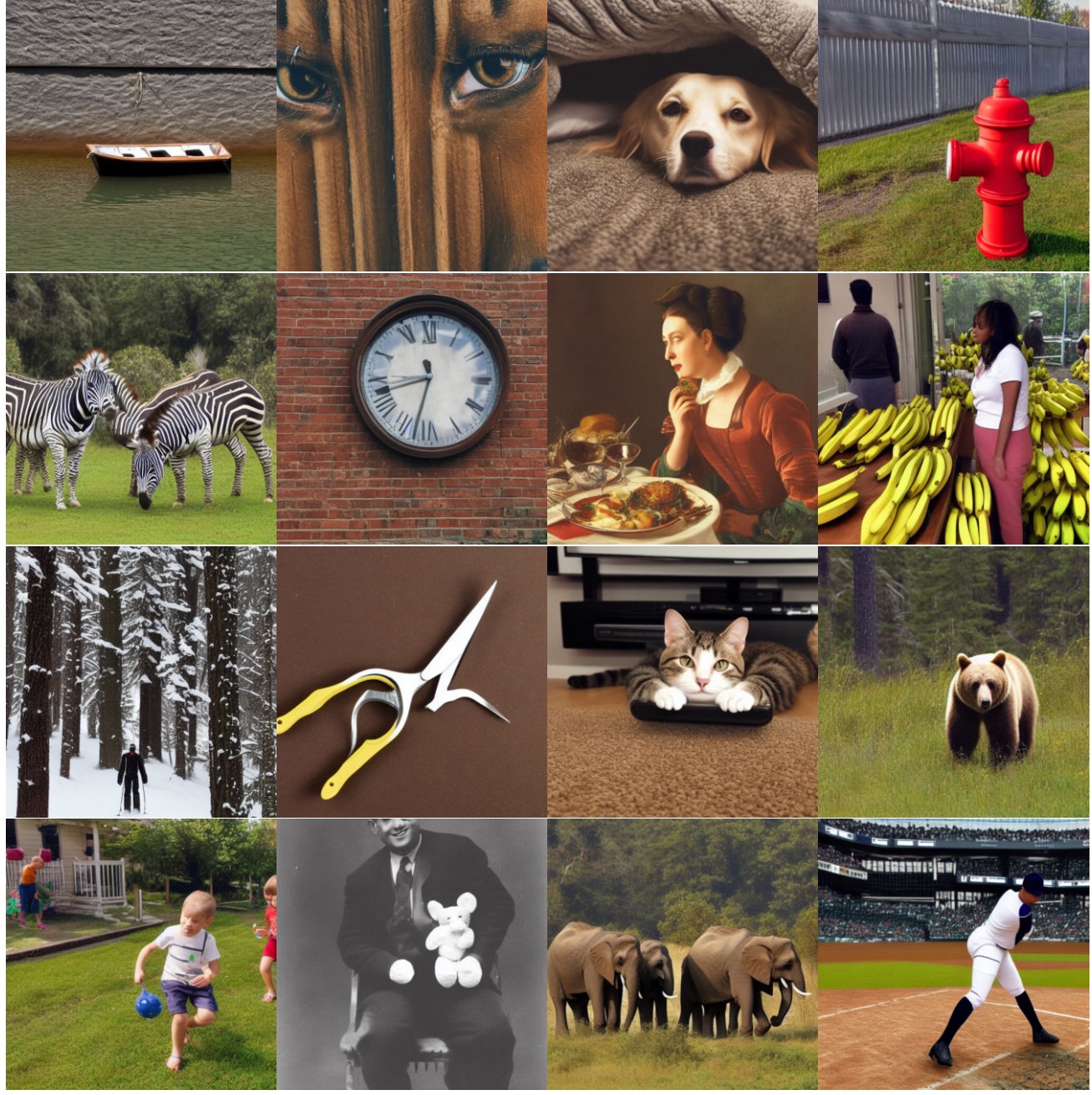

*Figure 17.* Uncurated samples created using *Rex* (RK4) and Stable Diffusion v1.5 ($512 \times 512$) and 10 discretization steps.

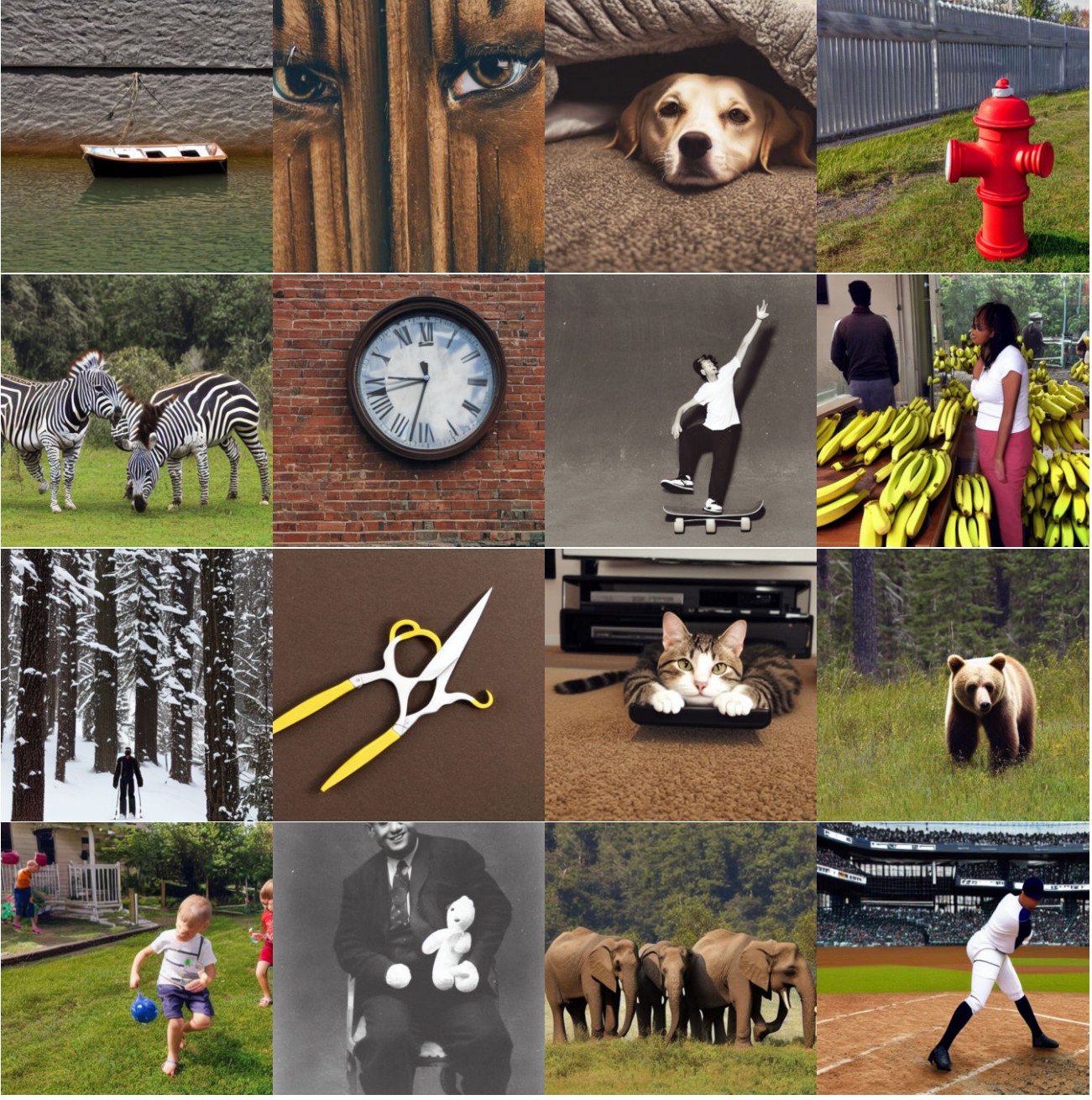

*Figure 18.* Uncurated samples created using *Rex* (RK4) and Stable Diffusion v1.5 ($512 \times 512$) and 50 discretization steps.

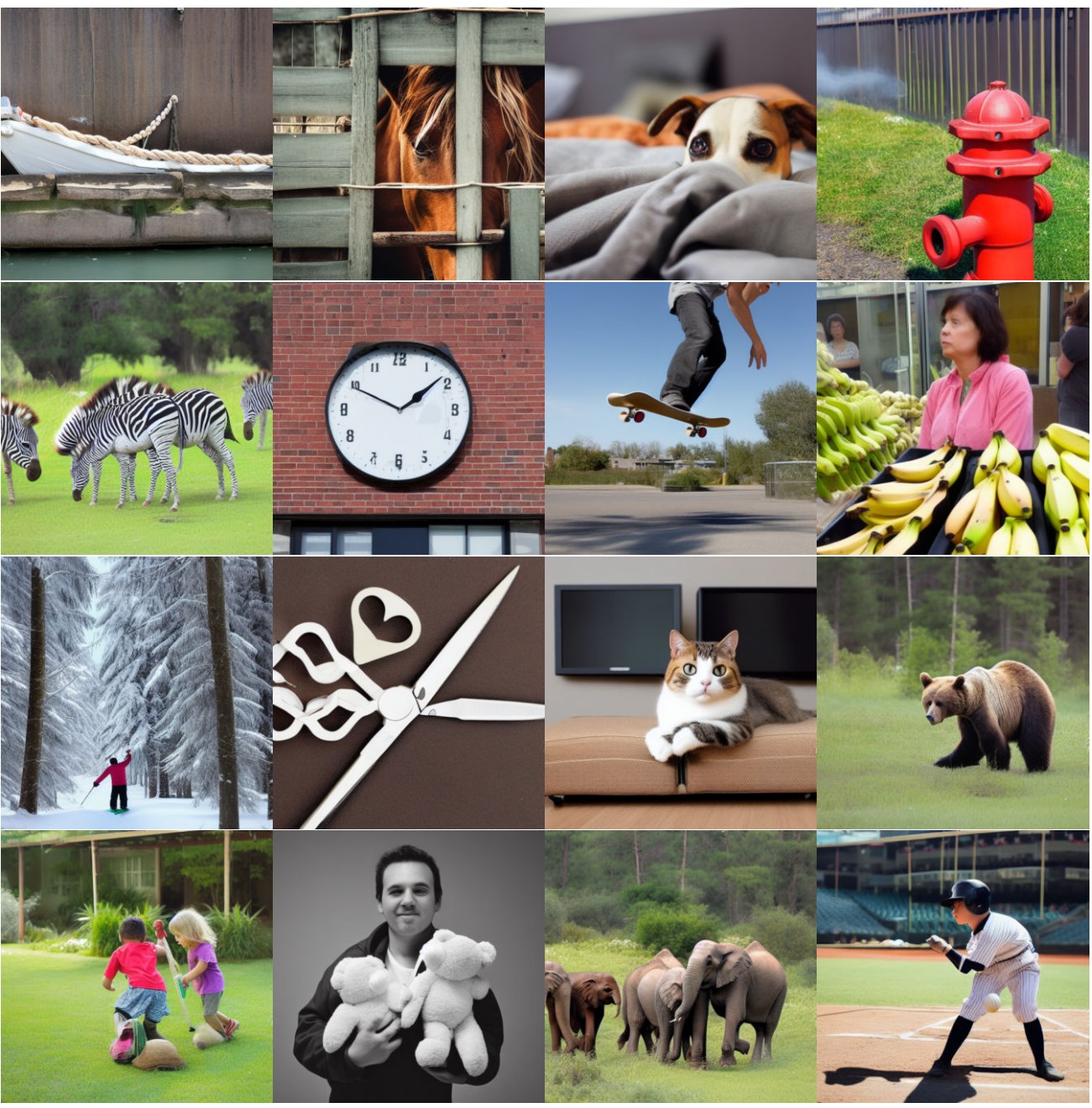

*Figure 19.* Uncurated samples created using *Rex* (ShARK) and Stable Diffusion v1.5 (512 × 512) and 10 discretization steps.

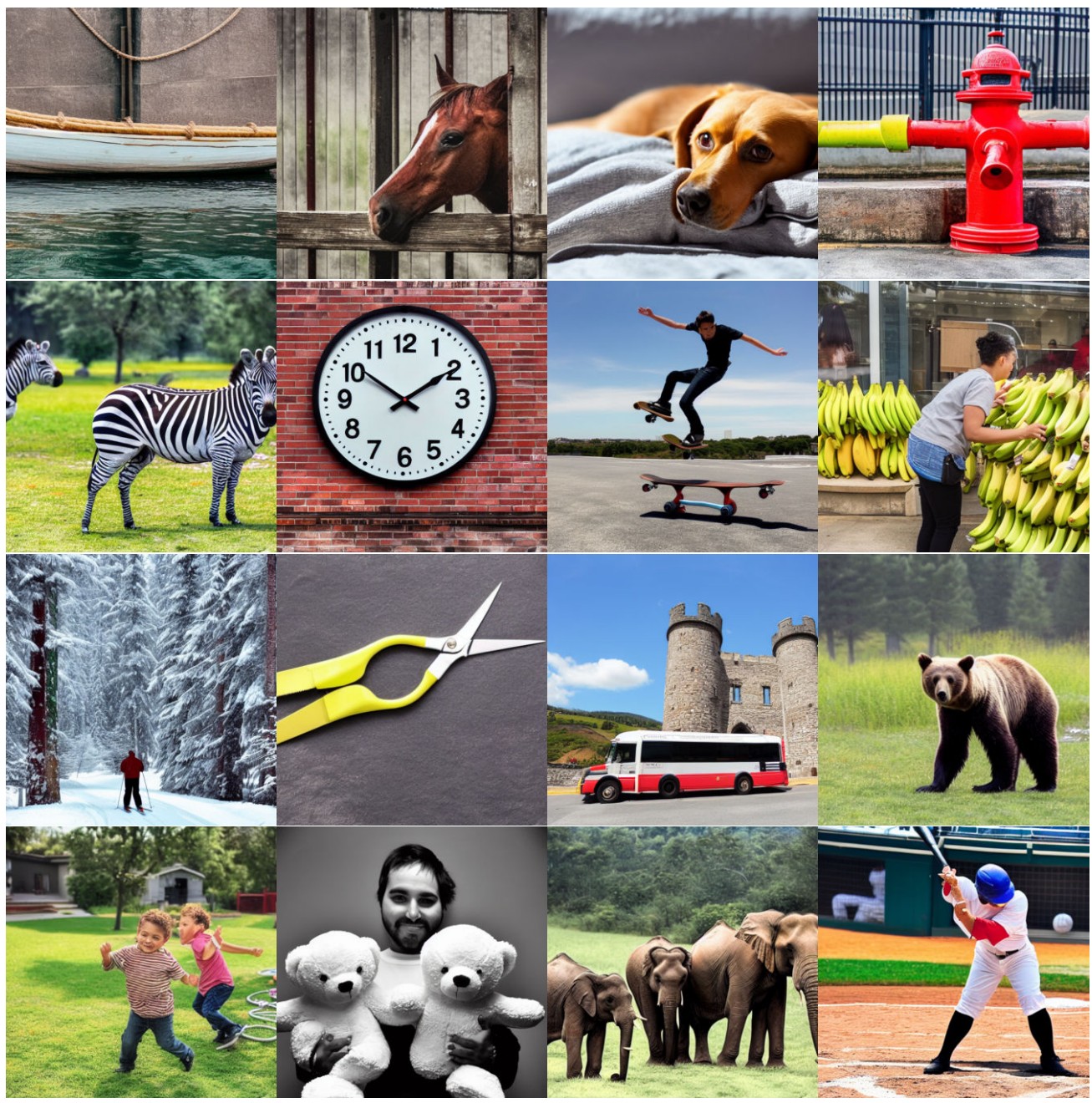

*Figure 20.* Uncurated samples created using *Rex* (ShARK) and Stable Diffusion v1.5 (512 × 512) and 50 discretization steps.

