# OpenReview forum: "Rex: A Family of Reversible Exponential (Stochastic) Runge-Kutta Solvers"
_ICML.cc/2026/Conference — ICML 2026 spotlight_

### Official Review · Reviewer_gN6Y · 2026-03-11

**Soundness:** 3
**Presentation:** 2
**Significance:** 3
**Originality:** 3
**Overall Recommendation:** 5
**Confidence:** 3

**Summary:**

This paper introduces REX, a family of algebraically-reversible numerical ODE/SDE solvers for diffusion models. These solvers enable exact inversion, mapping samples from the data distribution back to the prior distribution without reconstruction errors. The key technical innovation is to use the McCallum-Foster method after a change of variables and then to go back to the original state variables using Lawson methods. With this one can construct solvers for both the probability flow ODE and reverse-time SDE formulations of diffusion models. Empirical results on image generation and interpolation show the effectiveness of the proposed method.

**Compliance With Llm Reviewing Policy:**

Affirmed.

**Final Justification:**

The authors have addressed all our concerns and we are happy to raise our score to 5.

**Key Questions For Authors:**

1.) A key point of this paper is the ability to achieve "arbitrarily high order of convergence". However, the empirical results in Tables 1, 2, and 4 show that the high-order Rex (RK4) performs worse than its low-order counterparts (Rex-Euler, Rex-EM). These significant findings are never dicussed nor analyzed.

2.) Figure 3. How do I have to interpret the qualitative findings? What does it tell me that the Rex image is qualitatively and visually quite different to the other images?

3.) For the unconditional sampling task (Table 1), the Rex-ODE solvers are outperformed by the O-BELM baseline at 10, 20 and even 50 steps. This suggests that for pure sampling, the theoretical stability of Rex does not necessarily translate to superior FD. Can this be discussed further?

4.) What is the limiting factor that the presented solvers only work for diffusion models and are not a general family of reversible solvers?

**Limitations:**

There is an impact statement but this is done very badly and very generic. The authors state: "There are many potential societal consequences of our work, none which we feel must be specifically highlighted here."

I would like to know which are the societal consequence (bad or good) that exist. Also, in terms o f technical limitations, there could be much more discussion of their method. See also above.

**Strengths And Weaknesses:**

Soundness:
The paper makes significant theoretical contributions by constructing methods for the exact SDE inversion without storing complete Brownian motion trajectories. The paper presents extensive derivations and proofs for the method itself as well as empirical evaluations of the capabilities of the scheme.
However, it is interesting or maybe even worrisome that although the higher order capabilities of the Rex solvers results show that O-BELM still performs better than Rex. And even the more simplistic Rex-Euler method is better than the higher order Rex schemes. These findings are not well enough discussed. In general, higher order methods do not seem to lead to performance improvements, which is not sufficiently reflected upon and for low order baseline methods, Rex reduces to known techniques (which is still interesting) Similarly, one of the weaknesses of traditional reversible solvers is poor stability, while the McCallum-Foster methods have a non-trivial non-zero region of stability, the reversibility / stability tradeoff might benefit from more practical discussion.


Presentation:
The results are presented well, though throughout the paper in parts the writing contains many errors and missing words which disturb the reading flow.

Significance:
The paper provides a deep and valuable theoretical derivation and analysis of algebraically reversible solvers. This demonstrates that recent reversible solvers are fundamentally variants of the leapfrog/midpoint method and are thus nowhere linearly stable. Furthermore the paper shows how the family of Rex solvers reduces to known techniques and as such represents unifying solver family.

Originality: To the best of my knowledge, the work is original.

---

> ### Author Rebuttal · Authors · 2026-03-29
>
> We thank Reviewer gN6Y for their detailed review and interesting questions. We address all the questions below and are happy to provide further clarifications or answer any follow-up questions.
>
> ### Question 1
>
> Thank you for pointing out this missed discussion; we are revising our manuscript to now include this discussion. Prior works [1,2] have noticed that higher-order schemes actually performed worse than low-order schemes for sampling with diffusion models in certain settings; which seems to be the case for **Table 1**.
> However, for other scenarios (in particular flow matching) [3] noticed that adaptive higher-order scheme like Dormand-Prince 4/5 performs much better than Euler. These findings seem to be corroborated in recent papers within the AI4Science domain [4,5] which use the flow matching formalisms.
>
> Regarding Rex (RK4) vs. Rex (EM): this distinction is less surprising than it may initially appear, since Rex (RK4) is an ODE scheme whereas Rex (EM) is an SDE scheme. The SDE scheme benefits from a stochastic regularization effect that pulls trajectories back toward the data manifold, which can dominate any gains from higher-order discretization. Comparing convergence order across ODE and SDE schemes is therefore less informative than comparing two explicit ODE schemes or two SDE schemes with one another.
>
> To clarify our central claim: we do not assert that high-order schemes are universally superior, but that Rex **can** achieve arbitrary convergence order, giving practitioners the freedom to select the order appropriate for their setting. We will update the manuscript to reflect this more precisely.
>
> ### Question 2
>
> Thank you for raising this interesting question. **Figure 3** is a qualitative supplement to **Tables 1 and 7** and should be interpreted alongside those quantitative results. At a low number of function evaluations, the choice of numerical scheme has a pronounced effect on the trajectory explored during sampling, leading to visually distinct outputs. We acknowledge that the assessment of which image appears more "natural" is inherently subjective and we do not draw quantitative conclusions from it; we include the figure only to give an intuitive sense of this trajectory-level difference. We refer the reviewer to **Tables 1 and 7** for the primary quantitative evidence.
>
> ### Question 3
>
> This is a fair observation. For the unconditional sampling experiment we used $\zeta = 0.999$, which actually yields a smaller region of stability than is theoretically achievable (stability is maximized as $\zeta \to 0$). We made this choice deliberately to ensure a valid comparison: lower $\zeta$ values introduce $\zeta^{-1}$ terms in the conceptual backward solve which can cause numerical instability, and [6] provides a more detailed discussion of this trade-off. As a consequence, Rex's stability advantage is not fully exploited in the unconditional setting, allowing O-BELM to outperform Rex. However, in text-guided generation with CFG, the stability benefit becomes clearer, as perturbing the vector field for guidance places greater demands on stability.
>
> ### Question 4
>
> Rex is broadly applicable beyond diffusion models; we are updating the manuscript to make this clearer. The Rex scheme holds for any SDE of the form
>
> $$\mathrm d \boldsymbol X_t = \left[a(t) \boldsymbol X_t + b(t) \boldsymbol f(t, \boldsymbol X_t)\right]  \mathrm dt + g(t) \mathrm d\boldsymbol W_t,$$
> where $g(t) = \sqrt{a(t)b(t)}, b(t)/a(t) \to \infty$ as $t \to \infty$ and $b(t) > 0$. The first condition ensures the exponentially reweighted SDE admits a closed form via a time-domain change-of-variables, and can be relaxed for a slightly less elegant formulation. This covers a large class of semi-linear SDEs with additive noise.
>
> ### Limitations
> We recognise that improving generative AI efficiency carries broader societal risks, including: (1) misuse for synthetic media generation, where more efficient inversion lowers the barrier to creating deepfakes or deceptive content; (2) amplification of training-data biases in higher-fidelity outputs; and (3) increased accessibility of generative tools, which may facilitate disinformation at scale. We will expand the impact statement in the revised manuscript to reflect these points.
>
> ## References
>
> [1] Lu, Cheng, et al. Dpm-solver++: Fast solver for guided sampling of diffusion probabilistic models.
>
> [2] Cheng Lu et al. Dpm-solver: A fast ode solver for diffusion probabilistic model sampling in around 10 steps.
>
> [3] Tong, Alexander, et al. Improving and generalizing flow-based generative models with minibatch optimal transport.
>
> [4] Rehman, D., Akhound-Sadegh, T., Gazizov, A., Bengio, Y.,
> and Tong, A. FALCON: Few-step accurate likelihoods for
> continuous flows.
>
> [5] Tan, C. B., Bose, A. J., Lin, C., Klein, L., Bronstein,
> M. M., and Tong, A. Scalable equilibrium sampling
> with sequential boltzmann generators.
>
> [6] McCallum et al. Efficient, Accurate and Stable Gradients for Neural ODEs

---

> > ### Author Rebuttal · Reviewer_gN6Y · 2026-04-01
> >
> > We thank the authors for their thorough answers to our comments. We have raised our score accordingly to 5.

---

> > > ### Author Response · Authors · 2026-04-06
> > >
> > > We are grateful to Reviewer gN6Y for your thoughtful questions and for raising the score after our rebuttal. We are pleased that our clarifications on the higher-order scheme behavior, the stochastic regularization effect distinguishing ODE and SDE comparisons, the stability trade-off in the unconditional sampling setting, and the broader applicability of Rex beyond diffusion models were all satisfactory. We have also expanded the impact statement in the revised manuscript as suggested.

---

### Official Review · Reviewer_6D4M · 2026-03-12

**Soundness:** 2
**Presentation:** 3
**Significance:** 2
**Originality:** 3
**Overall Recommendation:** 4
**Confidence:** 3

**Summary:**

This paper studies a recurring need in neural differential-equation-based generative models (e.g., diffusion models and continuous normalizing flows). Standard ODE/SDE solvers accumulate discretization errors, so backward integration fails to retrace the forward trajectory, making “exact” inversion practically impossible. The authors propose Rex, a family of Reversible Exponential (Stochastic) Runge–Kutta solvers. The key idea is to exploit the semi-linear structure common in diffusion ODE/SDEs via exponential/Lawson-type transformations, and to systematically “reversibilize” explicit (stochastic) Runge–Kutta schemes so that forward and backward steps are algebraically exactly invertible. Experiments on image generation/editing and a scientific Boltzmann sampling task show advantages over prior reversible solvers.

**Compliance With Llm Reviewing Policy:**

Affirmed.

**Final Justification:**

Since my score is already positive, I maintain my score.

**Key Questions For Authors:**

1. How does “exact reversibility” behave in realistic GPU settings? Please report forward→backward return-to-start error as a function of (i) number of steps, (ii) dtype (fp32 vs. bf16/fp16), and (iii) resolution. Also clarify whether any non-deterministic GPU operations must be disabled.

2.  Which splittable PRNG / Brownian tree implementation is used? How are Lévy areas sampled/reconstructed consistently? Provide wall-clock and memory overhead, plus robustness statistics for long horizons / higher order schemes.

3. What are the necessary conditions on the noise schedule and equation form (e.g., must it be VP, must it be semi-linear decomposable)? Does the approach extend to rectified flow / general flow matching or non-isotropic diffusions?


4. Provide clean ablations that remove (i) reversible coupling, (ii) Lawson/exponential transformation, and (iii) time reparameterization, and report the impact on (a) inversion error, (b) editing consistency, and (c) generation quality at matched compute. This would strengthen the mechanism→outcome story.

I would like to increase my score after the questions are resolved.

**Limitations:**

Yes

**Strengths And Weaknesses:**

**Strengths**
The overall technical direction is coherent: the paper leverages the semi-linear form of diffusion dynamics to motivate why handling the linear part exponentially can reduce systematic drift relevant to inversion. It clearly distinguishes “analytic reversibility” (often requiring iterative inversion) from “algebraic reversibility” (closed-form, step-wise invertibility). The paper provides theoretical arguments about preserving the convergence order of the underlying explicit (S)RK method, discusses stability properties, and evaluates both ODE and SDE variants in low-step regimes and in common inversion/editing use cases.


**Weaknesses**

1. How “exact” is exact reversibility in finite-precision implementations? The paper emphasizes algebraic exactness, but real diffusion sampling often runs on GPUs with float16/bfloat16 mixed precision, non-associative reductions, and fused kernels. These can break strict algebraic identities. The paper should provide direct numerical diagnostics, e.g., forward→backward return-to-start error vs. steps / dtype, and show it remains substantially better than baselines. Without this, the practical meaning of “exact inversion” is not fully established.

2. SDE inversion relies on complex stochastic reconstruction machinery. Splittable PRNGs, virtual Brownian trees, and (for higher order) Lévy area reconstruction are theoretically plausible but significantly raise implementation and reproducibility barriers. The paper argues it avoids storing the entire Brownian path, yet it lacks a systematic evaluation of reconstruction overhead, error, and robustness across different implementations (e.g., Brownian Interval vs. Virtual Brownian Tree).

3. Assumptions / applicability boundaries are not fully pinned down. Several derivations appear to rely on VP-like schedules, semi-linear decompositions, and specific boundary conditions. The paper should more explicitly state what breaks outside these settings (non-VP schedules, more general flow-matching formulations, non-isotropic diffusions, etc.). If the scope is narrower, the contribution should be framed accordingly.

4. Evidence for the central mechanism is not fully closed-loop. The paper shows improved generation metrics, but to attribute gains to reversibility/forward–backward consistency itself, it should provide stronger causal ablations: disable the reversible coupling, disable Lawson/exponential handling, or disable the time reparameterization, and measure (i) inversion error and (ii) downstream quality/editing metrics under matched compute. Some ablations exist, but the “mechanism → effect” chain can be made much tighter.

5. High conceptual density and a deep stack of technical components. Exponential integrators, time reparameterization, reversible coupling, SRK/Lévy area, and Brownian-tree reconstruction are all introduced, making it easy for readers to lose the main thread. A “minimal working recipe” page (with a clear block diagram and language-agnostic pseudocode) would greatly improve readability.

---

> ### Author Rebuttal · Authors · 2026-03-31
>
> We thank Reviewer 6D4M for their detailed review and interesting feedback coupled with a focus on numerics/SDE machinery. The suggested experiments have greatly improved the strength of the paper and we appreciate the reviewer for suggesting them to us. We answer their questions below.
>
> **Due to space limits we attached the results of our new experiments via an anonymous link here: https://anonymous.4open.science/r/Rex-rebuttal-C287/rebuttal.pdf**
>
> ### Question 1
> As the reviewer points out in practice the inversion isn't perfect due to numerics issues. We study this and find that **Rex is on average 3 orders of magnitude better than O-BELM and 8 orders of magnitude better than DDIM** w.r.t. inversion error.
> We record the image inversion/reconstruction error in **Rebuttal Table 1** for the Stable Diffusion runs. We report both MSE in pixel space and MSE in latent space, i.e., removing VAE error and isolating error due to numerics and we look at both float32 and float16 settings.
>
> Rex outperforms every model w.r.t. inversion error in nearly every tested scenario, and often by a wide margin. This is particularly noticeable in the regime of limited number of steps where EDICT fails.
>
> For our experiments we disabled non-deterministic GPU ops for fair comparisons on seed matched runs. We noticed that enabling them *did not impact inversion error.*
>
>
> ### Question 2
> We used Kidger's Brownian Interval as discussed in Appendix G.4 from [1]. Since the Foster-Reis-Strange schemes only use the Space-time Levy area it is actually quite simple to calculate these areas. Whilst not discussed in [1] the resulting `torchsde` repo supports it, a more in depth discussion on the mechanisms can be found in Kidger's follow up work with Foster in [2, Section 3]. In short since we don't approximate the space-space Levy area (which is notoriously difficult to compute) but rather use the tractable space-time Levy area see [3, Remark 3.6]. The Brownian interval has $\mathcal O(1)$ query times and the GPU memory is also $\mathcal O(1)$ compared to $\mathcal O(n)$ for storing all $n$ steps in memory, see [1, Appendix E.2] for further details.
>
> In **Rebuttal Table 2** we include a profiler of Rex and show that the Brownian Interval adds essentially no meaningful overhead both in VRAM and wall-clock time.
>
> For robustness for long horizons / higher order schemes: the Brownian intervals have exact queries so there is no error in querying both $\boldsymbol W_{s,t}$ and $\boldsymbol H_{s,t}$.The behavior of the higher-order schemes would be governed by the underlying behavior of the numerical scheme not the sampling mechanism for the Brownian Interval (assuming we don't use space-space Levy areas which we avoid). We would be happy to discuss this point more.
>
> Our work is agnostic to the choice of underlying querying mechanism for the Brownian and space-time Levy increments. We chose the Brownian Interval as it seemed a lightweight and useful tool for what we needed.
>
> ### Question 3
> We follow the standard diffusion conventions for the SDE formulation which we outline along with assumptions on the schedule in **Section 2.2**. As noted in **Remark 3.1** this also holds for a broad class of flow matching models including the most commonly used formulations [*cf.* 4, Table 1]. For the SDE case we do require additive noise. We only analyzed the case with semi-linear formulation. We discuss this with **Review gN6Y** in **Question 4** w.r.t. the SDE case.
>
>
> ### Question 4
>
> We report these ablations in **Rebuttal Table 3** and summarize our key observations.
>
> * Removing the reversible couplings destroys the performance with the non-coupled scheme having an inversion error that is at least $1,000,000\times$ greater than the reversible scheme which leads to poor downstream performance. As per **Corollary 3.5.1**, Rex (Euler) without reversible coupling coincides with DDIM.
>
> * Removing the exponential transform was interesting. In short this component moves the solver towards higher-quality edits at a small cost w.r.t. to proximity to the original as measured in LPIPS.
>
> * Removing the time reparameterization essentially changes the distribution of timestep schedule which is known to have an impact on diffusion models. Using Rex with our default settings of uniform steps in $\varsigma_t$ results in uneven step sizes in $t$. Disabling this results in evenly spaced steps in $t$. We found this improved the inversion error at slightly worse editing performance.
>
>
> ## References
>
> [1] Kidger, Patrick, et al. Efficient and accurate gradients for neural sdes.
>
> [2] Jelinčič, Andraž, James Foster, and Patrick Kidger. Single-seed generation of Brownian paths and integrals for adaptive and high order SDE solvers.
>
> [3] Foster, J., Lyons, T., and Oberhauser, H. An optimal polynomial approximation of brownian motion.
>
> [4] Lipman, Y. et al. Flow matching guide and code.

---

> > ### Author Rebuttal · Reviewer_6D4M · 2026-04-01
> >
> > My concerns have been adequately addressed.

---

> > > ### Author Response · Authors · 2026-04-06
> > >
> > > We thank Reviewer 6D4M for their constructive and detailed feedback. Your questions directly motivated meaningful new experiments, *viz.* the studies on the impact of numerics on inversion, Brownian Interval overhead profiling, and ablation studies, which we believe substantially strengthen the paper. We are glad these additions fully addressed your concerns and appreciate your willingness to engage closely with the rebuttal material.

---

### Official Review · Reviewer_empu · 2026-03-12

**Soundness:** 4
**Presentation:** 2
**Significance:** 3
**Originality:** 4
**Overall Recommendation:** 5
**Confidence:** 3

**Summary:**

In the paper a new family of algebraically reversible solvers of the probability flow ODE and reverse SDE is introduced. These solvers generally called Rex can be built upon many known ODE/SDE solvers like DDIM, DPM-Solver-1,2,3 and so on resulting in their reversible counterparts. The key ideas behind building this family of solvers is change of variables allowing to get rid of terms linear in data variable x (as in exponential solvers) and applying (a variant of) McCallum-Foster method to ensure reversibility. Rex solvers show excellent results both as conventional solvers in image-related tasks and as reversible solvers in specific tasks requiring this property both in image domain and in sampling from Boltzmann distributions.

**Compliance With Llm Reviewing Policy:**

Affirmed.

**Final Justification:**

This is a very exciting paper of high value to everyone who is interested in diffusion modeling. I wrote about its strong points in my initial review, so I won't repeat them again. The only weakness was, in my opinion, its presentation style, and I believe that the authors will make it better, given their thorough approach to the studied topic and to writing rebuttals. I think than the conference will benefit from this paper, and highly recommend to accept it.

**Key Questions For Authors:**

1. The key ingredient for reversibility is McCallum-Foster method, applied first to Y-domain where differential equations are nice enough, i.e. have only NN-dependent RHS without the term linear in data variable and, in case of SDEs, unit covariance due to properly chosen change of time variable, and later the formulae are translated to the initial X-domain. The mentioned method, as follows from the Definition 2.1, requires computing $\Phi$ in two different consecutively calculated points both in forward and reverse step. It means that if $\Phi$ involves n functions evaluations (e.g. n=1 for Euler or Euler-Maruyama solvers), then the overall number of function evaluations (with neural network) is 2n for each solver step. And, moreover, neural network cannot make these 2n computation in batch mode since the input points are computed consecutively. Am I correct?
2. As for the adaptive solver in Table 3, what was the average number of solver steps on that dataset?
3. As far as I understand, you avoid using common stochastic Runge-Kutta methods mentioned in (Robler 2025) because they employ iterated integrals with respect to Brownian motion increments which are usually approximated by random variables just with the matching moments. You propose to use Foster-Reis-Strange scheme which instead of iterated integrals employs space-time Levy area (basically, the area between Brownian motion and its approximation with a line connecting endpoints, scaled by the time length). But the corresponding formulae (24) for the SRK solver are said to hold for Stratonovich SDEs only, while stochastic integrals in diffusion analysis are (almost) always understood in Ito's sense. In particular, the way noise is added to data (formulae (3-4)) is derived under Ito's calculus, and Anderson's theorem allowing to write the reverse SDE (5) is also derived for Ito's integrals. Isn't there some contradiction? Or maybe due to only linear dependence in X in the drift of the forward SDE (3) (and independence of X of the diffusion term) the formulae (3-5) hold for Stratonovich integrals too?

**Limitations:**

Yes

**Strengths And Weaknesses:**

In my opinion, this is a very good paper and a very interesting reading for everybody who is interested in diffusion modeling. Unfortunately, I couldn't thoroughly read all the appendices - I went through almost everything in Appendices A-B (which is a must for this particular paper because they contain the final formulae for Rex solvers of ODE and SDE in cases of data/noise prediction and related derivations), but I did not dive into convergence stuff, stability regions and everything related with detailed theoretical comparison with other popular solvers and their Rex counterparts. This is partially due to the fact that this year I had 6 (six!) papers for review some of them, like the present one, having important and voluminous appendices. But this is definitely not the problem of the authors (in fact, I believe this is the problem ICML organizers should definitely do something with). Anyway, I believe that I got to understand the core part of the paper well, and I think this is the paper of high value. Let me write about its strong points first:
1. This paper deals with algebraically reversible solvers meaning that it introduces solvers with both forward and backward schemes exactly matching each other. The authors introduce reversibility by a variant of the McCallum-Foster method (see Definition 2.1).
2. A remarkable point of the paper is introducing (for the first time in diffusion-related literature) algebraically reversible SDE solvers without the necessity of storing the whole path of Brownian motion. Although the method the authors use for this is not novel (see discussion in Section 3.2), this is the first time a technology allowing to revert Brownian motion trajectory without keeping it memory is applied to algebraically reversible solvers resulting in the first efficiently reversible SDE solver which is quite impressive.
3. The Rex family of solvers is extremely flexible, it allows for reversible counterparts of many known ODE/SDE solvers used in diffusion modeling and also for solvers with adaptive step sizes.
4. In the paper it is clearly and convincingly shown that Rex solvers outperform existing methods for exact diffusion inversion even without hyperparameters tuning (e.g. tuning $\zeta$).
5. Rex solvers can also be used as conventional solvers in the scenarios not requiring reversibility and they are quite competitive in this case as well (e.g. better than DDIM). In particular, stochastic Rex solvers perform very well even when the number of steps is small, which is quite unusual for diffusion models.
6. In general, as I mentioned in the beginning, this paper has, as far as I can judge, solid theoretical grounds and can be an interesting and fruitful reading for those specializing in diffusion modeling.

As for the weaknesses, the only serious weakness I can mention is the presentation style including typos, tables without references, referencing formulae from the Appendices in the main part of the paper, etc. I strongly encourage the authors to fix the issues that I found (I will list them below) and look once again for typos in Appendices because, as I wrote, I thoroughly went only through Appendices A-B. Anyway, I think these are only the minor issues not affecting the overall quality of the paper, although they are numerous enough.
1. In Proposition 3.1 there should be $\zeta_t=\alpha_t \sigma_t / \beta_t^2$ (change numerator and denominator). First of all, it accords with what is written right before this proposition in lines 188-191. Second, it follows from the proof of this proposition (see formulae (36) and (40)).
2. It is hard to understand where the formula (10) follows from. Later, when you read Appendix B.2.2 and B.2.3, you finally find similar formulae for data/noise prediction cases with the proofs. I suggest to put reference to these appendices next to the formula (10). Also, $\Xi$ is defined in different way in line 177 of the main text and in line 1456 of the Appendix B.2.2 which is confusing.
3. I couldn't find formulae (12a-c) in (Foster et al. 2024) as said in lines 225-226. Maybe the authors could fix the reference or add explicit reference to the particular section of that paper. I could not find formulae (24) in that paper either.
4. Also, I think that the formulae (12-13) imply that in the formula (13c) the increments of Brownian motion and space-time Levy area (line 254) should be multiplied by $\Xi(\zeta_{n+1})$.
5. It is better to add references to solvers EDICT, BDIA, O-BELM, etc. (e.g. in Tables 1-3) to let the reader know which solver is described in which paper.
6. The main text does not contain references to Table 3.
7. $\hat{I}$ introduced in line 929 seems not to be used anywhere (at least in the places I have read).
8. In appendix A.3 the description of Shark is not quite clear, I could not understand what the second b-line in the Butcher tableau stand for.
9. It follows from line 1000 that noise prediction networks predict $\mathbb{E}[X_T|X_t=x]$, but the latter just equals to $x\cdot \alpha_T / \alpha_t$ as follows from the formulae (20-22) from the appendix of the paper "Variational Diffusion Models". I think that noise prediction networks predict $\mathbb{E}[\varepsilon_t | X_t=x]$ where $X_t=\alpha_t X_0 + \sigma_t \varepsilon_t$.
10. In the formula (30) RHS should be $\beta_T / \beta_t$. In line 1033 change of variables should look like $y_t=x_t\cdot \beta_T / \beta_t$. In the formula (31) the argument of the function f should be $y_t\cdot \beta_t / \beta_T$. In the formula (32) the numerator should be $\sigma_t\dot{\alpha_t}-\dot{\sigma_t}\alpha_t$.
11. In contrast with Rex (SDE), Rex (ODE) have pretty similar formulae for data and noise prediction cases, so maybe Lemma B.1 and B.2 could be unified. Anyway, their proofs are almost identical which leads to the typos, e.g. in the formula (61): in Lemma B.2 we consider noise prediction case but we see value like $x_{0|...}$ typical for data prediction case.
12. In the formula (73) in Ito's formula $1/2$ is missing before second-order derivative term.
13. Something is wrong in lines 1430-1434, Appendix B.2.2 is referenced in itself, and "reparamdatasde" artifact can be found.
14. In lines 1456-1457 missing brackets after exp.
15. It was difficult for me to understand why (103) holds until I remembered that $\Xi_t$ defined in lines 1456-1457 is expressed as (100). So, I suggest to remind the reader of that in the formula (103).
16. In Lemma B.10 in (138) there is $\alpha_n + 1$ instead of $\alpha_{n+1}$.
17. In the generalized Rex (SDE) in Proposition B.11 in the formula (147) there are no $2$ multipliers before $a$ and $b$ for noise prediction case as in Lemma B.10.

---

> ### Author Rebuttal · Authors · 2026-03-29
>
> We thank Reviewer empu for their highly detailed and thorough review and in particular for meticulously catching all of our typos and other mistakes. We are currently revising the paper to fix all of these and catch any other ones in the appendices. We answer the questions below and are happy to answer any further questions or provide additional clarification.
>
> ### Question 1
> Yes, you are correct, the trade-off with these reversible schemes is that they double the NFE of the underlying scheme and as the two steps are consecutively calculated they cannot be computed in parallel via batch mode. The upside is that the reconstruction error is zero (up to numerics) for even a small number of steps, whereas non-reversible schemes will take far more steps to converge requiring more NFE to achieve similar reconstruction performance.
>
> ### Question 2
> Thanks for pointing out that we forgot to discuss this. The number of steps only slightly increases (on average) with the reversible version. For the adaptive solvers in Table 3, averaging over all the generated 10k samples we have mean number of steps of 50.26 for Dopri5 and 51.28 steps for Rex (Dopri5).
>
> ### Question 3
> This is a great question! When we introduced the Foster-Reis-Strange SRK scheme we discussed it in the very general case of non-commutative noise in the Stratonovich convention. However, for additive noise SDEs (like our diffusion models) **the Ito and Stratonovich interpretations coincide**! This means we are free to convert to use whichever convention we wish. *N.B.*, in general one can convert between the two forms by adding a second-order correction term to the drift. In short we used Stratonovich calculus for legacy reasons as it is the preferred formalism within the reversible solver community; however, in our case the distinction is semantic and so all the analysis is valid to apply to the SDEs we worked with. The same reasoning would also extend to Rößler's scheme for additive noise SDEs.
>
>
> ### Miscellanea
>
> We are very thankful for the detailed notes on typos and related mistakes which we are working to fix along with streamlining clarity and consolidating the proofs into more general scenarios. We discuss a few of these corrections in more detail.
>
> **2. Equation (10) and abuse of notation w.r.t. $\Xi$.** We added (9) & (10) as a generalization of the data/noise prediction formulations to try and make the main paper a bit more clear. In a sense they follow in a similar spirit to (7); however, the two SDE formulations are more substantially different with fairly different $a(t)$ and $b(t)$ terms, so the abstraction is less clear. We are improving the writing to make things clearer. Thank you for flagging this.
>
> **3. Equation (12) and Foster's paper.** So this is an interesting one. The equation is taken directly from the diffrax library documentation, which cites [1] as the basis for the ShARK scheme and other related schemes. The more general formulation in the library docs does not appear explicitly in Foster's original paper, but is the generalization of the schemes proposed in [1, Equation (6.1)].
>
> **4. Missing $\Xi$ scaling term.** You are correct that is a typo and we've fixed that now.
>
> **5. Missing references in tables.** We generally agree with the reviewer; we omitted the references for space constraints as the author-year citations take up a lot of horizontal space which is limited in most of our tables. We will try and see if we can reformat them to fit; otherwise, we will try to make it more clear in the main text.
>
> **8. Extended Butcher tableaus.** The second line is for an adaptive step-sizing scheme, we explain this in lines 990-992 and provide a link to the diffrax implementation of the Butcher tableau in the paper.
>
> **9. Meaning of conditional expectation for noise prediction.** We follow the conventions used in the flow matching literature, in particular [2, Equation (4.59)] making the appropriate notational conversions from diffusion time to flow matching time. Importantly we define our interpolant as $\boldsymbol X_t = \alpha_t \boldsymbol X_0 + \sigma_t \boldsymbol X_T$ where $\boldsymbol X_T$ is drawn from our source (noise) distribution.
>
> ## References
>
> [1] Foster, J. M., Dos Reis, G., and Strange, C. High order
> splitting methods for sdes satisfying a commutativity con-
> dition. SIAM Journal on Numerical Analysis, 62(1):500–
> 532, 2024.
>
> [2] Lipman, Y., Havasi, M., Holderrieth, P., Shaul, N., Le, M.,
> Karrer, B., Chen, R. T., Lopez-Paz, D., Ben-Hamu, H.,
> and Gat, I. Flow matching guide and code. arXiv preprint
> arXiv:2412.06264, 2024.

---

> > ### Author Rebuttal · Reviewer_empu · 2026-04-03
> >
> > I keep my score 5.

---

> > > ### Author Response · Authors · 2026-04-06
> > >
> > > We sincerely thank Reviewer empu for their extraordinarily detailed and careful review, in particular the meticulous identification of typos and technical issues throughout the appendices. We are grateful that our clarifications on the NFE trade-off, adaptive step counts, and the Itô–Stratonovich equivalence for additive noise SDEs were satisfactory. We have incorporated all of the noted corrections into the revised manuscript.

---

### Official Review · Reviewer_BQnR · 2026-03-15

**Soundness:** 4
**Presentation:** 3
**Significance:** 3
**Originality:** 3
**Overall Recommendation:** 5
**Confidence:** 2

**Summary:**

This manuscript presents Rex, a recipe to construct reversible solvers for semi-linear SDEs and ODEs, such as the diffusion SDE/ODE. Specifically, the recipe needs to first devise a Runge-Kutta scheme of the SDE/ODE, named as Princeps. Then, by employing an algebraically reversible McCallum-Foster solver, one can construct a reversible solver based on the given SDE/ODE Princeps while preserving the convergence order of the underlying Runge-Kutta solver. Experimental results demonstrate that the proposed Rex recipe has superior performance compared to vanilla DDIM inversion and other exact inversion methods on unconditional, conditional, image editing, and Boltzmann distribution sampling tasks.

**Compliance With Llm Reviewing Policy:**

Affirmed.

**Key Questions For Authors:**

1. Could the authors comment on why the SDE could deliver better performance in the exact inversion task?
2. For prior works that tackle the SDE case mentioned in the introduction, can they benefit from the PRNG trick used in Rex? If so, what is the difference, and how will they perform compared with Rex?

**Limitations:**

yes

**Strengths And Weaknesses:**

## Strengths
* The main body is well-written, although it is a highly technical work; the presentation is clear and easy to follow for a general audience in the machine learning community.
* The proposed recipe is general and flexible to support constructing reversible solvers based on existing popular diffusion ODE/SDE solvers, which is a solid technical contribution to the community.
* The experiment session is well-executed, and the performance is convincing to demonstrate the advantage of the proposed art.

## Weaknesses
I do not have any major concerns about this manuscript. Please see the question part.

---

> ### Author Rebuttal · Authors · 2026-03-29
>
> We thank Reviewer BQnR for their review and insightful questions which we will answer below.
>
> ### Question 1
>
> For the actual inversion task of simply encoding the original samples $\boldsymbol x_0$ to some noisy state $\boldsymbol x_t$, $t \in (0, T]$ and then perform another ODE/SDE solve to bring the sample $\hat{\boldsymbol x}_0$ back to data, all reversible solvers should have identical performance, *i.e.*, $\boldsymbol x_0 = \hat{\boldsymbol x}_0$ up to numerics (as pointed out by Reviewer 6D4M) and VAE reconstruction error for latent models.
> However, once we perform alterations to $\boldsymbol x_t$ then the difference between reversible SDE and ODE schemes becomes more pronounced. *At a high level we can think of the noise in the SDE acting a stochastic regularizer, pulling deviant trajectories back into alignment with the data manifold.*
>
> In particular we can leverage prior analysis of non-reversible solvers.
> In [Theorem 3.2, 1] they state that given two time points $s, t \in [0, T]$ and $s > t$ errors in the edited distribution are preserved in KL.
>
> *I.e.,* let $p_s, p_t$ be the marginal distributions of the empirical PF ODE at time $s$ and $t$ and let $\hat p_s$ be our "edited" distribution and let $\hat p_t = \boldsymbol \Phi_{s,t}^\theta \sharp \hat p_s$ be the induced distribution created by our solver at time $t$ starting from time $s$. Then they show that under mild regularity conditions if $\hat p_s \neq p_s$
>
> $$D_{KL}(\hat p_t \\| p_t) = D_{KL}(\hat p_s \\| p_s).$$
>
> Conversely for SDEs in [Theorem 3.1, 1] they show that
>
> $$D_{KL}(\hat p_t \\| p_t) < D_{KL}(\hat p_s \\| p_s).$$
>
> In other words, these SDEs schemes can "correct" errors made in the editing process. This observation has been noted in other works as well which use non-reversible SDE solvers [2] and the empirical differences between ODEs and SDEs for editing has been more rigorously examined in [1].
>
> ### Question 2
>
> Unfortunately, the PRNG trick we employed (in particular we used Kidger's Brownian Interval [3]) is not that useful for the approaches used in [1,4] as they record the entire solve from data to noise. We will briefly summarize these approaches here and refer to our more detailed explanation in **Appendix E.3.4** for further details. Essentially, these approaches solve the forward SDE in closed form to get samples $\\{\boldsymbol x_n\\}_{n=1}^N$. They then rearrange the Euler-Maruyama SDE discretization (although this could be in theory some arbitrary explicit SRK scheme). Concretely, if we have an EM SDE update of form
>
> $$\boldsymbol x\_{n+1} = \boldsymbol x\_{n} + \boldsymbol f\_\theta(t\_n, \boldsymbol x\_n) (t\_{n+1} - t\_n) + g\_{t\_n} \boldsymbol z\_n, \quad \boldsymbol z\_n \sim \mathcal{N}(\boldsymbol 0, \boldsymbol I),$$
>
> then one could write
>
> $$\boldsymbol z_n = g_{t_n}^{-1} \left(\boldsymbol x\_{n+1} - \boldsymbol x\_{n} - \boldsymbol f\_\theta(t\_n, \boldsymbol x\_n) (t\_{n+1} - t\_n) \right),$$
>
> and then store $\\{\boldsymbol z_n\\}$. Using the PRNG trick here isn't too helpful as we don't need to recalculate the $\boldsymbol z_n$ term and so it would just be additional overhead. To summarize, as this scheme already stores the entire trajectory in memory there would not be much benefit to using the PRNG trick here to replace just drawing Gaussian samples. *N.B.*, the main benefits of these PRNG tricks are not storing the solution trajectory in memory and also enabling the use of adaptive step-sizing schemes [5].
>
>
> We are happy to answer any additional questions or provide further clarification.
>
> ## References
>
> [1] Nie, Shen, et al. "The Blessing of Randomness: SDE Beats ODE in General Diffusion-based Image Editing." The Twelfth International Conference on Learning Representations.
>
> [2] Meng, Chenlin, et al. "SDEdit: Guided Image Synthesis and Editing with Stochastic Differential Equations." International Conference on Learning Representations.
>
> [3] Kidger, Patrick, et al. "Efficient and accurate gradients for neural sdes." Advances in Neural Information Processing Systems 34 (2021): 18747-18761.
>
> [4] Wu, Chen Henry, and Fernando De la Torre. "Unifying diffusion models' latent space, with applications to cyclediffusion and guidance." arXiv preprint arXiv:2210.05559 (2022).
>
> [5] Jelinčič, Andraž, James Foster, and Patrick Kidger. "Single-seed generation of Brownian paths and integrals for adaptive and high order SDE solvers." arXiv preprint arXiv:2405.06464 (2024).

---

> > ### Author Rebuttal · Reviewer_BQnR · 2026-04-02
> >
> > Thank the authors for their response, and I will keep my current recommendation unchanged.

---

> > > ### Author Response · Authors · 2026-04-06
> > >
> > > We thank Reviewer BQnR for their thoughtful questions and for taking the time to carefully engage with our rebuttal. We are glad that our responses regarding the stochastic regularization benefits of SDE-based inversion and the scope of the PRNG trick were helpful and fully addressed your concerns. We appreciate your continued positive assessment of the work and your support for its acceptance.

---

### Decision · Program_Chairs · 2026-04-30

**Decision:**

Accept (spotlight)

**Comment:**

This submission introduces Rex, a method that turns ODE solvers into a reversible solvers for time-reversed  diffusion SDEs. All reviewers praise the methodological construction, presentation (including theory), and experimental results. Minor concerns have been resolved during the rebuttal period. Therefore, I recommend accepting this work.